# Application of logistic regression to simulate the influence of rainfall genesis on storm overflow operations: a probabilistic approach

Bartosz Szeląg[1], Roman Suligowski[2], Jan Studziński[3], Francesco Di Paola[4]

[1] Department of Geotechnics and Water Engineering, Kielce University of Technology, 25-314 Kielce, Poland
[2] Department of Hydrology and Geo-Information, Jan Kochanowski University, 25-406 Kielce, Poland
[3] Centre for Computer Science Applications in Environmental Engineering, Systems Research Institute Polish Academy of Sciences, 01-447 Warszawa, Poland
[4] Department of Civil, Architectural and Environmental Engineering, University of Naples Federico II, via Claudio 21, Napoli 80125, Italy

*Correspondence to*: Bartosz Szeląg (bszelag@tu.kielce.pl)

**Abstract.** One of the key parameters constituting the basis for the operational assessment of stormwater systems is the annual number of storm overflows. Since uncontrolled overflows are a source of pollution washed away from the surface of the catchment area, which leads to imbalanced receiving waters, there is a need for their prognosis and potential reduction. The paper presents a probabilistic model for simulating the annual number of storm overflows. In this model, an innovative solution
is to use the logistic regression method to analyse the impact of rainfall genesis on the functioning of a storm overflow in the example of a catchment located in the city of Kielce (central Poland).

The developed model consists of two independent elements. The first element of the model is a synthetic precipitation generator, in which the simulation of rainfall takes into account its genesis resulting from various processes and phenomena occurring in the troposphere. This approach makes it possible to account for the stochastic nature of rainfall in relation to the
annual number of events. The second element is the model of logistic regression, which can be used to model the storm overflow resulting from the occurrence of a single rainfall episode. The paper confirmed that storm overflow can be modelled based on data on the total rainfall and its duration. An alternative approach was also proposed, providing the possibility of predicting storm overflow only based on the average rainfall intensity. Substantial simplification in the simulation of the phenomenon under study was achieved compared with the works published in this area to date. It is worth noting that the
coefficients determined in the logit models have a physical interpretation, and the universal character of these models, facilitates their easy adaptation to other examined catchment areas.

The calculations made in the paper using the example of the examined catchment allowed an assessment of the influence of rainfall characteristics (depth, intensity, duration) of different genesis on the probability of storm overflow. Based on the obtained results, the range of the variability of the average rainfall intensity, which determines the storm overflow, and the
annual number of overflows resulting from the occurrence of rain of different genesis were defined. The results are suited for the implementation in the assessment of storm overflows only based on the genetic type of rainfall. The results may be used to develop warning systems in which information about the predicted rainfall genesis is an element of the assessment of the

rainwater system and its facilities. This approach is an original solution that has not yet been considered by other researchers. On the other hand, it represents an important simplification and an opportunity to reduce the amount of data to be measured.

## 1 Introduction

One of the important criteria for assessing the operation of stormwater systems is the annual number of storm overflows, which is confirmed by foreign guidelines (US EPA, 1995; Zabel et al., 2001; ÖWAV, 2003). The physics of the phenomenon is complex and depends on the dynamics of rainfall, the changes in rainfall over time and the characteristics of urban catchment areas with storm overflows. Currently, the annual number of overflows in the catchment areas can be assessed based on long-term observations of their operation (Price, 2008; Andrés-Doménech et al., 2010; Gemerith et al., 2011), but it is a costly

solution due to the need for the continuous monitoring of flows. An alternative approach is to build a hydrodynamic model of the catchment, which requires detailed data about the basin, precipitation from a long period (30 years according to DWA A–118) and flows to calibrate the model from a period of at least two years (Szeląg et al., 2016). Using the model of the catchment built based on long-term rainfall measurements, it is possible to perform the so-called continuous simulation, which will allow an estimation of the annual number of storm overflows. Such a solution may be a source of reliable estimation of the number

of overflows, although its technical implementation is complex and the results obtained (numerical simulations of the catchment model) are not always satisfactory (Romanowicz and Beven, 2006; Beven and Binley, 2014).

Considering the above, the article uses probabilistic models to predict the annual number of overflows, which take into account the stochastic nature of rainfall and the complex nature of runoff in urban catchment areas. This problem was discussed by Thorndahl and Willems (2008), who used the FORM (first-order reliability model) method to simulate the storm overflow event; its application in engineering practice is limited due to the complexity of its implementation. In subsequent works

addressing this problem, rainfall generators and episode of storm overflow models (Grum and Aalderink, 1999) were used to simulate the annual number of overflows. Multidimensional scaling methods and fractal geometry (Rupp et al., 2009; Licznar et al., 2015; Müller-Thomy and Haberlandt, 2015) are used to simulate rainfall series. An alternative solution is an approach based on the multidimensional distributions created based on theoretical distributions and copula functions (Vandenberghe et al., 2010; Vernieuwe et al., 2015). Despite numerous applications, these solutions are relatively complex and require expert

knowledge. For the storm overflow simulation, hydrodynamic models are usually used, and less frequently, empirical models are used (Szeląg et al., 2018). Nevertheless, this approach to the simulation of the annual number of overflows is very local and, in many cases, requires the construction of a catchment model. Szeląg et al. (2018) presented a model of simulation of storm overflow (single rainfall episode) determined by the logistic regression method. A significant disadvantage of the above mentioned solutions is the fact that in the probabilistic models, simplified rainfall generators were used and the variability of

precipitation characteristics was taken into account in relation to only one episode of overflow. At the same time, the issue of precipitation genesis was not addressed at all. Therefore, the question arises whether the information on the nature of precipitation (e.g., season of the year, precipitation genesis) could not find practical use in its modelling. It seems puzzling that the time course and the dynamics of the rainfall as the result of air masses advection (Serrano et al., 2009; Alhammoud et al.,

2014; Dayan et al., 2015) were not taken into account when rainfall generators were used to simulate storm overflows. The problem of the modelling of complex atmospheric phenomena is the subject of many works (Madsen et al., 1995; Paquet et al., 2006; Vicente-Serrano et al., 2009; Garavaglia et al., 2010; Abushandi and Merkel, 2011). The models created consider simulations of meteorological conditions changing in time and determining the distribution of temperature, pressure and humidity, which affects the dynamics of air advection and, consequently, the patterns of precipitation phenomena. According to the literature, the information concerning the genesis of precipitation allows for preliminary assessment (quantitative and qualitative) of the course and estimation of the average rainfall intensity (Suligowski, 2004). This information may be the basis for control of the systems and the development of an early warning system against the risks of flash floods. This problem has not been considered so far and, at the same time, it seems possible that modelling the functioning of the stormwater system and the facilities located in it based only on forecasts and identification of the rainfall genesis could be accomplished.

Rainfall is universally classified into three types (Sumner, 1988; Smith, 1993): convective, cyclonic and orographic. The main distinguishing feature between convective precipitation in an air mass and frontal precipitation in mid-latitudes is its spatial extent and duration. The range of convective precipitation associated with local air circulation is much smaller than in the case of travelling extratropical cyclones with weather fronts. Convective precipitation induced by single thunderstorm cells, their complexes or squall lines is short-lived but is characterized by high average intensity (Kane et al., 1987) and causes flash floods (Gaume et al., 2009; Marchi et al., 2010; Bryndal, 2015). On the other hand, the lifespan of the mechanisms of creating cyclonic precipitation is much longer than that of convective precipitation – on the order of days rather than hours. Hence, the effect of this is long-term rainfall with a high depth (Frame et al., 2017) often cause regional floods (Barredo, 2007).

The presented classification of precipitation types distinguished by Sumner (1988) due to the origin, developed for the British Isles and Western Europe, cannot be directly applied in practical hydrology in other regions of the continent, especially in its eastern and central parts due to exceptional variability of meteorological conditions occurring in the temperate zone of the warm transition climate – on the border of air masses coming from the Atlantic and continental masses from the east (Twardosz and Niedźwiedź, 2001; Niedźwiedź et al., 2009; Twardosz et al., 2011; Łupikasza, 2016). The analysis of maximum rainfall of different duration in Poland carried out at the end of the 1990s (Kupczyk and Suligowski, 1997, 2011), supplemented by an analysis of the synoptic situation (based on surface synoptic charts of Europe, published in the Daily Meteorological Bulletin of the Institute of Meteorology and Water Management – IMGW in Warsaw) and a calendar describing the types of atmospheric circulation together with air masses and air fronts (Niedźwiedź, 2019), led to the separation of three types of genetic precipitation: convective in air mass, frontal and generated in convergence zone.

Taking into account the above considerations, an innovative probabilistic model for calculating the number of storm overflows is proposed in the paper. This model is composed of two independent elements: a synthetic rainfall generator and a model for predicting storm overflows. The identification of overflows takes place only based on information about average rainfall intensity. In the rainfall generator, the genesis of rainfall is taken into account, which allows determining the curves that show the influence of rainfall genesis on the occurrence of storm overflow in a single rainfall episode. Based on the performed analyses, the ranges of variability of the average rainfall intensity assigned to rainfall of different genesis, for which storm

overflow may occur in the examined city catchment, were determined. The calculation experiments carried out in the study facilitated the determination of the influence of the distribution of the annual number of rainfall episodes of different genesis on the variability of the number of storm overflows.

## 2 Object of study

The object of analysis was a 62 ha urban catchment located in the south-eastern part of Kielce (Figure 1). The city covers an area of 109 km$^2$ and is located in Świętokrzyskie Voivodeship. The average population of the area in question is 21.4 people· ha$^{-1}$. Impervious areas, including pavements, roads, parking lots, roofs and school playgrounds, constitute 47.2 %. On the other hand, pervious areas, i.e., lawns, including green areas, occupy 52.8 %. On this basis, it was established that the weighted average value of the catchment retention is $d_{av}$= 3.81mm (Szeląg et al., 2016). The length of the main canal is 1.6 km, and its diameter changes between 0.60–1.25 m. The height difference in the catchment is 12.0 m and the average slope is 7.1 %. The analysis of the measurement data (2008-2017) concerning the analysed catchment area showed that the antecedent period lasted 0.16–60 days, and storms occurred 27–47 times a year. The average annual air temperature during the period under consideration varied from 8.1 to 9.6 °C. In addition, the analysis of the measurement data of flows recorded with the MES1 flow meter showed that in the antecedent periods, the temporary stream of the stormwater was from 0.001 to 0.009 m$^3$·s$^{-1}$, which indicates the occurrence of infiltration in the stormwater system under study.

Stormwater from the catchment is discharged to the Silnica River. Detailed information about the catchment can be found in Dąbkowski et al. (2010). The stormwater from the catchment is drained via channel S1 into the diversion chamber (DC). If the chamber level is less than $h = 0.42$ m, the stormwater is discharged into the stormwater treatment plant (STP). If the chamber level is filled above $h$, the stormwater is discharged by a storm overflow (OV) into the S2 canal, from where it flows into the Silnica River.

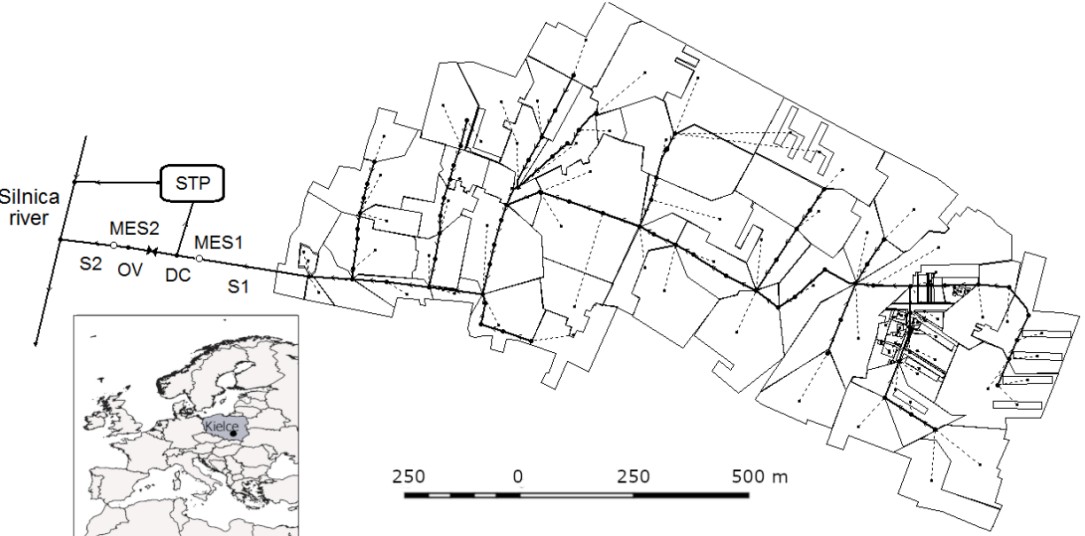

**Figure 1: Diagram of the analysed urban catchment.**

In the years 2009-2011, the amount of stormwater outflowing the catchment area was measured with the use of the MES1 flow meter located in the canal (S1) at a distance of 3.0 m from the inlet to the DC chamber. On the other hand, since 2015 in the inlet (S1) and discharge (S2) channels, MES1 and MES2 flowmeters measuring the values of filling and flows have been installed. A detailed description of the installed measuring equipment can be found in Szeląg (2016).

## 3 Rainfall data and analysis

The source material for the study presented was data from May – October 1961-2000 obtained from the records of a traditional float pluviograph (precipitation depth, duration, and mean intensity) installed at the IMGW meteorological station in Kielce. Only these data were taken into account in the conducted analyses, as the launch of a new device, the SEBA electronic rain gauge (tipping-bucket SEBA rain gauge), a few years later in the state measuring network resulted in the recording of significantly lower precipitation levels (by several percent). The data concerns events with high intensity and short duration (Kotowski et al., 2011). In the period 1961-2000 there were 1312 precipitation events in Kielce with a depth above 3 mm, with an average of 32.8 episodes per year. The greatest number of rainfall episodes (54) was observed in 1974.

In Kielce, precipitation classified as the first genetic type (convective in an air mass) lasts up to 150 min. This precipitation is caused by single convective cells with intensive ascending and descending currents, or by complexes of cells forming systems in the form of bands (squall line). The average annual number of these precipitation events in Kielce is 14.3 (1961-2000). These precipitation events are characterized by a low depth (Figure 2a) but rapidly increase with the increase in duration (max. 40.5 mm in 137 min). Due to the short duration of all rainfall episodes (from 5 min to 2.5 h), these events have a high intensity (median 7.97 L·s$^{-1}$·ha$^{-1}$; max. 97.8 L·s$^{-1}$·ha$^{-1}$) (Figure 2b).

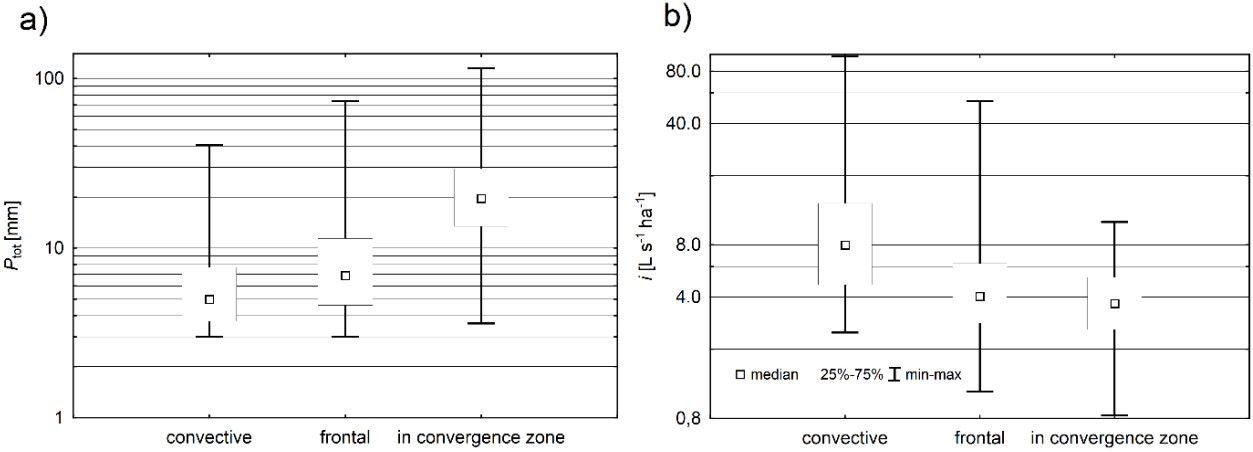

**Figure 2: Positional statistics of rainfall depth (a) and rainfall intensity (b) in genetic types.**

The second type (frontal rainfall) forms a group of precipitation in Kielce, in which the duration is very variable and ranges from 2.5 h to 10.5 h. These are the most frequent precipitation events in Kielce (average 16 events per year). These precipitation

events are associated with the movement of weather fronts, while the fast cold front together with dynamic processes in its zone leads to a high intensity of precipitation lasting 2.5–5.5 h (max. 53.8 L·s$^{-1}$·ha$^{-1}$). On the other hand, the 3 processes in the zone affected by the warm front usually generate higher precipitation levels, but due to their duration (5.5–10.5 h), two-fold lower precipitation intensity (max. up to 10.8 L·s$^{-1}$·ha$^{-1}$). The transformation of the air masses over the western part of the

continent, the lower speeds of the air advection and the weakening of the dynamics of processes in the front zone cause precipitation in Kielce to differ in intensity and duration in relation to precipitation defined by Sumner (1988) as cyclonic. On the other hand, the precipitation associated with convergence zone occurs in Kielce on average 2.3 times a year. These events are the result of the passage of deep low pressure centres or a series of low pressures with two clearly marked frontal areas. The high dynamics and magnitude of processes operating within them cause a long-term continuous precipitation

(> 10.5 h) that is recorded near the ground surface, with a clearly increasing sum (Figure 2a) and variability (weakening of intensity after passing a warm front, increase of intensity on a cold front), although with a low mean intensity (Figure 2b).

**4 Methodology**

An innovative probabilistic model is proposed for modelling the annual number of storm overflows (Figure 3). This model

allows for the prediction of the annual number of overflows and the simulation of the number of events per year, taking into account the genesis of rainfall, which is typical for countries located in central Europe and other regions of the world. Although the paper focuses on the genesis of rainfall developed by Kupczyk and Suligowski (1997, 2011), the proposed approach is universal. The distribution of rainfall data may be based on local conditions determining the advection of air masses, which has a key impact on the dynamics of rainfall events. The time range of particular rainfall groups can then be determined based

on meteorological, synoptic and statistical analysis in the periods of high precipitation totals or precipitation intensity in a given area (Llasat, 2001; Rigo and Llasat, 2004; Millán et al., 2005; Langer and Reimer, 2007; Federico et al., 2008; Lazri et al., 2012; Berg and Haerter, 2013). The classification of precipitation proposed by Sumner (1988) for Western Europe can also be used in such analyses. A literature review (Vendenberge et al., 2008) shows that the seasons of the year were taken into account in models for predicting rainfall distribution based on copula functions. The aspects related to the genesis of rainfall

have so far not been included in probabilistic models for the operational analysis of stormwater systems operation. The model proposed in the present study consists of three components. The first components are synthetic precipitation generators, which are realized in two variants. In the first variant it was assumed that the basis for the simulation of rainfall series is their genesis. In the second variant, precipitation is predicted regardless of its origin – in the annual cycle. Another component is a logit model, which is used to simulate the occurrence of a storm overflow. The third component of the model is a calculation block,

in which the annual number of overflows is simulated for the generated rainfall series. On this basis, distribution functions (CDF) are determined that describe the probability of non-exceeding the number of storm overflows.

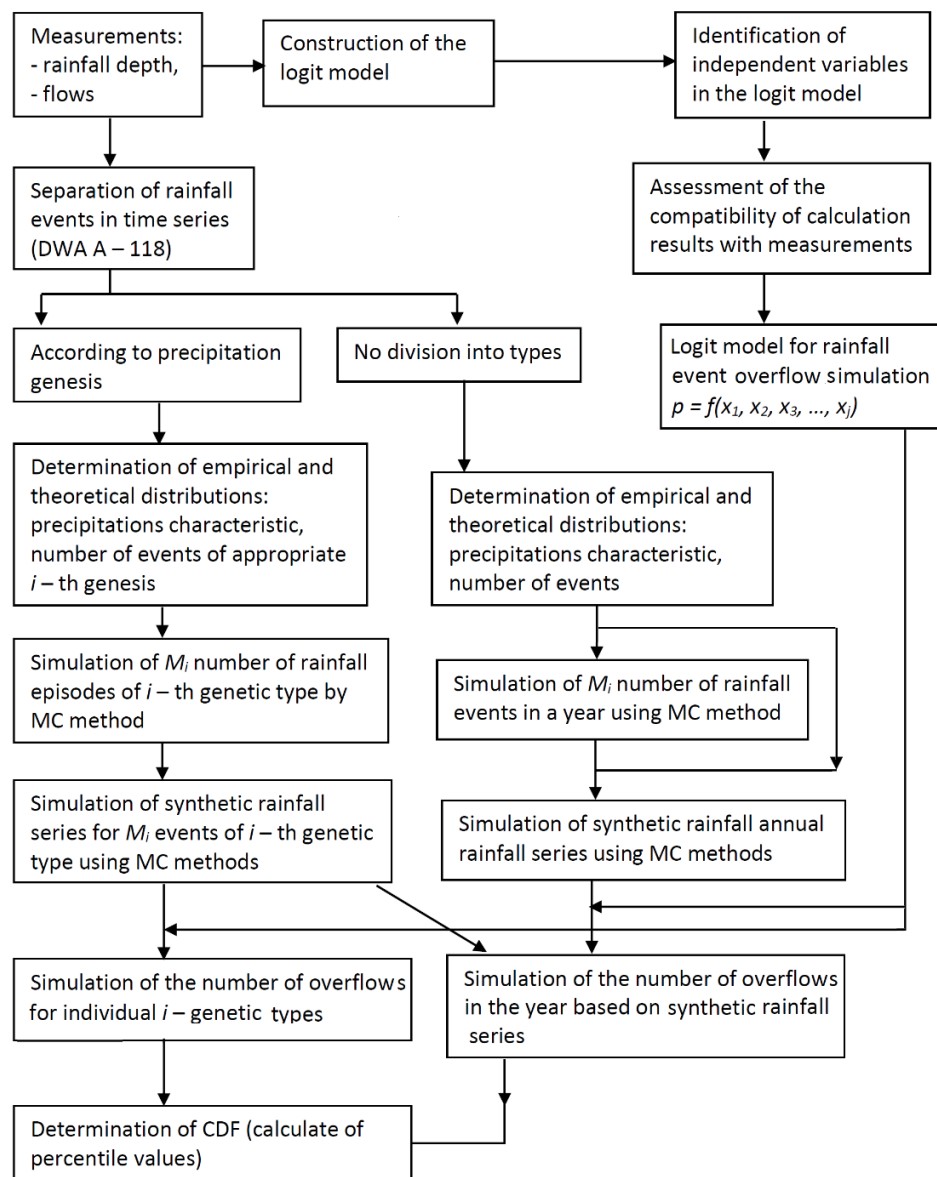

**Figure 3: Calculation diagram of the algorithm of building a probabilistic model for modelling the annual number of storm overflows.**

The proposed algorithm includes the following steps:

5   a) separation of precipitation events in rainfall measurement series,

b) identification of independent variables ($x_i$) in a logit model at the accepted confidence level and estimation of empirical coefficients,

c) determination of empirical distributions and theoretical rainfall characteristics of the different types of genetic precipitation (convective in air mass, frontal, and in convergence zone),

d) simulation of the number of precipitation events by means of the Monte Carlo method; or based on a fixed average number of rain episodes,

e) Monte Carlo simulation of rainfall characteristics for the number of rain events generated,

f) determination of the number of storm overflows for the generated rainfall series:

– per year,

– in different genetic types.

## 4.1 Simulation of the annual number of storm overflow events using a hydrodynamic model

One of the possible solutions allowing for the verification of empirical dependencies describing the operation of stormwater systems is the simulation of this operation with the use of a calibrated hydrodynamic model. It is an approach applied in engineering practice, which is confirmed by a number of works in this field (Bacchi et al., 2008; Andrés-Doménech et al., 2010). The simulations performed with a hydrodynamic model based on rainfall data allow the verification of the prediction capabilities of the probabilistic models designed to simulate the quantity and quality of stormwater, and the operation of separate objects located in the stormwater system (tanks, overflows).

Within the framework of the conducted analyses, a calibrated model of the catchment basin made in the SWMM (storm water management model) program was used to verify the annual number of storm overflows (Figure 1). The total area of the analysed catchment is 62 ha, while the area of partial catchments ranges from 0.12 ha to 2.10 ha. The number of stormwater junctions in the catchment area is 200, and the number of stormwater pipes is 72. The retention depth of the imperviousness surfaces of the catchment area is 2.5 mm and the retention depth of the pervious surfaces is 6.0 mm. The roughness coefficient of impervious areas is equal to 0.025 $m^{-1/3} \cdot s$, and that of pervious areas is 0.250 $m^{-1/3} \cdot s$. The roughness coefficient of the stormwater channels is equal to 0.018 $m^{-1/3} \cdot s$.

The results of the simulation with the hydrodynamic model were compared with the results of calculations made with the use of logit models for the data from the examined catchment from the period 2008-2017, which allowed the verification of the determined relation $p = f(x_1, x_2, x_3, ..., x_i)$. Simultaneously, using the measurement data from the period of 1961-2000, a simulation of the annual number of storm overflow events was performed, taking into account the precipitation genesis, which enabled the verification of the developed probabilistic model.

## 4.2 Logistics regression

The logistic regression model is also called the binomial logit model and is usually used to simulate binary data. Therefore, this model is commonly used for probability modelling. The logit model is often used to simulate phenomena and processes in medicine, social sciences and psychology (Bagley et al., 2001). This model is also successfully used to model processes in ecology, water engineering, geotechnics (Hayer et al., 2013, Inglemo et al., 2011) and wastewater treatment (Bayo et al., 2006;

Szeląg et al., 2019). This model is also used to simulate the influence of constructions on flow processes (Szeląg et al., 2018). The logit model takes the following form:

$$p = \frac{\exp(X)}{1+\exp(X)} = \frac{\exp(\sum_{i=1}^{j} \alpha_i \cdot x_i + \alpha_0)}{1+\exp(\sum_{i=1}^{j} \alpha_i \cdot x_i + \alpha_0)} \qquad (1)$$

where the following are defined: p [–] – probability of occurrence of storm overflow in a single rainfall episode, $\alpha_i$ – empirical
coefficients estimated with the method of maximum likelihood estimation (MLE), $x_i$ – independent variables, which in this paper include: rainfall depth ($P_{tot}$ [mm]), rainfall duration ($t_r$ [min]), and average intensity of rainfall event ($i$ [L·s$^{-1}$·ha$^{-1}$]). The calculations assume that a storm overflow occurs when $p$ is not less than 0.50, which corresponds to the following condition:

$$\sum_{i=1}^{j} \alpha_i \cdot x_i + \alpha_0 = 0 \qquad (2)$$

To evaluate the predictive capacity of the logit model, the following measures were used to match the calculation results to the measurements: sensitivity – SENS (determines the correctness of data classification in the set of data, including events when a storm overflow occurred), specificity – SPEC (determines the correctness of data classification in the set of data constituting cases when no storm overflow occurred) and counting error – $R_z^2$ (determines the correctness of identification of the simulation of events: storm overflow occurred/did not occur). These measures are discussed in detail in McFadden's paper (1963).

Two variants of the logit model were considered. In the first variant, the rainfall depth and its duration were assumed to be independent variables, as described in Szeląg et al. (2018). The second variant is a simplification. This variant considers only a single independent variable, i.e., average rainfall intensity. In the urban catchment area, continuous flow measurements were carried out in the period 2008-2011 (69 storm overflows during 188 rainfall events were separated at that time), whereas in the years 2012-2014, only fillings in the diversion chamber were measured (42 overflows during 93 rainfall episodes were
separated). The reason for this was the construction works carried out in the analysed catchment and a large amount of suspended solids limiting the operation of the measuring devices. Since 2015, MES1 and MES2 flow meters have been installed, which also allow the measurement of the volume of stormwater discharged by overflow. Thus, based on data from the period 2008-2014, a logit model was developed, while data from the period 2015-2017 were used to verify it.

**4.3 Separation of the rain event and synthetic rainfall simulator**

One of the basic conditions allowing for the completion of a synthetic precipitation generator is the separation of single independent rain events (episodes) in rainfall time series. For this purpose, the guidelines DWA A–118 (2006) were used, in which the basis for precipitation separation is a minimum antecedent period of 4 h. As a precipitation event, such a precipitation episode has been assumed for which the rainfall depth is not less than 3.0 mm (Fu and Kapelan, 2013; Fu et al., 2014).

Based on the precipitation observed in the period 1961-2000, independent precipitation events were separated, for which statistical distributions were determined. To obtain the best possible fit of the theoretical data to the empirical precipitation

data (including: rainfall depth – $P_{tot}$, rainfall duration – $t_r$ and average rainfall intensity – $i$, for precipitation of appropriate genesis), the following statistical distributions were considered (Adams and Papa, 2000; Bacchi et al., 2008; Domenech et al., 2010): Weibull, chi-square, exponential, GEV, Gumbel, gamma, Johnson, log-normal, Pareto and beta. Kolmogorov-Smirnov and chi-square tests were used to assess the conformity of the empirical and theoretical distributions. The empirical

distributions were also determined, and the theoretical distributions were adjusted to the data describing the number of precipitation events in the year and the number of episodes resulting from convective precipitation, frontal rainfall (warm and cold fronts), and rainfall in convergence zone. Within the theoretical distributions, Poisson, geometric, Bernoulli and binomial distributions were considered. Kolmogorov-Smirnov tests were used to assess the conformity of empirical and theoretical distributions.

Taking into account the computational algorithm described at the beginning of Section 4 (Methodology), based on the determined distributions of the theoretical rainfall characteristics describing the operation of a storm overflow, a model for the simulation of a synthetic rainfall series was adopted for further analysis. The simulations carried out for this purpose included the Monte Carlo method with modification of Iman-Conover (1982). This model provides the possibility simulating the independent variables based on the determined theoretical distributions. In this method, the variability of the considered

variables is described by boundary (theoretical) distributions, and the basis for the evaluation of their correlation is the Spearman correlation coefficient. The conditions that must be met in order for the results obtained to be considered correct, are as follows:

– in the data obtained from simulation and measurements, the mean values $(\mu_1(x_1), \mu_2(x_2),...,\mu_i(x_i))_s$ and the standard deviations $(\sigma_1(x_1), \sigma_2(x_2),...,\sigma_i(x_i))_s$ of the variables $(x_i)$ considered in $j$ samples do not differ by more than 5 %.

– the theoretical distributions of $x_i$ variables obtained from simulation should be consistent with those obtained from measurements; in order to meet this condition it is recommended to use the Kolmogorov-Smirnov (KS) test.

– the value of the correlation coefficient (R) between the individual dependent variables $(x_i)$ obtained for the data from Monte Carlo (MC) simulation does not differ by more than 5 % from the value of R obtained for empirical data.

If the above mentioned conditions are met, the results of the simulation performed with the Iman-Conover (IC) method can be

considered correct. If these conditions are not met, the sample size of the MC needs to be increased (Wu and Tsang, 2004). To limit the sample size and improve the efficiency of the Iman-Conover (IC) algorithm, a modification has been developed by using the Latin hybercube (LH) algorithm, which is part of the layered sampling methods aimed at improving the "uniformity" of the numbers generated from the boundary distributions.

Based on the determined boundary distributions of rainfall characteristics, the simulations of the synthetic rainfall series were

performed with the use of the Monte Carlo (MC) method with Iman-Conover (IC) modifications and taking into account the Latin hybercube (LH) algorithm.

## 4.4 Simulator of annual synthetic rainfall series

The paper presents two approaches to the simulation of rainfall series. In the first approach to the simulation, the average annual number of precipitation events (convective rainfall in air mass, frontal rainfall and rainfall in convergence zone was assumed (Figure 4). In the second approach, it was assumed that the number of rainfall events of the appropriate genetic type is stochastic.

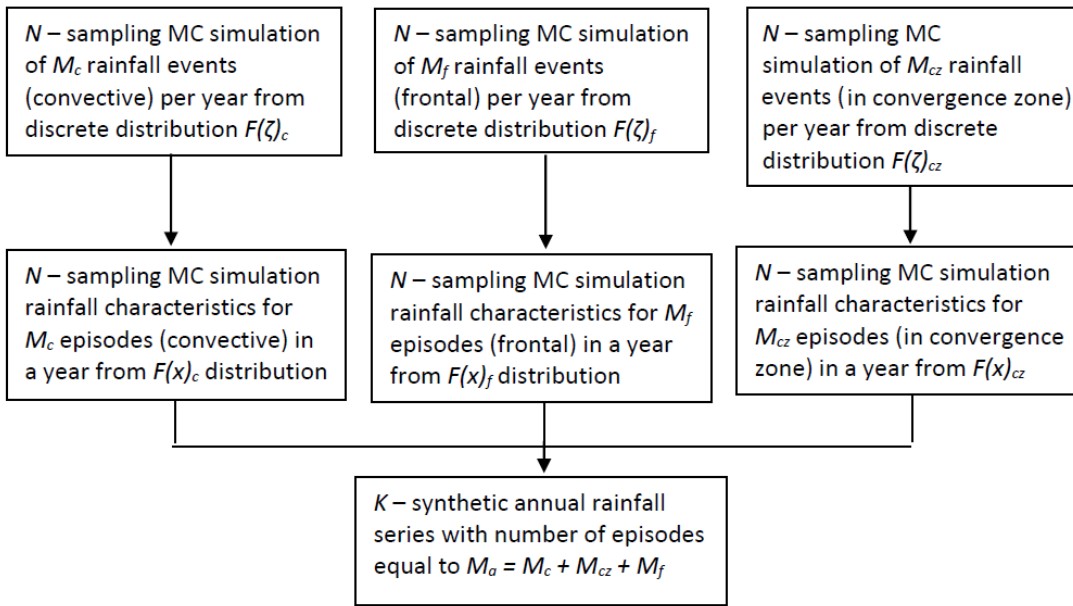

**Figure 4: Algorithm of the simulation of annual rainfall series, taking into account the generation of rainfall. $N$ – number of modelled samples for $M$ rainfall episodes (N= 500); $K$ – number of samples of rainfall characteristics modelled with IC + LH method ($K$ = 1000); Explanations of the other symbols are in the Appendix.**

Within the framework of the conducted analyses, an attempt was made to establish simplifications related to the omission of the impact of rainfall genesis on the annual number of storm overflows. Based on the calculation algorithm (Figures 3, 4) and the proposed rainfall generator, several simulation variants were considered. In the simplest case, all rainfall events were considered regardless of their genesis. Thus, the average number of rain events per year – $M=const(x_i)$ – was assumed. In the next case rainfall of different genesis (convective, frontal, and in convergence zone) was taken into account. This approach reflects the different dynamics of rainfall and the changing number of rainfall episodes in the year - $M=var(x_i)$. In addition, an attempt was made to analyse the intermediate variants. The differentiated number of rainfall events in the year was taken into account, but the genesis – $M=f(x_i, M=var)$ – was omitted.

## 5 Results

Following the above mentioned methodology concerning the structure of the individual elements of the probabilistic model (Figure 3), calculations were made. The calculations consisted of the determination of the logit model, the identification of

empirical distributions and theoretical rainfall characteristics, the simulation of synthetic rainfall series with the inclusion of rainfall genesis and the modelling of the annual number of overflows.

## 5.1 Logit model and verification

Based on the results of the measurements of storm overflow (OV) and rainfall described in detail in the sections above and in the works of Szelag et al. (2013, 2018), independent rain events were separated, on the basis of which the logit model was determined. In the case of independent variables such as $P_{tot}$ and $t_r$, the logit model takes the form (Szeląg et al., 2018):

$$p = \frac{\exp(0.566 \cdot P_{tot} - 0.004 \cdot t_r - 2.152)}{1 + \exp(0.566 \cdot P_{tot} - 0.004 \cdot t_r - 2.152)} \tag{3}$$

The logit model is characterized by satisfactory predictive abilities because the value of SPEC= 96.90 % (out of 106 overflows
in 103 episodes the model correctly identified the event), SENS= 98.20 % (out of 165 events, 162 overflows were correctly classified) and $R_z^2$= 97.78 % (out of 271 observed episodes, in 265 events the calculation results were consistent with the measurements).

If only the average rainfall intensity ($i$) was included in the analyses, a logit model of the form was obtained:

$$p = \frac{\exp(0.312 \cdot i - 1.257)}{1 + \exp(0.312 \cdot i - 1.257)} \tag{4}$$

An interesting result of the research is the fact that it is possible to simulate storm overflows with satisfactory accuracy only on the basis of rainfall intensity ($i$). This fact is important from the perspective of constructing models for modelling rainfall. The result obtained in the study indicates the possibility of significant simplification of the construction of the probabilistic model for the simulation of the annual number of overflows. This model is also characterized by satisfactory predictive abilities, because the value of SPEC= 84.90 % (out of 106 overflows, in 90 episodes the model correctly identified the event),
SENS= 90.20 % (out of 165 events, 148 overflows were correctly classified using the model) and $R_z^2$= 87.82 % (out of 271 observed episodes, in 229 events, the calculation results were consistent with the measurements).

The values of the SENS (sensitivity) and SPEC (specificity) coefficients are usually calculated to assess the predictive capability of the models. However, SENS = $f$(SPEC) may also be used for this purpose. In this case, the greater the area value (the maximum value is AUC = 1) between the SENS = SPEC and SENS = $f$(SPEC) curves is, the more accurate the model
will be (Figure 5). To verify the predictive capabilities of the logit models and their dependencies, a calibrated hydrodynamic model was used. The results of the simulation obtained with the hydrodynamic model for data from the period 2008-2017 were compared with the results of the calculations made with the use of the logit models $p = f(i)$ and $p = f(P_{tot}, t_r)$ (see Table 1).

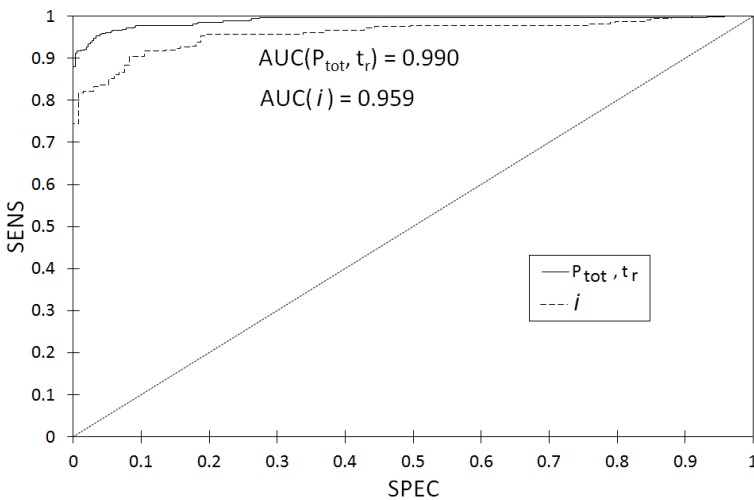

**Figure 5: Relation SENS = *f*(SPEC) for the logit curves *p* = *f*(*i*) and *p* = *f*($P_{tot}$, $t_r$)**

**Table 1: Comparison of the measurements of the annual number of overflow events with the calculations results obtained with hydrodynamic and logit models**

| Year | $Z_a$ | $Z_{SWMM}$ | $Z_{logit(P,t)}$ | $Z_{logit(i)}$ |
|------|-------|------------|------------------|----------------|
| 2008 | 13 | 15 | 14 | 12 |
| 2009 | 15 | 16 | 16 | 14 |
| 2010 | 17 | 18 | 19 | 16 |
| 2011 | 19 | 20 | 19 | 17 |
| 2012 | 13[*] | 21 | 20/14[*] | 19/11[*] |
| 2013 | 13[*] | 22 | 20/13[*] | 20/11[*] |
| 2014 | 16[*] | 29 | 28/16[*] | 27/14[*] |
| 2015 | 25 | 26 | 26 | 23 |
| 2016 | 26 | 22 | 21 | 24 |
| 2017 | 16 | 17 | 17 | 14 |

[*] – storm overflow events in the period 2012-2014 determined on the basis of the maximum filling of the diversion chamber (DC). Explanations of symbols are in the Appendix.

The above relations (3) and (4) are of local character and can be applied in principle only in relation to the analysed catchment area. With this in mind, an attempt was made to build a universal model by modifying these relations. Based on theoretical considerations of Thorndahl and Willems (2008), who investigated the relationships between the characteristics of catchment areas, including the limit retention that determines the volume of overflow, the following equations were determined:

$$0.566\,P_{tot} - 0.004\,t_r - 2.152 = 0 \ / \cdot \frac{1}{0.566} = P_{tot} - 0.007\,t_r - 3.802 = 0 \tag{5}$$

$$P_{tot} - 0.007t_r - 3.802 = P_{tot} - 0.007t_r - d_{av} = 0 \tag{6}$$

$$0.312\,i - 1.257 = 0\ /\ \cdot \frac{1}{0.312} = i - 4.03 = 0 \tag{7}$$

$$i - 4.03 = i - d_{\mathrm{av}} = 0 \tag{8}$$

On the basis of these relationships, it can be concluded that the values of the intercept obtained in them are similar to the weighted average value of the catchment retention ($d_{\mathrm{av}}$). The relative difference between the values of intercept and retention

of the catchment area does not exceed 5 %. The simulation calculations of the annual number of overflow events in the tested catchment area confirm the above statement (Table 1). In addition, this fact is supported by the results obtained in Figure 5, which indicate that the results of the simulation of the annual number of overflow events using the hydrodynamic model are within the scope of a probabilistic solution. However, because the presented analyses were performed only for a single catchment, it is necessary to verify the obtained results using examples of other catchments with different physical and

geographical characteristics.

**5.2 Identification of the empirical and theoretical distributions of the selected rainfall characteristics**

Table 2 presents the results of the Kolmogorov-Smirnov (KS) and chi-square (Chi) tests of fitting the empirical distributions to theoretical distributions for the rainfall characteristics ($P_{\mathrm{tot}}$, $t_{\mathrm{r}}$, $i$, $M$) depending on the genesis of precipitation and shows the

determined parameters in equations describing theoretical distributions.

Based on the data in Table 2 it appears that the empirical distributions can be expressed by means of the following theoretical distributions:

Weibull:

$$f(x) = \frac{\beta}{\gamma} \cdot \left(\frac{x-\mu}{\gamma}\right)^{\beta-1} \cdot \mathrm{e}^{-\left(\frac{x-\mu}{\gamma}\right)^{\beta}} \tag{9}$$

log-normal:

$$f(x) = \frac{1}{x\cdot\sigma\cdot\sqrt{2\cdot\pi}} \cdot \mathrm{e}^{\frac{(\ln(x)-\mu)^2}{2\cdot\sigma^2}} \tag{10}$$

generalized maximum value (GEV):

$$f(x) = \frac{1}{\sigma} \cdot (1 + \xi \cdot (x-\mu)/\sigma)^{\frac{-1}{\xi}-1} \cdot \mathrm{e}^{\left[-1\cdot(1+\xi\cdot(x-\mu)/\sigma)^{\frac{-1}{\xi}}\right]} \tag{11}$$

beta:

$$f(x) = \frac{\Gamma(\alpha+\beta)}{\Gamma(\alpha)\cdot\Gamma(\beta)} \cdot \frac{(x-c)^{\alpha-1}\cdot(x-d)^{\beta-1}}{(d-c)^{\alpha+\beta-1}} \tag{12}$$

Poisson:

$$f(x) = \frac{\mathrm{e}^{-\lambda}\cdot\lambda^x}{x!} \tag{13}$$

where the following are defined: $\beta$, $\gamma$, $\lambda$, $\mu$, $\sigma$, $\xi$ – parameters of distributions determined by the maximum likelihood estimation (MLE) method.

**Table 2: Results of KS and Chi tests and parameters of theoretical models for considered random variables ($P_{tot}$, $t_r$, $i$, $M$) depending on rainfall genesis.**

| Variable | Distribution | Model parameters | $p$ (KS) | $p$ (Chi) |
|---|---|---|---|---|
| $P_{tot}$ (all events) | Weibull | $\beta = 0.772$; $\gamma = 5.158$; $\mu = 3.00$ | 0.121 | 0.096 |
| $t_r$ (all events) | GEV | $\zeta = 0.466$; $\sigma = 129.355$; $\mu = 108$ | 0.096 | 0.071 |
| $i$ (all events) | log-normal | $\sigma = 1.932$; $\mu = 0.855$ | 0.112 | 0.096 |
| $M$ (all events) | Poisson | $\lambda = 32.80$ | 0.624 | 0.053 |
| $P_{tot}$ (convective) | Weibull | $\beta = 0.821$; $\gamma = 3.102$; $\mu = 3.00$ | 0.477 | 0.412 |
| $t_r$ (convective) | beta | $\alpha = 1.391$; $\beta = 1.173$; $c = 5.5$; $d = 150$ | 0.268 | 0.173 |
| $i$ (convective) | log-normal | $\sigma = 2.557$; $\mu = 0.694$ | 0.238 | 0.211 |
| $M$ (convective) | Poisson | $\lambda = 14.33$ | 0.871 | 0.756 |
| $P_{tot}$ (frontal) | Weibull | $\beta = 0.968$; $\gamma = 6.054$; $\mu = 3.00$ | 0.353 | 0.314 |
| $t_r$ (frontal) | Weibull | $\beta = 1.201$; $\gamma = 164.99$; $\mu = 150$ | 0.639 | 0.589 |
| $i$ (frontal) | log-normal | $\sigma = 1.485$; $\mu = 0.644$ | 0.906 | 0.878 |
| $M$ (frontal) | Poisson | $\lambda = 15.95$ | 0.372 | 0.831 |
| $P_{tot}$ (frontal, type I) | Weibull | $\beta = 0.862$; $\gamma = 4.535$; $\mu = 3.00$ | 0.631 | 0.425 |
| $t_r$ (frontal, type I) | beta | $\alpha = 1.221$; $\beta = 1.372$; $c = 150$; $d = 270$ | 0.200 | 0.145 |
| $i$ (frontal, type I) | log-normal | $\sigma = 1.701$; $\mu = 0.612$ | 0.104 | 0.085 |
| $P_{tot}$ (frontal, type II) | Weibull | $\beta = 1.065$; $\gamma = 7.222$; $\mu = 3.00$ | 0.397 | 0.342 |
| $t_r$ (frontal, type II) | beta | $\alpha = 0.829$; $\beta = 1.562$; $c = 266$; $d = 650$ | 0.270 | 0.226 |
| $i$ (frontal, type II) | log-normal | $\sigma = 1.289$; $\mu = 0.611$ | 0.059 | 0.056 |
| $P_{tot}$ (in convergence zone) | log-normal | $\sigma = 0.603$; $\mu = 3.00$ | 0.969 | 0.856 |
| $t_r$ (in convergence zone) | Weibull | $\beta = 0.802$; $\gamma = 276.138$; $\mu = 650$ | 0.947 | 0.879 |
| $i$ (in convergence zone) | log-normal | $\sigma = 1.296$; $\mu = 0.497$ | 0.942 | 0.923 |
| $M$ (in convergence zone) | Poisson | $\lambda = 2.55$ | 0.067 | 0.652 |

where: type I – cold front, type II – warm front.

The calculations performed (Table 2) showed that in most cases the Weibull distribution in the form (eq. 9) is the best suited to the empirical data describing the variability of the total depth of rainfall ($P_{tot}$) in a rainfall episode. In addition, the variation in rainfall duration of the episodes resulting from rainfall of different genesis is described, in most cases, by the Weibull distribution and only in the case of the data measured over an annual cycle is it expressed better by the GEV distribution (eq.

10    11). The values of the rainfall intensities of different genesis are, in most cases, described by log-normal distributions (eq. 10), whereas in the case of the rain intensity caused by frontal precipitation, its dynamics are described by a generalized distribution of extreme values. Satisfactory adjustment of the calculation results to the measurements of the tested variables of a continuous nature is confirmed by the curves shown in Figures 6–7.

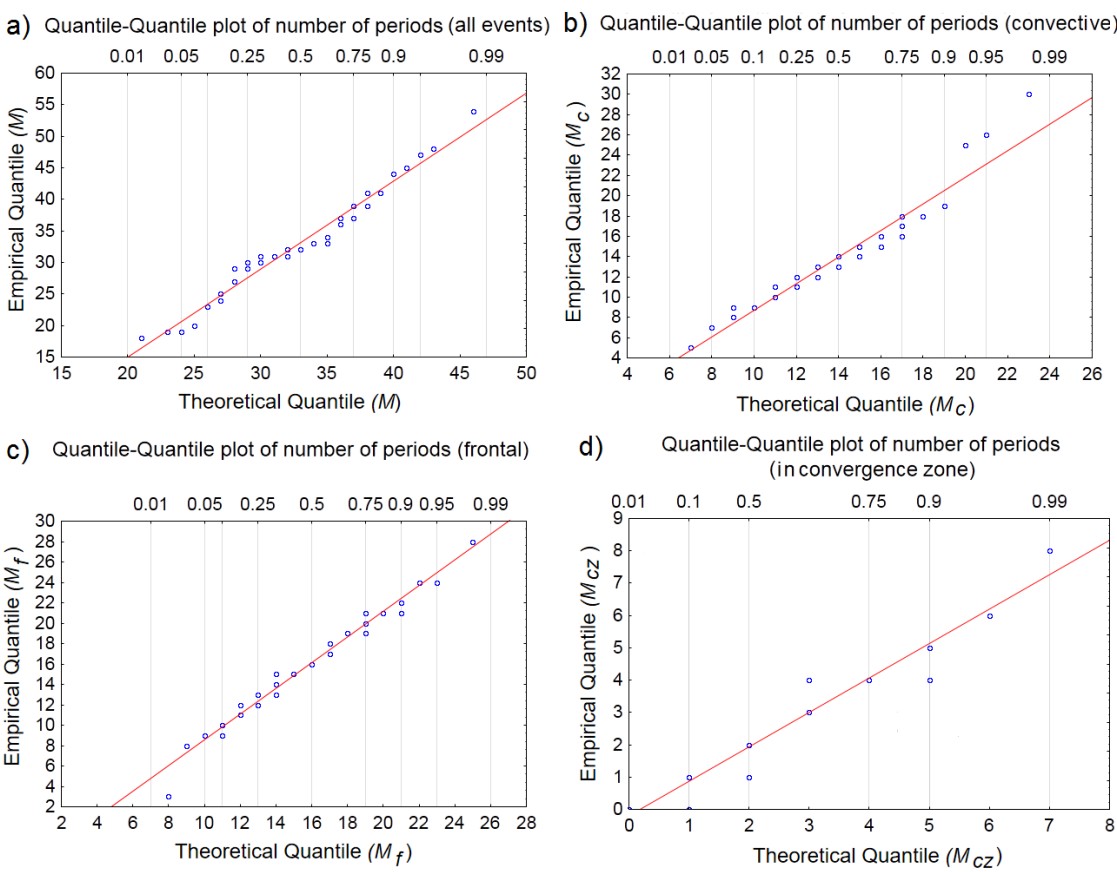

**Figure 6: Comparison of empirical and theoretical quantiles concerning the number of rainfall episodes and distinguishing rainfall types: (a) all events (b) convective, (c) frontal, (d) in convergence zone.**

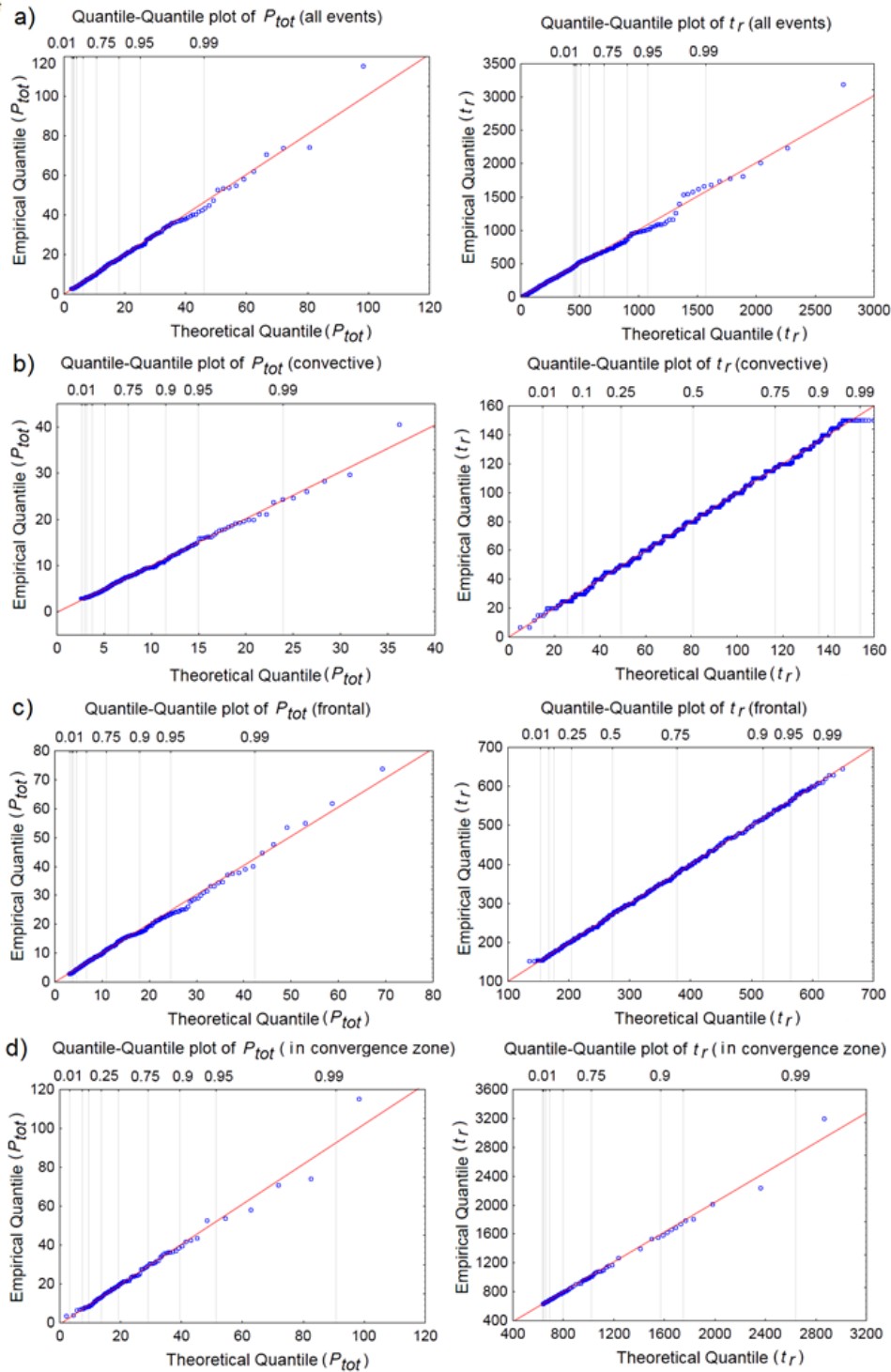

**Figure 7: Comparison of empirical and theoretical quantiles concerning $P_{tot}$ and $t_r$ values for: (a) all events, (b) convective, (c) frontal, and (d) rainfall in convergence zone.**

The curves in Figure 6 and Figure 7, illustrating the fitting of the quantile values obtained from the measurements and based on the specific theoretical distributions in most of the cases considered, show a high agreement of the data in relation to the number of rainfall events per year ($M_{c,f,cz}$), precipitation altitude and duration ($P_{tot}$, $t_r$). For the distribution describing the annual number of events, the values of the minimal quantiles are overstated by 5, while the maximal values are understated by 7. As a result, this distribution may affect the results of the determined number of storm overflows.

### 5.3. Impact of rainfall genesis on the probability of storm overflow occurrence

Based on the models determined for the modelling of storm overflow and the determined theoretical rainfall distributions, the impact of rainfall genesis on the occurrence of a single storm overflow was calculated in the first place. Taking into account the different predictive abilities of the logit models obtained, the analyses were limited to calculations with a model that best represented the existing state. The results of the simulations are shown in Figure 8.

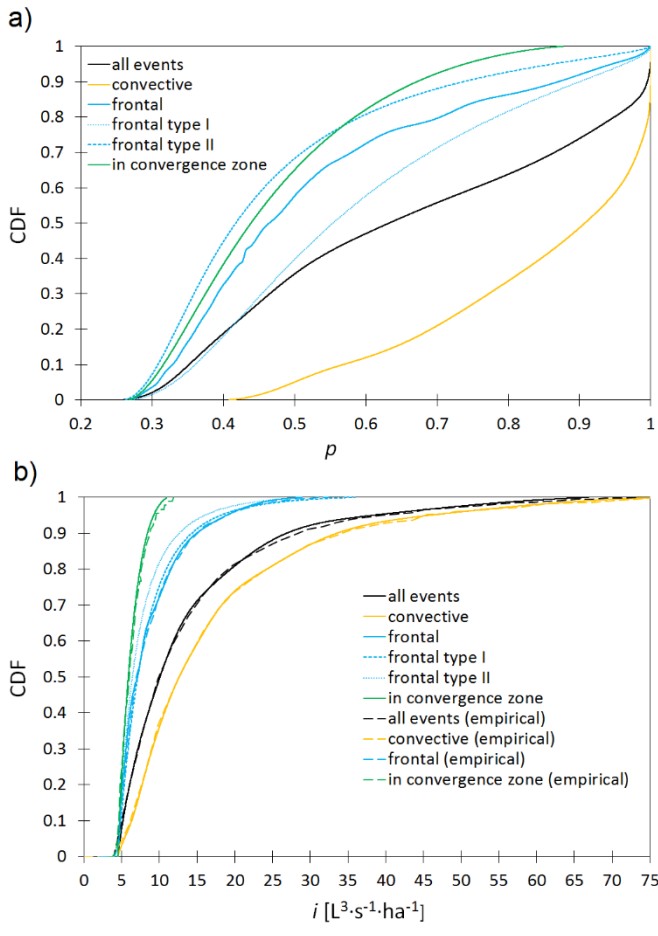

**Figure 8: (a) Impact of rainfall genesis on: a) the probability of storm overflow, (b) rainfall intensity distribution determining storm overflow in a single episode.**

Within the framework of the conducted analyses, the division of frontal rainfall events of the duration not longer than 4.5 h (related to the cold front – frontal type I) and exceeding the given value (due to the displacement of the warm front – frontal type II) was additionally distinguished. Based on the curves shown in Figure 8a, it can be concluded that the greater probability of storm overflow was obtained for events caused by convective rainfall episodes. For precipitation of this genesis, the

minimum probability of storm overflow is not less than 0.40 and the percentile value 0.50 is as high as 0.90. In order to verify the obtained relationship using a probabilistic model, empirical curves plotted based on the measurement data (Figure 8b). Curves obtained on the basis of measurements and simulations show high compliance, which confirms the usefulness of the determined model.

**5.4. Impact of rainfall genesis on the average rainfall intensity occurrence of a storm overflow**

Based on the results of the calculations obtained above, the ranges of the variability of the average rainfall intensity ($i$) were determined depending on the genesis conditioning the occurrence of a storm overflow (Figure 8). Based on the determined curves, it can be stated that among the analysed rainfall, the highest range of variability of values $i = 4.49$–$75.00$ $L{\cdot}s^{-1}{\cdot}ha^{-1}$ was obtained for convective rainfall (Figure 8b). From the point of view of modelling and forecasting, this result seems to be

interesting because the information about the genesis of precipitation (it is now generally available information) enables assessing, to some extent, the conditions of the functioning stormwater systems. Based on the analysis of the variability of curves from Figure 8a and 8b, it can be stated that the lowest probability of storm overflow is obtained in the case of rainfall in the convergence zone $i = 4.49$–$10.94$ $L{\cdot}s^{-1}{\cdot}ha^{-1}$ and rainfall in the warm front zone (with a duration longer than 4.5 h), for which $i = 4.49$–$28.50$ $L{\cdot}s^{-1}{\cdot}ha^{-1}$. In the analysed case, it was determined that for $p = 0$–$0.57$ in the examined catchment the

lowest probability value is obtained for episodes caused by precipitation in the warm front zone ($t_r \geq 4.5$ h), while for higher $p$ values, the lowest probability of overflow occurrence is obtained as a result of rainfall in the convergence zone (Figure 8a). This fact can be justified by the physics of the studied phenomenon. In the case of rainfall in convergence zone a lasting relatively long time, the average rainfall intensity is lower than in the case of frontal rainfall with a rainfall duration longer than 4.5 h.

Dependencies requiring commentary were also obtained for frontal precipitation ($t_r \geq 2.5$ h), for which $i = 4.49$–$35.50$ $L{\cdot}s^{-1}{\cdot}ha^{-1}$. The conducted analyses showed that with the increase in $p$ values, the difference in absolute values of the probability of storm overflow caused by precipitation being the effect of the cold ($t_r = 2.5$–$4.5$ h) and warm ($t_r \geq 4.5$ h) front increases (Figures 8a, 8b). The difference between individual values of $p$ ($p_I$–$p_{II}$) between individual curves increases to 0.20 for $p = 0.60$ and then decreases. It is worth noting that this difference (its maximum value) includes episodes in the case of frontal

precipitation caused by a cold and warm front (it corresponds to $i = 4.49$–$35.50$ $L{\cdot}s^{-1}{\cdot}ha^{-1}$), when storm overflows take place, i.e., when $p > 0.50$. The attention is drawn to the fact that the variability of the curve describing the probability of storm overflow by the effect of frontal rainfall, for which $t_r \leq 4.5$ h, shows in the range of $p = 0$–$0.55$ a similar variability to the curve obtained for rainfall data regardless of their genesis. For $p > 0.55$, a greater probability of storm overflowing due to all rainfall

events than precipitation lasting $t_r$ = 2.5–4.5 h was found. The curves determined in Figure 8b may indicate lower values of $i$ for rainfall episodes $t_r$ = 2.5–4.5 h than when $i > 4.86$ L·s$^{-1}$·ha$^{-1}$ which corresponds to all rainfall.

**5.5 Impact of rainfall genesis on the annual number of overflow events**

5    The simulation calculations performed with the use of the logit model described by eq. (3) showed that the annual number of overflows resulting from convective rainfall, frontal rainfall, and rainfall in convergence zone is lower in the case of using the simplified model taking into account only the $i$ value (Figure 9). These results are confirmed by the value of average retention in logit models, i.e., when $d_{av}$ = 3.80 mm (in the model that takes into account the genesis of rainfall) and $d_{av}$ = 4.32 mm (in the simplified model with lower predictive capabilities).

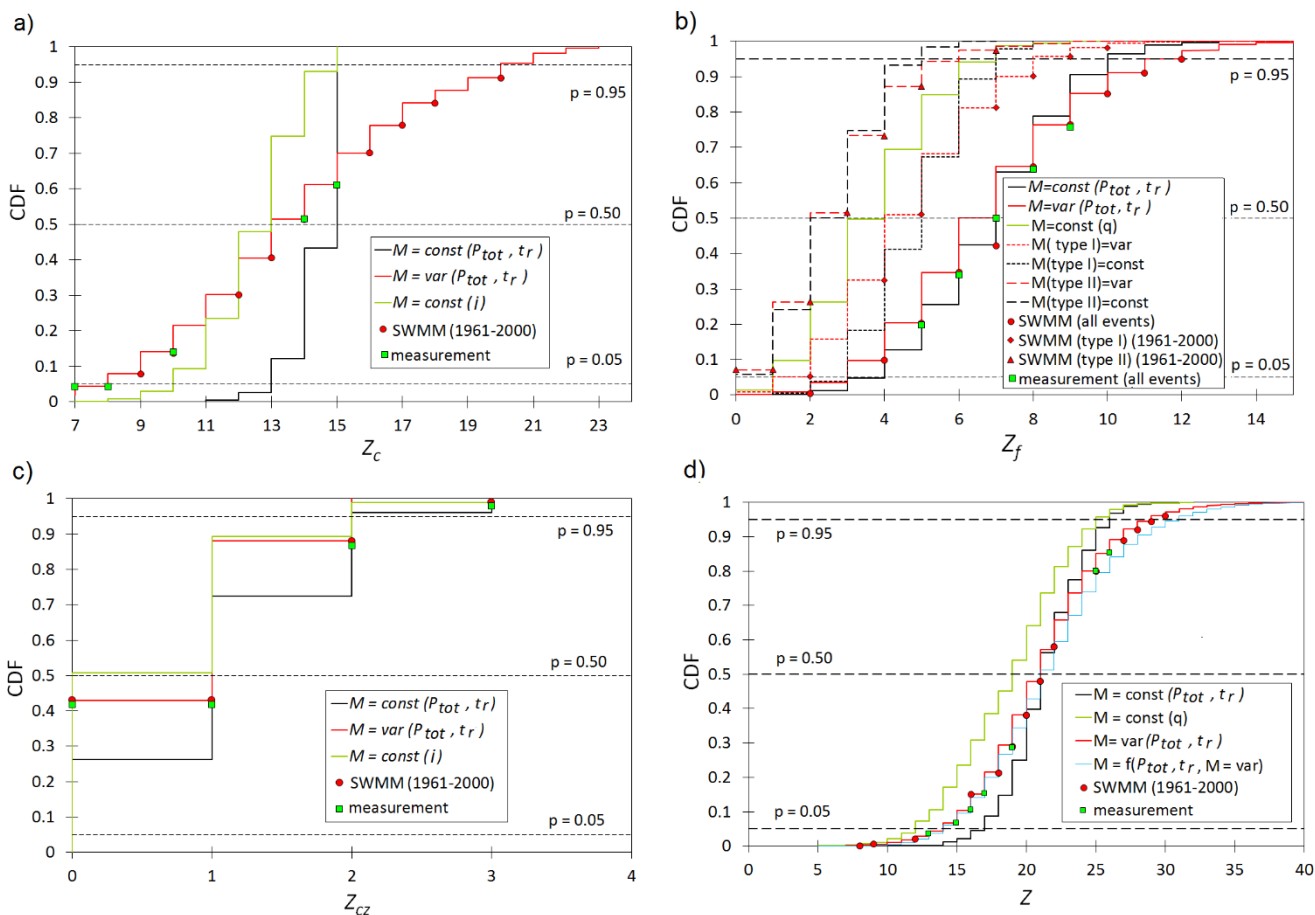

Figure 9: Distribution function (CDF) showing the annual number of storm overflow events (Z) caused by: (a) convective rainfall, (b) frontal rainfall, (c) rainfall in convergence zone; (d) Curve showing the probability of non-exceeding the annual number of overflows (Z). Explanations of the other symbols are in the Appendix.

Based on the calculations, it was found that the inclusion in the simulation of the annual number of storm overflows with a stochastic character of rain events has a significant impact on the results of the analyses. The analyses showed that in the case of precipitation caused by convection in air mass and frontal rainfall, the annual number of overflows (percentile value 0.50) is lower when the number of episodes is described by Poisson's distribution.

The calculations show that in the case of storm overflows resulting from convective precipitation, the stochastic nature of the precipitation has a significant impact on the values of the lower and upper percentiles (Figure 9a). For example, for the percentile value 0.05, the difference in the annual number of overflows obtained with the assumption of $M$ = var is 5 times greater than the solution when the average number of precipitation events per year was assumed. For the percentile value 0.95, the difference in the annual number of overflows between the considered solutions is 4. In the case of storm overflows resulting

from frontal rainfall (Figure 9b), the difference in the annual number of overflows obtained for the variants when $M$ = const and $M$ = var is much smaller than for convective rainfall (Figure 9a). This difference may be due to a significant variation in the number of convective and frontal rainfall events over an annual cycle and the variation in rainfall intensity in both cases. By analysing the results of the simulation, it can be concluded that when the number of precipitation episodes (convective, frontal, and in convergence zone) is determined by Poisson's distribution, the calculated annual number of overflows for $p <$

$0.50$ is smaller than when $M$ = const (Figure 9). On the other hand, for $p > 0.50$, the inverse relation is maintained. The influence of the theoretical distribution of the number of rainfall events per year on the values of percentiles 0.50–0.99 is confirmed by Szeląg et al. (2018). Based on the implemented simulations and designed curves, it can be established that the average annual number of overflows resulting from convective rainfall is 15 (Figure 9a), and in the case of frontal rainfall and rainfall in convergence zone it is much smaller and equals 7 and 1, assuming that $M$ = const (Figures 9b, 9c). In the case of storm

overflows caused by frontal overflow, it was found based on the determined curves that out of 7 overflows, as many as 5 are caused by rainfall connected with a cold front (type I), for which the duration of rainfall does not exceed $t_r$ = 4.5 h (Figure 9b). Very interesting results were obtained in the calculation variant, in which the number of rain events in a year was taken into account, regardless of their genesis. The average annual number of storm overflows for $p > 0.5$ is then greater than the number of overflows taking into account the rainfall genesis (Figure 9d). In conclusion, the omission of rainfall genesis in the

calculations may result in the overestimation of the average annual number of storm overflows. This overestimation affects the designed storm overflow height and the functioning of facilities located in the sewage network, e.g., stormwater treatment plants.

The innovative synthetic precipitation generator proposed in the paper enable the quantification of the impact of rainfall genesis on the annual number of storm overflows (Figure 9d), which until now, has not been included in the models developed by

other authors. This approach can be transferred to other facilities located in stormwater systems and used to assess the effectiveness of stormwater drainage systems. Ultimately, the results obtained with this approach may be the basis for the construction of an expert system for early warning about torrential phenomena caused by heavy rainfall. Another advantage of the developed probabilistic model is the possibility it provides to perform simulations of a long-term nature (multiannual),

and thus, to assess the impact of the distribution of individual precipitation types on the annual number of storm overflows and its variability.

The simulation calculations performed with the calibrated hydrodynamic model of the catchment area show that the results of the simulation of the annual number of storm overflow events (Figure 9d) caused by precipitation of different genesis are within the scope of the solution obtained with the probabilistic model. This finding confirms that the probabilistic model developed in the paper is an alternative solution to the hydrodynamic model. This fact is also confirmed by the annual number of overflow events caused by rainfall of different genesis obtained on the basis of measurements (Figure 9d). The conformity of the simulation results obtained with the probabilistic and hydrodynamic model may indicate that the values of the intercept determined in the relevant equations may correspond to the depth of the weighted average retention of the catchment area.

## 6 Conclusions

The calculations performed showed that the measurements of average rainfall intensity can be used to simulate (using a logit model) the storm overflow occurrence in a single rain episode. The simulation results obtained do not differ significantly from the calculations made based on the rainfall depth and duration. In both models, it was shown that the numerical value of the intercept in the model equations does not differ by more than 10 % from the depth of the weighted average retention of the catchment area. This fact has a significant practical meaning because it provides the possibility of using the results obtained in other urban catchments. However, in order to confirm this, further analyses on catchment areas with different physical and geographical characteristics are advisable.

The computational experiments carried out in the study allowed an assessment of the influence of the rainfall genesis (convective in air mass, frontal, and in the convergence zone) on the occurrence of storm overflow. Moreover, the ranges of the variability of the average rainfall intensity were determined, for which storm overflows were found. The information obtained may be used in engineering practice because on this basis, it is possible to determine whether a storm overflow will take place.

The identification of the operational state of the stormwater system (in this case, storm overflow) based on forecasting the weather front may be of practical significance. This identification of the operational state provides an opportunity to develop an early warning system against the occurrence of emergencies (spill of stormwater to the surface, hydraulic overload of pipes, overfilling of tanks) in stormwater systems.

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

# Appendix 1. List of symbols and abbreviations

| | |
|---|---|
| AUC | area under curve |
| CDF | cumulative distribution function of probability density |
| Chi | chi-square test |
| DC | diversion chamber |
| $d_{av}$ | weighted mean of the retention depth of the catchment area |
| $f(x)$ | probability density function |
| $F(x)_c$ | theoretical distribution to simulate rainfall characteristics due to convective rainfall |
| $F(x)_f$ | theoretical distribution to simulate rainfall characteristics due to frontal rainfall |
| $F(x)_{cz}$ | theoretical distribution to simulate rainfall characteristics due to rainfall in the convergence zone |
| $F(\zeta)_c$ | theoretical distribution to simulate the annual number of convective rainfall events |
| $F(\zeta)_f$ | theoretical distribution to simulate the annual number of frontal rainfall events |
| $F(\zeta)_{cz}$ | theoretical distribution to simulate the annual number of rainfall events in the convergence zone |
| FORM | first-order reliability model |
| $i$ | average rainfall intensity; 166.7 $P_{tot}/t_r$ [L·s$^{-1}$·ha$^{-1}$] |
| IC | Iman-Conover method |
| $K$ | number of samples modelled using the Monte Carlo method of annual rainfall series |
| KS | Kolmogorov-Smirnov test |
| LH | Latin-hypercube method |
| $M$ | annual number of rainfall events |
| $M = const(i)$ | calculation variant (annual number of overflow events), in which for simulation constant average annual number of rainfall events is used and to identify the overflow in a rainfall episode the following logit models were applied logit model $p = f(i)$ |
| $M = const(P_{tot}, t_r)$ | calculation variant (annual number of overflow events), in which for the simulation, the constant average annual number of rainfall events is used, and for identifying the overflow in a rainfall episode, the following logit models were applied: logit model $p = f(P_{tot}, t_r)$ |
| $M = var(i)$ | calculation variant (annual number of overflow events), in which the annual number of rainfall events caused by precipitation (convective, frontal, and in the convergence zone) is modelled, and for identification of overflow in a rainfall episode, the following logit models were applied: logit model $p = f(i)$ |
| $M = var(P_{tot}, t_r)$ | calculation variant (annual number of overflow events), in which the annual number of rainfall events caused by precipitation (convective, frontal, and in the convergence zone) is modelled, and for identification of overflow in a rainfall episode, the following logit models were applied: logit model $p = f(P_{tot}, t_r)$ |
| $M_c$ | annual number of convective rainfall events |
| $M_f$ | annual number of frontal rainfall events |
| $M_{cz}$ | annual number of rainfall events in the convergence zone |
| MC | Monte Carlo simulation |

| | |
|---|---|
| MES1, MES2 | flowmeter measures |
| MLE | maximum likelihood estimation |
| $N$ | number of samples in the Monte Carlo simulation |
| OV | storm overflow |
| $p$ | probability of a storm overflow event |
| $P_{tot}$ | total rainfall [mm] |
| $t_r$ | rainfall duration [min] |
| $R$ | Spearman's correlation coefficient |
| $R_z^2$ | counting error |
| SENS | sensitivity |
| SPEC | specificity |
| STP | stormwater treatment plant |
| SWMM | Stormwater Management Model |
| $Z$ | annual number of storm overflow events |
| $Z_c$ | annual number of storm overflow events due to convective rainfall |
| $Z_f$ | annual number of storm overflow events due to frontal rainfall |
| $Z_{cz}$ | annual number of storm overflow events due to rainfall in the convergence zone |
| $x_i$ | independent variables included in the logit model |
| $\alpha_i$ | values of estimated coefficients in the logit model |
| $\alpha, \beta, \gamma, \lambda, \mu, \sigma, \zeta$ | empirical coefficients estimated in statistical distributions |
| $(\mu_1(x_1),..., \mu_i(x_i))_s$ | mean value of variable $x_i$ in the data set obtained from simulation using the Iman-Conover method |
| $(\sigma_1(x_1),..., \sigma_i(x_i))_s$ | value of standard deviation of variable $x_i$ in the data set obtained from simulation using the Iman-Conover method |