# Peer review of "Application of logistic regression to simulate the influence of rainfall genesis on storm overflow operations: a probabilistic approach"

_Hydrology and Earth System Sciences, 2019_

## Referee Comment (RC1) · Anonymous Referee #1 · 26 Aug 2019

Comments on "Application of logistic regression to simulate the influence of rainfall genesis on storm overflow operation: a probabilistic approach" submitted by SzelÄĚg et al.

General comments: In their manuscript the authors present a comprehensive study on predicting combined sewer overflow (CSO) events through employing a probabilistic model. This model utilizes a generator of rainfall per year, a synthetic precipitation generator (yielding rainfall characteristics for each event), and a logistic regression (logit) model that predicts the number of CSO events, whereby two configurations are considered in the logit model: (i) rainfall height and duration as independent variables;

[Figure]

and (ii) average rainfall intensity as the only independent variable. In the model chain, a sub-division into three different types according to the synoptic situation is applied: convective, frontal, or convergence zone.

Predicting the number of CSO events utilizing stochastic approaches is an interesting approach worth to explore. However, it is hard to judge whether the approach presented by the authors is suitable to pursue this goal. I have to admit that I am not an expert in stochastic modelling which is why I focus on some more general aspects of the work. Even though the paper presents a lot of thorough analyses which are timely and well suited for the publication in Hydrology and Earth System Sciences, it lacks sufficient comprehensibility and an independent validation. Moreover, the authors claim – even though stating that further analyses are still required – that the approach is universal and transferrable. I have some doubts as to whether this statement is valid, given the results shown in the manuscript which are only supported by one single case study. I think it's OK to have only one case study but I would suggest rephrasing universal by a more conservative term.

However, I believe that there is a lot of potential to improve this article. Maybe some of my concerns might arise from my limited understanding of the topic. Even if my comments seem too critical, my intention is to improve the article.

My major concerns pertain to:

Wording: a lot of terms not common in the literature are used throughout the manuscript, e.g., receiver should be replaced by receiving waters. What do you mean by episode? Is it an event or a period or the event duration? Please be more specific! Movement of air could be replaced by advection. The term forecast is not appropriate in this context. I would replace this term by prediction. There is a lot of confusion regarding the usage of overflow and discharge. You do not predict discharge, only the number of overflow events.

Independent validation: Even though the logit model seems to perform very well, it

does not become clear how well the model works in my opinion. Is there any chance to add some comparisons with observations in the Figure 9 (at least averages as vertical lines)? Is it possible to perform a split sample test in order to analyze if the model chain provides reliable results for an independent period of time not involved into parameter estimation?

Transferability of the stochastic approach: as already mentioned, for me it remains unclear how the method could be transferred to other urban catchments. You argue that the catchment detention is physically meaningful. I agree in principle but I could imagine that a lot of other catchment characteristics might be relevant too. For instance, what is the impact of the network structure on the results? Even though the detention is tangible, its empirical deviation (Eq. 5 and 6) is rather empirical. Is there any chance to quantify this value with simplified hydrological calculations? This would support your argumentation regarding the transferability and the practical relevance. It might be worth to discuss the added value when compared to long-term simulation using hydrological (and hydrodynamic) models.

I would suggest revising the manuscript. Some things are too detailed (Section 3) or too general (e.g., the discussion on rainfall models that are not used here in Sect. 4.3; I would expect some methodology rather than an introduction to this topic). Sect. 5.2 should be sub-divided in order to increase readability.

Specific comments:

P2L5: National guidelines?

P3L18: I would suggest adding some more details on the catchment area (e.g., dry weather flow).

Figure 4: When I read the paper for the first time, my first idea was that you compare genesis of rainfall vs. not distinguishing the genesis as shown in Figure 4. In Figure 9, however, you mainly compare the average number of events vs. a modelled number of

events (including the comparison genesis vs. generalization of events?).

P5L9: The discussion on an "increase in the roughness of the substrate" is awkward in my opinion and not correct. Is this discussion really needed here? I would suggest rephrasing this section in a way that makes the explanations more concise, given the topic of the paper.

P17L4p: This statement remains unclear to me. Please be more precise! Why do you abandon this approach? I found this explanation confusing. I thought that M was modelled or even assumed to be constant? How is this related to your argumentation? P18L31: I don't think that there is any exact model. Please rephrase.

P19L10pp.: I do not understand how the numbers in the text are related to Figure 9. What means N (which has never been defined before)? Is it the simulation mentioned in P8L1p? Maybe it's worth to provide a table that summarizes the symbols used throughout the manuscript?

Technical comments:

P2L19: Do you mean Thornsal and Willams (2008)?

P3L2: Consider dropping knowledge.

P11L3: What is IC? Iman-Conover?

---

## Author Comment (AC1) · 9 Sep 2019

**Comment No. 1:**

The authors fully agree with the comments made. The text has been amended and supplemented as suggested. Precipitation episodes in the manuscript mean independent precipitation events, which have been separated from long-term rainfall series. As a result, the first sentence in Chapter 4.1 (P9L4) was modified as follows:

One of the basic conditions allowing for the completion of a synthetic precipitation generator is the separation of single independent rain events (episodes) in the ranks of rainfalls.

**Comment No. 2:**

Thank you very much for your comments. Chapter 4 has been modified. The order of its subchapters has been changed and in the new layout it is as follows:

4. Methodology
4.1. Simulation of the annual number of storm overflow events using a hydrodynamic model
4.2. Logistic regression
4.3. Separation of rain events and synthetic rainfall simulator
4.4. Annual rainfall series simulator

A new Subchapter 4.1 has been introduced in Chapter 4 entitled 'Simulation of the annual number of storm overflow events using a hydrodynamic model' (P9L3). The content of it is as follows:

One of the possible solutions allowing for the verification of empirical dependencies describing the operation of stormwater systems is the simulation of this operation with the use of a calibrated hydrodynamic model. It is an approach applied in engineering practice, which is confirmed by a number of works in this field (Bacchi et al. 2008; Doménech et al. 2010). Simulations performed with a hydrodynamic model on the basis of rainfall data allow to verify the prediction capabilities of probabilistic models designed to simulate the quantity and quality of stormwater and the operation of separate objects located on the stormwater system (tanks, overflows).
Within the framework of the conducted analyses, a calibrated model of the catchment basin made in the SWMM (Storm Water Management Model) program was used to verify the number of storm overflow events. The total area of the examined catchment is 62 ha, while the area of partial catchments is from 0.12 ha to 2.10 ha. The number of stormwater junction in the catchment area is 200 and the number of stormwater pipes located in this area is 72. The retention height of the imperviousness surfaces of the catchment area is 2.5 mm and the retention height of the pervious surfaces is 6.0 mm. The roughness coefficient of imperviousness areas is equal to 0.025 $[m^{-1/3} s]$ and of pervious areas 0.250 $[m^{-1/3} s]$. The roughness coefficient of the stormwater channels is equal to 0.018 $[m^{-1/3} s]$.
The results of simulation with hydrodynamic model were compared with the results of calculations made with the use of logit models for the data from the examined catchment from the period 2008-2016, which allowed to verify the determined relation $p = f(x_1, x_2, x_3, ..., x_i)$. Simultaneously, using the measurement data from the period 1961-2000, a simulation of the annual number of storm overflow events was performed, taking into account the precipitation genesis, which allowed for verification of the developed probabilistic model.

Subchapter 5.1 has also been modified (P12L12). Its title is now 'Logit model and its verification'. The following text is included in this subchapter below the logit models description therein (P12L13-P13L11):

The values of the SENS (sensitivity) and SPEC (specificity) coefficients are usually calculated to assess the predictive capability of the models. However, SENS = f(SPEC) may also be used for this purpose. In this case, the higher the area value (the maximum value is AUC = 1) between the SENS = SPEC and SENS = f(SPEC) curves, the more accurate the model will be (Fig. A1).

[Figure]

Fig. A1. Relation SENS = f(SPEC) for the logit curves p = f(q) and p = f($P_{tot}$, $t_r$).

To verify the predictive capabilities of the logit models and their dependencies, a calibrated hydrodynamic model was used. The results of the simulation obtained with the hydrodynamic model for data from the period 2008-2016 were compared with the results of calculations made with the use of logit models p = f(q) and p = f(($P_{tot}$, $t_r$)) (see Table A1).

Table A1: Comparison of measurements of the annual number of overflow events with the calculations results obtained with hydrodynamic and logit models

| Year | Z | $Z_{SWMM}$ | $Z_{logit(P,t)}$ | $Z_{logit(q)}$ |
|------|-----|------|--------|--------|
| 2008 | 13 | 15 | 14 | 12 |
| 2009 | 15 | 16 | 16 | 14 |
| 2010 | 17 | 18 | 19 | 16 |
| 2011 | 19 | 20 | 19 | 17 |
| 2012 | 13[*] | 21 | 20/14[*] | 19/11[*] |
| 2013 | 13[*] | 22 | 20/13[*] | 20/11[*] |
| 2014 | 16[*] | 29 | 28/16[*] | 27/14[*] |
| 2015 | 25 | 26 | 26 | 23 |
| 2016 | 26 | 22 | 21 | 24 |
| 2017 | 16 | 17 | 17 | 14 |

The markings in Tab. A1 mean: Z – the measured annual number of storm overflow events in the analyzed urban catchment area, SWMM – the modeled number of storm overflow events by means of a calibrated hydrodynamic model of the catchment area, $Z_{logit(P,t)}$ – the modeled

number of storm overflow events by means of the relation $p = f(P_{tot}, t_r)$, $Z_{logit(q)}$ − modeled number of storm overflow events using the relation $p = f(q)$, * − storm overflow events in the period (2012-2014) determined on the basis of the maximum filling of the distribution chamber (DC).

Referring to comment No. 2, the text in Subchapter 5.1 (from P13L12 to P13L23) has also been modified:

The above relations (3), (4) are of a local character and can be applied in principle only in relation to the analysed catchment area. With this in mind, an attempt was made to build a universal model by modifying these relations. Based on the theoretical considerations of Thorndahl and Willems (2008), who investigated the relationships between the characteristics of catchment areas, including the limit retention that determines the volume of overflow, the following equations were determined:

$$0.566\,P_{tot} - 0.004\,t_r - 2.152 = 0\ /\cdot \frac{1}{0.566} = P_{tot} - 0.007\,t_r - 3.802 = 0 \tag{a1}$$

$$P_{tot} - 0.007t_r - 3.802 = P_{tot} - 0.007t_r - d_{av} = 0 \tag{a2}$$

$$0.312\,q - 1.257 = 0\ /\cdot \frac{1}{0.312} = q - 4.03 = 0 \tag{a3}$$

$$q - 4.03 = q - d_{av} = 0 \tag{a4}$$

Analysing the values of coefficients in equations (a2) and (a4), one can see that the values of the free word in these equations are close to the value of the weighted mean of the retention height of the catchment area ($d_{av}$). The difference between the weighted mean of the catchment retention and the values of the free words in the equations does not exceed 5 %. Simulation calculations of the annual number of overflow events in the tested catchment area confirm the above statement (Table A1). At the same time, this fact is supported by the results obtained in Fig. A2, which indicate that the results of the simulation of the annual number of overflow events using the hydrodynamic model are within the scope of a probabilistic solution. However, due to the fact that the presented analyses were performed only for a single catchment, it is necessary to verify the obtained results using examples of other catchments with different physical and geographical characteristics.

**Comment No. 3:**

Thank you very much for your comments. The contents of Chapters 3, 4 and 5 have been modified. We consider Chapter 3 to be an important part of our work in terms of the goals we have set. The article discusses the problem of storm overflows in the urban catchment area, caused by rainfall of different origins. One of the elements of the proposed probabilistic model for modeling the number of storm overflow events is a generator of synthetic rainfall. In order to simulate this model, rainfall events with different genesis had to be taken into account. The conducted analyses allowed to determine the influence of the distribution of the average annual number of precipitation episodes with different genesis of precipitation on the variability of the number of storm overflow events. Therefore, in our opinion, it is important to familiarize the reader with the main processes and mechanisms occurring in the troposphere over central Europe, which are reflected in high precipitation. The presentation of this problem is presented in the introduction to Chapter 3 of the manuscript. The main characteristics of convective precipitation, frontal precipitation and convergence zones presented in this chapter are the result of the exceptional variability of meteorological conditions in this part of the European continent and are therefore presented in a broader context. However, we agree with the Reviewer's

comment that the considerations on this subject presented in Chapter 3 may be shortened and partly moved to the introductory chapter of the article.

Therefore, the following changes have been made to the manuscript:

– the title of Chapter 3 (P4L7) has been changed as follows: 'Rainfall data and analysis'
– the previous content presented in the text from P4L8 to P5L17 was modified in the following way and moved to Chapter 1 (after P3L7):

Rainfall is universally classified into three types (Sumner, 1988): convective, cyclonic and orographic. The main distinguishing feature between convective precipitation in air mass and frontal precipitation in mid-latitudes is its spatial extent and duration. The range of convective precipitation associated with local air circulation is much smaller than in the case of travelling extratropical cyclones with weather fronts. Convective precipitation induced by single thunderstorm cells, their complexes or squall lines is short-lived, but is characterized by high average intensity (Kane et al., 1987) and causes flash floods in many areas (Gaume et al., 2009; Marchi et al., 2010; Bryndal, 2015). On the other hand, the lifespan of the mechanisms of creating cyclonic precipitation is much longer than that of convective precipitation – in the order of days rather than hours. Hence, the effect of this is long-term rainfall with a high depth (Frame et al., 2017), often causing regional floods (Barredo, 2007). The presented classification of precipitation types distinguished by Sumner (1988) due to the origin of the phenomenon, developed for the British Isles and Western Europe, cannot be directly applied in practical hydrology in other regions of the continent, especially in its eastern and central parts. This is the result of exceptional variability of meteorological conditions occurring in the temperate zone of warm transition climate – on the borderline of air masses coming from the Atlantic and continental masses from the east (Twardosz and Niedźwiedź, 2001; Niedźwiedź et al., 2009; Twardosz et al., 2011; Łupikasza, 2016). Analysis of maximum rainfall of different duration in Poland carried out at the end of the 1990s (Kupczyk and Suligowski, 1997, 2011), supplemented by the analysis of synoptic situation (on the base of surface synoptic charts of Europe, published in Daily Meteorological Bulletin of the Institute of Meteorology and Water Management – IMGW in Warsaw) and a calendar describing the types of atmospheric circulation together with air masses and air fronts (Niedźwiedź, 2019), led to the separation of three types of genetic precipitation: convective in air mass, frontal and convergence zones.

– also Figure 2 (P6L1) has been removed from Chapter 3.

The main changes in Chapter 4 have been discussed earlier. In addition, the following introduction text from the existing Chapter 4.3 ('Synthetic precipitation generator'), i.e. contained from P10L17 to P10L23, has been moved to the introductory chapter (P2L22):

Multidimensional scaling and fractal geometry methods are used to simulate rainfall series (Rupp et al., 2009; Licznar et al., 2015; Müller-Thomy and Haberlandt 2015). Alternative approaches are solutions based on multidimensional distributions created on the basis of theoretical distributions and copula functions (Vandenberghe et al., 2010; Vernieuwe et al., 2015). However, these solutions are relatively complex and require expert knowledge. To predict the annual number of storm overflow events, hydrodynamic models are usually used, whereas less frequent are empirical models (Szeląg et al. 2018). However, this approach is very local and in many cases requires the construction of a catchment model.

In view of the above changes, the text after P9L19 has also been modified as follows:

Taking into account the calculation algorithm described in Chapter 4 (Methodology) and developed on the basis of theoretical distributions of rainfall characteristics describing the operation of a storm overflow, a model for simulating synthetic rainfall series was used for further analyses. The model was determined using the Monte Carlo method modified by Iman-Conover (1982). This model gives the possibility of simulation of independent variables taken into account in the research on the basis of theoretical distributions determined for them. In the Monte Carlo method used the variability of the considered variables is described by boundary (theoretical) distributions, and the basis for evaluation of their correlation is the Spearman correlation coefficient. The conditions, which must be met in order for the results obtained to be considered correct, are as follows:

– in the data obtained from simulation and measurements, the mean values ($\mu_1$, $\mu_2$,...,$\mu_i$)$_s$ and the standard deviations ($\sigma_1$, $\sigma_2$,...,$\sigma_i$)$_s$ of the variables ($x_i$) considered in j samples do not differ by more than 5%,

– theoretical distributions of $x_i$ variables obtained from simulation are consistent with those obtained from measurements; in order to meet this condition it is recommended to use the Kolmogorov-Smirnov test,

– the value of the correlation coefficient (R) between individual dependent variables ($x_i$) obtained for data from MC simulation does not differ by more than 5% from the value of R obtained for empirical data.

If the above mentioned conditions are met, the results of the simulation performed with the IC method can be considered correct. If this is not the case, the sample size of the MC needs to be increased (Wu and Tsang, 2004). In order to limit the sample size and improve the efficiency of the Iman-Conover algorithm, a modification has been developed by using the Latin-Hybercube algorithm, which is part of the layered sampling methods aimed at improving the "uniformity" of the numbers generated from the boundary distributions.

On the basis of the determined boundary distributions of rainfall characteristics, simulations of synthetic rainfall series were performed with the use of the Monte Carlo method with Iman-Conover modifications and taking into account the Latin-Hypercube algorithm.

With reference to Subchapter 5.2, the text of it was divided into the following three Subchapters:

**5.2. Identification of empirical distributions and theoretical rainfall characteristics**

Subchapter 5.2 includes analyses related to the determination of statistical distributions of the following variables: rainfall depth ($P_{tot}$), rainfall duration ($t_r$), average rainfall intensity (q), and number of rain events in a year of varied genesis.
Chapter 5.2 in the manuscript contains the text from P13L24 to P17L5.

**5.3. Impact of rainfall genesis on the probability of storm overflow occurring**

Subchapter 5.3 presents the determined relationship between the genesis of rainfall and the probability of overflow event, as well as the ranges of variation of average rainfall, taking into account the rainfall genesis, which determines the occurrence of storm overflow event.
Subchapter 5.3 in the manuscript contains the text from P17L6 to P18L27.

**5.4. Impact of precipitation genesis on the annual number of overflow events**

Subchapter 5.4 presents the annual number of overflow events caused by rainfall (convective, frontal, convergence zones). At the same time, a comparative analysis of the annual number of overflow events obtained with a simplified logit model (based on the average rainfall intensity) and the number of overflow events obtained with an accurate model (based on the total amount of rainfall and its duration) is carried out in this chapter.

Subchapter 5.4 in the manuscript contains the text from P18L28 to P20L16 in the manuscript.

The text in P18L30 has been modified as follows:

Simulation calculations performed with the calibrated hydrodynamic model of the catchment area show that the results of the simulation of the annual number of storm overflow events (Fig. A2) caused by precipitation of different genesis are within the scope of the solution obtained with the probabilistic model. This confirms that the probabilistic model developed in the paper is an alternative solution to the hydrodynamic model. The above fact is also confirmed by the annual number of overflow events caused by rainfall of different genesis obtained on the basis of measurements. (rys. A2). The conformity of simulation results obtained with the probabilistic and hydrodynamic model may indicate that the values of free words determined in the relevant equations may correspond to the weighted average retention of the catchment area.

[Figure]

Fig. A2. (a) Distribution function (CDF) showing the annual number of storm overflow events caused by convective rainfall ($Z_c$); (b) Distribution function showing the annual number of overflow events caused by frontal rainfall ($Z_f$); (c) Distribution function showing the annual number of overflow events caused by rainfall in convergence zones ($Z_{cz}$); (d) Curve showing the probability of exceeding the annual number of overflow events ($Z$); M − number of rain events in a year: M = const($x_i$) − in the probabilistic model the average number of rain events from the period 1961-2000 was adopted in the calculations; M = var($x_i$) − in the probabilistic

model theoretical distributions describing the number of rain events in a year were adopted (Table A1).

**Specific comments:**
**P2L5:**

Indeed, the quoted guidelines are foreign. The paper also takes into account national Polish guidelines in the form of the Regulation of the Minister of the Environment of 18 November 2014 on the conditions to be met when discharging sewage into water or soil and on the substances particularly harmful to the aquatic environment.

**P3L18:**

The characteristics of the catchment area have been extended. In item P3L24 of the manuscript the following additional information has been added:

Analysis of the measurement data (2008-2016) concerning the analysed catchment area showed that the antecedent period there lasted 0.16–60 days and storms occurred 27–47 times a year. The average annual air temperature during the period under consideration varied from 8.1 to 9.6°C. At the same time, the analysis of measurement data of flows recorded with the MES1 flow meter showed that in the antecedent periods the temporary stream of stormwater was from 0.001 to 0.009 $m^3s^{-1}$, which indicates the occurrence of infiltration in the stormwater system under study.

**Figure 4:**
Taking as a starting point the diagram of the algorithm in Fig. 4 (P8L8), two approaches to the simulation of the annual number of overflow events were analysed. In the first one the genesis of rainfall $M = const(P_{tot}, t_r)$ or $M = const(q)$ (Fig. A2d) was omitted. In the second approach the developed rainfall generator (Fig. 5), described in Subchapter 4.4 ('Simulator of synthetic annual rainfall series'), was used and the results obtained are shown in Fig. A2d in the form of $M = var(P_{tot}, t_r)$ curve. The additional dependencies showed in Fig. A2a, A2b, A2c are a detailed description of the results of the simulation analyses and illustrate the annual number of overflow events caused by rainfall of various origins (convective, frontal, convergence zones). Indeed, in the original version of the manuscript, this may lead to confusion. The corrected manuscript therefore contains the following table of symbols at the end of the article (Table A2):

| L.p. | Symbol | Meaning | |
|---|---|---|---|
| 1 | N | number of samples in the Monte Carlo simulation | |
| 2 | p | probability of a storm overflow event | |
| 3 | $P_{tot}$ | rainfall depth | |
| 4 | $t_r$ | rainfall duration | |
| 5 | q | average rainfall rate (q =166.7 P/t) | |
| 6 | f(x) | probability density function of the variable x | |
| 7 | CDF | cumulative distribution function of probability density | |
| 8 | M =const(q) | calculation variant (annual number of overflow events), in which for | logit model p =f(q) |
| 9 | M =const($P_{tot}$,$t_r$) | simulation constant average annual number of rainfall events is used and to | logit model p =f($P_{tot}$, $t_r$) |

| | | identify the overflow in a rainfall episode the following logit models were applied: | |
|---|---|---|---|
| 10 | $M = \mathrm{var}(q)$ | calculation variant (annual number of overflow events), in which the annual number of rainfall events caused by precipitation (convective, frontal, convergence zones) is modelled, and for identification of overflow in a rainfall episode the following logit models were applied: | logit model $p = f(q)$ |
| 11 | $M = \mathrm{var}(P_{tot}, t_r)$ | | logit model $p = f(P_{tot}, t_r)$ |
| 12 | $Z$ | annual number of storm overflow events | |
| 13 | $Z_c$ | annual number of storm overflow events due to rainfall: | convective |
| 14 | $Z_f$ | | frontal |
| 15 | $Z_{cz}$ | | convergence zones |
| 16 | $M_c$ | annual rainfall events number caused by rainfall: | convective |
| 17 | $M_f$ | | frontal |
| 18 | $M_{cz}$ | | convergence zones |
| 19 | $F(\zeta)_c$ | theoretical distribution to simulate the annual number of rainfall events: | convective |
| 20 | $F(\zeta)_f$ | | frontal |
| 21 | $F(\zeta)_{cz}$ | | convergence zones |
| 22 | $F(x)_c$ | theoretical distribution to simulate rainfall characteristics due to rainfall: | convective |
| 23 | $F(x)_f$ | | frontal |
| 24 | $F(x)_{cz}$ | | convergence zones |

**P5L9:**

Thank you for the valuable comment and signalling that one of the reasons given for the variability in the frequency, intensity and duration of rainfall associated with atmospheric fronts in Central Europe was unclear and awkward. The discussion on this issue was omitted, as mentioned above (in response to Comment No. 3). The existing content presented from P4L8 to P5L17 was modified and moved to Chapter 1 (after P3L7).

**P17L4:**

The text of the manuscript in P17L3 has been modified as follows:

The aim of the simulation calculations was to compare the results of calculations of the annual number of overflow events by means of a model taking into account the genesis of rainfall and a simplified model, in which only a differentiated number of rainfall events was included with the omission of the rainfall genesis. Reference data constituting the basis for the comparison was the number of storm overflow events obtained taking into account the rainfall genesis – M = var($P_{tot}, t_r$) (Fig. A2d). Due to the fact that the determined statistical distribution describing

the annual number of rainfall events (Fig. 6a) indicated overestimation of the minimum values and underestimation of the maximum values, the obtained results of the simulation of the annual number of overflow events on the basis of such distribution would differ from the values obtained on the basis of the empirical distribution. Thus, the comparison of the simulation results between the simplified model and the model taking into account the rainfall genesis would be burdened with an error, which would have a negative impact on the evaluation and interpretation of the calculation results. Obtained results of the simulation with a simplified model and one taking into account rainfall genesis can of course be compared, but it would be difficult to draw conclusions from this comparison to determine how the simplification introduced in the simulation of the annual number of overflow events (omitting rainfall genesis) affects the results of the simulation.

In the corrected manuscript, items P17L4 and P17L5 were removed.

**P18L31:**

Thank you for your comments. The term 'exact model' has been replaced by the term 'model that takes into account the genesis of rainfall'.

**P2L19:**

The order of names has been changed: 'Willems and Thorndahl (2008) is now replaced by ''Thorndahl and Willems (2008)'.

**P11L3:**

Indeed, IC is the abbreviation for the Iman-Conover method.

**P3L2:**

The word 'knowledge' has been removed from the text.

---

## Referee Comment (RC2) · Anonymous Referee #2 · 25 Sep 2019

**Review of the Paper Szelag et al. (2019), Hess 2019-271 Hydrology and Earth Sytem Sciences**

**Common comments**

Please consider, the reviewer is not a native speaker. Therefore all editorial remarks and recommendations should be checked. It cannot be excluded, that some editorial recommendations are not correct.

Nevertheless the reviewer has the impression, that some improvements are possible ore even necessary. Has the paper been checked by a native speaker?

Some formulations could be more comprehensible. Especially some long sentences are difficult to read. Often it is better to split such sentences into two sentences.

**Variables and symbols**

The authors have used some variables. It would be useful to add a table with the most important variables, their meaning and unit.

In line 9-11 $P_{tot}$ and $t_r$ are mentioned, but defined later in line 9-29. There it would be better to write $q = P_{tot} / t_r = 166.7 . . .$

In table 1 and figure 8 the terms "frontal, type I" and frontal, type II" are used, but they are defined first in line 18-2. Why do the authors not use the terms "cold front" and "warm front"?

The reviewer knows q as specific discharge ore runoff rate, but not as rain intensity. In English parers for rain intensity stands often I or i (sometimes PI for precipitation intensity).

In table 1 the symbol M is used, but nowhere declared.

In line 7-25 CDF's "describe the probability of exceeding the flow number of storm overflow discharges." But in the paper CDF is used generally, as customary. On the contrary in picture 8b) CDF is the distribution of the rain intensity. Beside "exceeding" seems to be not correct, since CDF's represent the probability of undershooting!

**Tables and Figures**

Table 1: The sequential arrangement of the variables is not perfect. The order could be all $P_{tot}$, $t_r$ q, and M ore all annual values, convective frontal . . .

Figures: Partly the units and symbols are missed. The caption of figures should be understandable and clear enough without reading the text.

Figure 6 and 7: It would be favourable, if the both axes would have the same range. Not every reader is experienced in such analysis. One sentence or two sentences would be useful to explain, what the background of such pictures is. Instead of "Observed Value" (y-axis) it is recommended to write "Empirical Quantile". The caption could be: "Comparison of empirical and theoretical quantiles concerning the number of rainfall episodes and distinguishing rainfall types"

Figure 6: Is it right, that the sum of the highest values of b), c) and d) = 55 should be equal to the highest value of a)?

Figure 8 (a): p is the probability of overflow discharge. But for wich variable stands CDF here? The reviewer has not found any remarks. Therefore he don't understand in line 18-5 why "the percentiel value p = 0,50 is as high as 0,90"? It looks like that the CDF-value 0,50 is as high as 0,90? But what means CDF here?

**Comments concerning the content**

Line 1-15: The Model is innovative, formulate more clear, what the reasons are.

Line 1-24 and 7-21: The generator should be mentioned as first element.

Line 2-14: write "such discharges". Mostly the words "overflow discharge" are used, but in some further cases only the word "discharge" (for example 8-4 and 18-29), while "overflow discharge" is meant. Please check such cases.

Line 5-22: write better "It concerns events with high intensity and short duration."

Lines 7-7 to 7-11: this paragraph seems to be a repeat.

Lines 8-9 to 9-2: Both sentences sound similar.

Line 9-5: What is meant by "in the ranks of"

Lines 9-9 to 9-18: this paragraph seems to be a repeat.

Line 9-24: "simulate objects" sounds strange, write better "simulate the influence of constructions on flow procsses"

Line 10-14: The investigation period is 1961 to 2000 (page 5). Here the years 2012-2014 are discussed?

Line 10-14: Whant is meant by "period separating subsequent rainfall events"?

Line 11-15 to 11-24: This part concerns not methods. Similar discussions are in the first parts of the paper.

Lines 12-8 to 12-11: This sentences are nearly a repeat of pages 7/8, but the steps are not denominated identical. Here 4 steps are listed, but the chapter consists of the two parts 5.1 and 5.2 only.

Line 13-19: The reviewer don't know what "values of free words" are? Possibly other readers will have the same problem.

**Editorial notes**

Line 1-16: The text within the brackets should be formulated as sub-clause ore as an additional sentence.

Line 1-21: write "great" instead of "big"

Line 1-29: two times determine

Line 1-31: the results are suited for implementation

Line 2-2: three times the word "of" in series

Line 2-29: **what** was not taken into account when rainfall generators **were used** to simulate

Line 2-32: "concern simulations" sounds strange, write perhaps better "consider"

Line 3-1: "course of precipitation phenomena" sounds strange

Line 3-4: "forecasting the operation" sounds strange, write "basis for the control of systems"

Line 3-11: Sometime it is written "model of the rainfall generator". The generator is a model, the word "model" seems to be unnessecary.

Line 3-22: a space is missed

Line 3-23: better "height difference", cancel "of ordinates"

Line 4-10: "generated" better as "shaped"

4-11: write "A third", since before only two mechanism are announced

4-11: write "which include both above mentioned components"

4-16: write at the end "are" instead of "is"

5-2 and 5-13: line break (new paragraph)

5-17: "convergence zone" is not a type of precipitation, write better "generated in convergence zones"

5-19: write "only these data were"

5-33: "variable" instead of "varied"

6-12: "precipitation emitted" sounds strange

7-25: overflow discharges per year

10-10: Write "The second variant is a simplification. It considers only a single"

10-18: set methods before the brackets

10-30: write better "should be consistent"

13-16: write "are valid" instead of "take place"

13-19: write "of this relationship" and cancel the "(eq. 5 and eq. 6)"

15-8: expressed better by

18-3: Write "distinguished" instead of "made".

18-6: line break (new paragraph)

**Following words seems to be unnessecary**

Line 1-20: was demonstrated

Line 1-31/32: knowledge concerning

2-10: collecting

2-19: in the work

Line 2-22: of simulation

Line 2-27: in its modeling

3-26: the work

4-6: article

4-17: in many areas (this mechanism is independent of the area)

5-3: of the phenomenon

9-6: in the paper

10-10: in the analysis performed

11-13: in the rainfall episode

11-14: better "seasonal differences"

13-1: using the model

---

## Author Comment (AC2) · 4 Oct 2019

**Reviewer #2**

We thank the reviewer for the effort and the time spend on this manuscript.
Thank you very much for putting forward very concrete proposals for improving it. The article after the correction was checked by a specialist in hydrology and mathematical modelling in urban drainage basins.

Below we present detailed answers to the following comments.

**Comment 1**
**The authors have used some variables. It would be useful to add a table with the most important variables, their meaning and unit.**

At the end of the paper there is a table with a list of all used symbols, parameters and abbreviations.

| N.m. | Symbol | Meaning | |
|------|--------|---------|---|
| 1 | $AUC$ | area under curve | |
| 2 | $CDF$ | cumulative distribution function of probability density | |
| 3 | $d_{av}$ | weighted mean of the retention depth of the catchment area | |
| 4 | $f(x)$ | probability density function | |
| 5 | $F(x)_c$ | theoretical distribution to simulate rainfall characteristics due to rainfall: | convective |
| 6 | $F(x)_f$ | | frontal |
| 7 | $F(x)_{cz}$ | | in convergence zones |
| 8 | $F(\zeta)_c$ | theoretical distribution to simulate the annual number of rainfall events: | convective |
| 9 | $F(\zeta)_f$ | | frontal |
| 10 | $F(\zeta)_{cz}$ | | in convergence zones |
| 11 | $i$ | average rainfall intensity [$L \cdot s^{-1} \, ha^{-1}$] | |
| 12 | $IC$ | Iman-Conover method | |
| 13 | $M$ | annual number of rainfall events | |
| 14 | $M = const(i)$ | calculation variant (annual number of overflow events), in which for simulation constant average annual number of rainfall events is used and to identify the overflow in a rainfall episode the following logit models were applied: | logit model $p = f(i)$ |
| 15 | $M = const(P_{tot}, t_r)$ | | logit model $p = f(P_{tot}, t_r)$ |
| 16 | $M = var(i)$ | calculation variant (annual number of overflow events), in which the annual number of rainfall events caused by precipitation (convective, frontal, in convergence zones) is modelled, and for identification of overflow in a rainfall episode the following logit models were applied: | logit model $p = f(i)$ |
| 17 | $M = var(P_{tot}, t_r)$ | | logit model $p = f(P_{tot}, t_r)$ |
| 18 | $M_c$ | annual number of rainfall events caused by rainfall: | convective |
| 19 | $M_f$ | | frontal |
| 20 | $M_{cz}$ | | in convergence zones |
| 21 | $MC$ | Monte Carlo method | |
| 22 | $N$ | number of samples in the Monte Carlo simulation | |
| 23 | $p$ | probability of a storm overflow event | |
| 24 | $P_{tot}$ | total rainfall [mm] | |

| 25 | $t_r$ | rainfall duration [min] | | |
|---|---|---|---|---|
| 26 | $R$ | Spearman's correlation coefficient | | |
| 27 | $R_z^2$ | counting error | | |
| 28 | $SENS$ | sensitivity | | |
| 29 | $SPEC$ | specificity | | |
| 30 | $SWMM$ | *Storm Water Management Model* | | |
| 31 | $Z$ | annual number of storm overflow events | | |
| 32 | $Z_c$ | annual number of storm overflow events due to rainfall: | | convective |
| 33 | $Z_f$ | | | frontal |
| 34 | $Z_{cz}$ | | | in convergence zones |
| 35 | $x_i$ | independent variables included in the logit model | | |
| 36 | $\alpha_i$ | values of estimated coefficients in the logit model | | |
| 37 | $\alpha, \beta, \sigma, \lambda, \mu, \gamma, \zeta$ | empirical coefficients estimated in statistical distributions | | |
| 38 | $(\mu_1(x_1), \mu_2(x_2),...,\mu_i(x_i))_s$ | mean value of variable $x_i$ in the data set obtained from simulation using the Iman-Conover method | | |
| 39 | $(\sigma_1(x_1), \sigma_2(x_2),...,\sigma_i(x_i))_s$ | value of standard deviation of variable $x_i$ in the data set obtained from simulation using the Iman-Conover method | | |

The following sentence (P10L28):

,,in the data obtained from simulation and measurements, the mean values $(\mu_1(x_1), \mu_2(x_2),...,\mu_i(x_i))_s$ and the standard deviations $(\sigma_1(x_1), \sigma_2(x_2),...,\sigma_i(x_i))_s$ of the variables $(x_i)$ considered in j samples do not differ by more than 5 %"

has been modified as follows:

,,in the data obtained from simulation and measurements, the mean values $(\mu_1(x_1), \mu_2(x_2),...,\mu_i(x_i))_s$ and the standard deviations $(\sigma_1(x_1), \sigma_2(x_2),...,\sigma_i(x_i))_s$ of the variables $(x_i)$ considered in j samples do not differ by more than 5 %."

**Comment 2**

**In line 9-11 Ptot and tr are mentioned, but defined later in line 9-29. There it would be better to write q = Ptot / tr = 166.7 . . .**

**The reviewer knows q as specific discharge ore runoff rate, but not as rain intensity. In English parers for rain intensity stands often I or i (sometimes PI for precipitation intensity).**

The following text (P9L11):

,,In order to obtain the best possible matching of theoretical data (precipitation characteristics including $P_{tot}$ and $t_r$ values for precipitation of appropriate genesis) with empirical data, the following statistical distributions were considered ..."

has been modified as follows:

,,In order to obtain the best possible fit of theoretical data to empirical precipitation data (including: total rainfall – $P_{tot}$, rainfall duration – $t_r$ and average rainfall intensity – $i$, for precipitation of appropriate genesis), the following statistical distributions were considered ..."

The designations in Figures 8 and 9 below have thus been changed:

[Figure]

Fig. 8: Impact of rainfall genesis on: (a) the probability of storm overflow, (b) rainfall intensity distribution determining overflow.

[Figure]

Fig. 9: (a) Distribution function (CDF) showing the annual number of overflows due to convective rainfall; (b) Distribution function (CDF) showing the annual number of overflows due to frontal rainfall; (c) Distribution function (CDF) showing the annual number of overflows due to rainfall in convergence zones; (d) Curve showing the probability of non exceeding the annual number of overflows.

In view of the modification of the designations in Figures 8 and 9, further following corrections have been made:
- P13L18; text „$q = 4.030 \approx d_{av}$" has been modified as follows: „$i = 4.030 \approx d_{av}$"
- P13L27; text „($P_{tot}$, $t_r$, q, M)" has been modified as follows: „($P_{tot}$, $t_r$, $i$, M)"
- P18L6; text „and the value q" has been modified as follows: „and the value $i$"
- P18L18; text „$q = 4.49–35.50 \ L \ s^{-1} \ ha^{-1}$" has been modified as follows: „$i = 4.49–35.50 \ L \ s^{-1} \ ha^{-1}$"
- P18L22; text „it corresponds to $q = 4.49–35.50 \ L \ s^{-1} \ ha^{-1}$" has been modified as follows: „it corresponds to $i = 4.49–35.50 \ L \ s^{-1} \ ha^{-1}$"
- P18L27; tekst „when $q > 4.86 \ L \ s^{-1} \ ha^{-1}$" has been modified as follows: „when $i > 4.86 \ L \ s^{-1} \ ha^{-1}$"

**Comment 3**
**In table 1 and figure 8 the terms "frontal, type I" and frontal, type II" are used, but they are defined first in line 18-2. Why do the authors not use the terms "cold front" and "warm front"? In table 1 the symbol M is used, but nowhere declared.**

Table 1 (P14L3) is supplemented by additional explanatory notes: frontal type I – cold front, frontal type II – warm front (see answer to Comment 5).
In Answer 1 there are descriptions of symbols included in the work.

**Comment 4**
**In line 7-25 CDF's "describe the probability of exceeding the flow number of storm overflow discharges." But in the paper CDF is used generally, as customary. On the contrary in picture 8b) CDF is the distribution of the rain intensity. Beside "exceeding" seems to be not correct, since CDF's represent the probability of undershooting!**

Of course, we agree with the reviewer. We have corrected the use of the CDF term in the text.

**Comment 5**
**Table 1: The sequential arrangement of the variables is not perfect. The order could be all $P_{tot}$, $t_r$, q, and M ore all annual values, convective frontal . ..**

Table 1 has been modified (in accordance with the remark above) to the following form:

| Variable | Distribution | Model parameters | p (KS) | p (Chi) |
|---|---|---|---|---|
| $P_{tot}$ (all events) | Weibull | $\beta = 0.772; \gamma = 5.158; \mu = 3.00$ | 0.121 | 0.096 |
| $t_r$ (all events) | GEV | $\zeta = 0.466; \sigma = 129.355; \mu = 108$ | 0.096 | 0.071 |
| $i$ (all events) | log-normal | $\sigma = 1.932; \mu = 0.855$ | 0.112 | 0.096 |
| $M$ (all events) | Poisson | $\lambda = 32.80$ | 0.624 | 0.053 |
| $P_{tot}$ (convective) | Weibull | $\beta = 0.821; \gamma = 3.102; \mu = 3.00$ | 0.477 | 0.412 |
| $t_r$ (convective) | beta | $\alpha = 1.391; \beta = 1.173; c =5.5; d = 150$ | 0.268 | 0.173 |
| $i$ (convective) | log-normal | $\sigma = 2.557; \mu = 0.694$ | 0.238 | 0.211 |
| $M$ (convective) | Poisson | $\lambda = 14.33$ | 0.871 | 0.756 |
| $P_{tot}$ (frontal) | Weibull | $\beta = 0.968; \gamma = 6.054; \mu = 3.00$ | 0.353 | 0.314 |
| $t_r$ (frontal) | Weibull | $\beta = 1.201; \gamma = 164.99; \mu = 150$ | 0.639 | 0.589 |
| $i$ (frontal) | log-normal | $\sigma = 1.485; \mu = 0.644$ | 0.906 | 0.878 |
| $M$ (frontal) | Poisson | $\lambda = 15.95$ | 0.372 | 0.831 |
| $P_{tot}$ (frontal, type I) | Weibull | $\beta = 0.862; \gamma = 4.535; \mu = 3.00$ | 0.631 | 0.425 |
| $t_r$ (frontal, type I) | beta | $\alpha = 1.221; \beta = 1.372; c =150; d = 270$ | 0.200 | 0.145 |
| $i$ (frontal, type I) | log-normal | $\sigma = 1.701; \mu = 0.612$ | 0.104 | 0.085 |
| $P_{tot}$ (frontal, type II) | Weibull | $\beta = 1.065; \gamma = 7.222; \mu = 3.00$ | 0.397 | 0.342 |
| $t_r$ (frontal, type II) | beta | $\alpha = 0.829; \beta = 1.562; c = 266; d = 650$ | 0.270 | 0.226 |
| $i$ (frontal, type II) | log-normal | $\sigma = 1.289; \mu = 0.611$ | 0.059 | 0.056 |

| $P_{tot}$ (convergence zone) | log-normal | $\sigma = 0.603$; $\mu = 3.00$ | 0.969 | 0.856 |
|---|---|---|---|---|
| $t_r$ (convergence zone) | Weibull | $\beta = 0.802$; $\gamma = 276.138$; $\mu = 650$ | 0.947 | 0.879 |
| $i$ (convergence zone) | log-normal | $\sigma = 1.296$; $\mu = 0.497$ | 0.942 | 0.923 |
| $M$ (convergence zone) | Poisson | $\lambda = 2.55$ | 0.067 | 0.652 |

frontal type I – cold front, frontal type II – warm front.

**Comment 6**

**Figures: Partly the units and symbols are missed. The caption of figures should be understandable and clear enough without reading the text. Figure 6 and 7: It would be favourable, if the both axes would have the same range. Not every reader is experienced in such analysis. One sentence or two sentences would be useful to explain, what the background of such pictures is. Instead of "Observed Value" (y-axis) it is recommended to write "Empirical Quantile". The caption could be: "Comparison of empirical and theoretical quantiles concerning the number of rainfall episodes and distinguishing rainfall types"**

In Figures 6 and 7 (P15L14-P16) the descriptions of X and Y axes have been modified in accordance with the drawings below.
The X axis is described as:
- theoretical quantiles of $P_{tot}$ values,
- theoretical quantiles of $t_r$ values,
- theoretical quantiles of $i$ values,
- theoretical quantiles of $M$ values,
The Y-axis is described as:
- empirical quantiles of $P_{tot}$ values,
- empirical quantiles of $t_r$ values,
- empirical quantiles of $i$ values,
- empirical quantiles of $M$ values,

[Figure]

Fig. 6. Comparison of empirical and theoretical quantiles concerning the number of rainfall episodes and distinguishing rainfall types: (a) all events (b) convective, (c) frontal, (d) in convergence zones.

[Figure]

Fig. 7: Comparison of quantiles of empirical and theoretical distributions of $P_{\text{tot}}$ and $t_r$ values for: (a) all events, (b) convective, (c) frontal, and (d) rainfall in convergence zones.

The title of Figure 6 has been corrected according to the reviewer's remark: „Comparison of empirical and theoretical quantiles concerning the number of rainfall episodes and distinguishing rainfall types: (a) all events, (b) convective, (c) frontal, (d) in convergence zones".

It is possible, but unreasonable, to perform Figure 6 in such a way as to maintain an identical range of variation in the annual number of rainfall events of different genesis. This is due to the large variation in the independent variable. Identically, this issue relates to the rainfall depth and its duration.

**Comment 7**
    **Figure 6: Is it right, that the sum of the highest values of b), c) and d) = 55 should be equal to the highest value of a)?**

Thank you for your valuable insight. The maximum values for the number of rainfall events caused by convectional, frontal and rainfall in convergence zones were recorded for the same year. This means that the sum does not have to be 55, but it may be less, as was obtained in the case under consideration.

**Comment 8**
    **Figure 8 (a): p is the probability of overflow discharge. But for wich variable stands CDF here? The reviewer has not found any remarks. Therefore he don't understand in line 18-5 why "the percentiel value p = 0,50 is as high as 0,90"? It looks like that the CDF-value 0,50 is as high as 0,90? But what means CDF here?**

Indeed, the markings introduced are misleading. The CDF value means the probability of non-exceedance the probability of stormwater overflow by a storm overflow in a rainfall episode. Thus, further modifications have been made to the manuscript:
- P18L5 now: ,,percentile value p = 0.50" after modification: ,,percentile value of 0.50"
- P19L9 now: ,, for example, for p = 0.05" after modification: ,, for example, for percentile value 0.05"
- P19L10 now: ,,for the percentile value p = 0.95" after modification: ,, for the percentile value 0.95"
- P20L4 now: ,,The influence of the theoretical distribution of the number of rainfall events per year on the values of $0.99 > p > 0.50$ is confirmed by Szeląg et al. (2018)" after modification: ,,The influence of the theoretical distribution of the number of rainfall events per year on the values of percentiel 0.50–0.99 is confirmed by Szeląg et al. (2018)".

**Comment 9**
    **Line 1-15: The Model is innovative, formulate more clear, what the reasons are**

The following manuscript text (P1L15): ,,This paper proposes an innovative probabilistic model to simulate the number of storm overflow discharges, which takes into atmospheric circulation and related rainfall in the research area (the city of Kielce located in the central part of Poland)."
has been modified to the following form:
,,The paper presents a probabilistic model for simulating the annual number of storm overflows. In this model, an innovative solution is to use the logistic regression method to analyze the impact of rainfall genesis on the functioning of a storm overflow on the example of a catchment located in Kielce city (central Poland).

The following manuscript text (P2L2): ,,They can be used to develop warning systems, in which information on the predicted rainfall genesis is a component of the assessment of the operation of the stormwater system and the facilities located on it."
has been expanded by the sentence:
,,This approach is an original solution that has not yet been considered by other researchers. On the other hand, it is an important simplification and an opportunity to reduce the data to be measured."

**Comment 10**
    **Line 1-24 and 7-21: The generator should be mentioned as first element.**

The text (P1L18) in the manuscript: ,, The first element is the model of logistic regression, which can be used to model storm overflow discharge resulting from the occurrence of a single rainfall episode. The paper confirmed that storm overflow discharge can be modeled on the basis of data on the total amount

of rainfall and its duration. An alternative approach was also proposed, in which the possibility of forecasting overflow discharge only on the basis of the average rainfall intensity was demonstrated, which is a big simplification in simulation of the phenomenon under study in comparison with the works published so far in this scope. It is worth noting that the coefficients determined in logit models have a physical interpretation and these models have a universal character, which is why they can be easily adapted to other examined catchment areas. The second element of the model is a synthetic precipitation generator, in which the simulation of rainfall takes into account its genesis resulting from various processes and phenomena taking place in the troposphere. This approach makes it possible to take into account the stochastic nature of rainfall also in relation to the annual number of events"

has been modified as follows (P1L18):

,,The first element of the model is a synthetic precipitation generator, in which the simulation of rainfall takes into account its genesis resulting from various processes and phenomena taking place in the troposphere. This approach makes it possible to take into account the stochastic nature of rainfall also in relation to the annual number of events. The first element is the model of logistic regression, which can be used to model storm overflow resulting from the occurrence of a single rainfall episode. The paper confirmed that storm overflow can be modeled on the basis of data on the total rainfall and its duration. An alternative approach was also proposed, in which the possibility of prediciting storm overflow only on the basis of the average rainfall intensity was demonstrated, which is a great simplification in simulation of the phenomenon under study in comparison with the works published so far in this scope. It is worth noting that the coefficients determined in logit models have a physical interpretation and these models have a universal character, which is why they can be easily adapted to other examined catchment areas."

The text (P7L21) in the manuscript: ,,The first component is a logit model, which is used to simulate the occurrence of a storm overflow discharge. Another component are synthetic precipitation generators, which are realized in two variants. In the first variant it was assumed that the basis for the simulation of rainfall series is their genesis. In the second variant precipitation is forecasted regardless of its origin – in the annual cycle."

has also been modified as follows:

,,The first component are synthetic precipitation generators, which are realized in two variants. In the first variant it was assumed that the basis for the simulation of rainfall series is their genesis. In the second variant precipitation is predicted regardless of its origin – in the annual cycle. Another component is a logit model, which is used to simulate the occurrence of a storm overflow."

**Comment 11**

**Line 2-14: write "such discharges". Mostly the words "overflow discharge" are used, but in some further cases only the word "discharge" (for example 8-4 and 18-29), while "overflow discharge" is meant. Please check such cases.**

The vocabulary was modified in the paper. Expression ,,overflow discharge" has been replaced by ,,storm overflow", and the expression ,,annual number of overflow discharges" has been replaced by ,,annual number of overflows".

**Comment 12**

**Line 5-22: write better "It concerns events with high intensity and short duration.**

The following manuscript text (P5L22): ,,These data were taken into account in the conducted analyses, as the launch of a new device, the SEBA electronic rain gauge (tipping-bucket SEBA rain gauge), a few years later in the state measuring network, resulted in the recording of significantly lower precipitation levels (by several percent) high intensity and short-lived, compared to measurements recorded by a traditional pluviograph (Kotowski et al., 2011)"

has been modified as follows:

,,These data were taken into account in the conducted analyses, as the launch of a new device, the SEBA electronic rain gauge (tipping-bucket SEBA rain gauge), a few years later in the state measuring network,

resulted in the recording of significantly lower precipitation levels (by several percent). It concerns events with high intensity and short duration (Kotowski et al., 2011)"

**Comment 13**
Lines 7-7 to 7-11: this paragraph seems to be a repeat.

Based on this Comment, the following manuscript text (P7L7 to P7L11): ,,Within the conducted analyses, an innovative probabilistic model was proposed for forecasting the number of storm overflow discharges (Figure 4). This model allows for the forecast of the annual number of discharges and the simulation of the number of events per year, taking into account the genesis of rainfall (convective in air mass, frontal, convergence zones precipitation), which is typical for countries located in central Europe and other regions of world. Although the paper focuses on the genesis of rainfall developed by Kupczyk and Suligowski (1997, 2011), the proposed approach is universal. The distribution of rainfall data may be based on local conditions determining the movement of air masses, which has a key impact on the dynamics of rainfall events. The time range of particular rainfall groups can then be determined on the basis of meteorological, synoptic and statistical analysis in the periods of high precipitation sums or precipitation intensity in a given area (Llasat, 2001; Rigo and Llasat, 2004; Millán et al., 2005; Langer and Reimer, 2007; Federico et al., 2008; Lazri et al., 2012; Berg and Haerter, 2013)."
has been modified as follows (P7L11):
,,An innovative probabilistic model was proposed for modelling the annual number of storm overflows (Figure 4). This model allows for the predict of the annual number of overflows and the simulation of the number of events per year, taking into account the genesis of rainfall, which is typical for countries located in central Europe and other regions of world. Although the paper focuses on the genesis of rainfall developed by Kupczyk and Suligowski (1997, 2011), the proposed approach is universal. The distribution of rainfall data may be based on local conditions determining the advection of air masses, which has a key impact on the dynamics of rainfall events. The time range of particular rainfall groups can then be determined on the basis of meteorological, synoptic and statistical analysis in the periods of high precipitation sums or precipitation intensity in a given area (Llasat, 2001; Rigo and Llasat, 2004; Millán et al., 2005; Langer and Reimer, 2007; Federico et al., 2008; Lazri et al., 2012; Berg and Haerter, 2013)."

**Comment 14**
Lines 8-9 to 9-2: Both sentences sound similar.

The following sentences are indeed similar: ,,The paper presents the following stages of construction of a probabilistic model on the example of an urban catchment located in the area of Kielce city. In the following sections the individual steps of the above mentioned calculation algorithm of the probabilistic model (separation of rainfall events, creation of a logistic regression model, development of a rainfall generator) are discussed in detail (Figure 4).

Therefore, the text has been modified as follows (P8L9): ,,The paper presents the following stages of construction of a probabilistic model on the example of an urban catchment located in Kielce city."

**Comment 15**
Line 9-5: What is meant by "in the ranks of"

The following sentence (P9L5) has been modified: ,, One of the basic conditions allowing for the completion of a synthetic precipitation generator is the separation of single independent rain events in the ranks of rainfall"
do postaci:
,, One of the basic conditions allowing for the completion of a synthetic precipitation generator is the separation of single independent rain events in rainfall time series".

**Comment 16**
Lines 9-9 to 9-18: this paragraph seems to be a repeat.

In the first quoted sentence (P9L9) the reader was informed only about empirical distributions, determined on the basis of separated rain episodes. In the next paragraph of the text this information is detailed and the methodology of research is presented. Thus, although the sentences P9L9 and P9L18 are similar, both are of significant importance in the applied research methodology.

**Comment 17**
        Line 9-24: "simulate objects" sounds strange, write better "simulate the influence of constructions on flow procsses"

The following sentence (P9L24): ,,It is also used to simulate objects located in rainwater drainage networks (storm overflows) (Szeląg et al., 2018)"
has been modified as follows:
,,It is also used to simulate the influence of constructions on flow processes."

**Comment 18**
        Line 10-14: The investigation period is 1961 to 2000 (page 5). Here the years 2012-2014 are discussed?

Description of the data used to construct the model (P10L11-15): ,,The second variant assumes simplification and considers a single independent variable, i.e. average rainfall intensity. To determine the logit model, the results of measurements of the operation of the investigated storm overflow have been used from the years 2009-2011, when 69 overflow of 188 precipitation events occurred, and from the years 2012-2014, when 42 overflow of 93 precipitation events occurred."
has been made more specific:
,,In the urban catchment area, continuous flow measurements were carried out in the period 2009-2011 (69 storm overflows during 188 rainfall events were separated at that time), whereas in the years 2012-2014 only fillings in the diversion chamber were measured (42 overflows during 93 rainfall episodes were separated). The reason for this was the construction works carried out in the analysed catchment and a large amount of suspended solids limiting the operation of the measuring devices. Since 2015, MES1 and MES2 flow meters have been installed, which also allow the measurement of the volume of stormwater discharged by overflow. Thus, on the basis of data from the period 2009-2014, a logit model was developed, while data from the period 2015-2017 were used to verify it."

**Comment 19**
        Line 11-14: Whant is meant by "period separating subsequent rainfall events"?

The following sentence (P11L14): ,,Currently conducted research in the field of rainfall simulators based on multidimensional boundary distributions combined with the so-called dome functions take into account the distribution of rainfall in the rainfall episode (Vernieuwe et al., 2015), spatial diversity of rainfall (Dai et al., 2014), seasons (Khedun et. al., 2014) and the period separating subsequent rainfall events (Balistrocchi and Bacchi, 2011)."
has been modified as follows:
        ,,Currently conducted research in the field of rainfall simulators based on multidimensional boundary distributions combined with the so-called copula functions take into account the distribution of rainfall in the rainfall episode (Vernieuwe et al., 2015), spatial diversity of rainfall (Dai et al., 2014), seasons (Khedun et. al., 2014) and the antecendent period (Balistrocchi and Bacchi, 2011)."

**Comment 20**
        Line 11-15 to 11-24: This part concerns not methods. Similar discussions are in the first parts of the paper.

The text from P11L15 to P11L24 and P10L17 to P10L24 is shortened and included in the introduction (P2L22) in the following form:

„Multidimensional scaling methods and fractal geometry (Rupp et al., 2009; Licznar et al., 2015; Müller-Thomy and Haberlandt 2015) are used to simulate rainfall series. An alternative solution is an approach based on multidimensional distributions created on the basis of theoretical distributions and copula functions (Vandenberghe et al., 2010; Vernieuwe et al., 2015). Despite numerous applications, these solutions are relatively complex and require expert knowledge. For the storm overflow simulation, hydrodynamic models are usually used, and less frequently empirical models (Szeląg et al., 2018). Nevertheless, this approach to the simulation of the annual number of overflows is very local and in many cases requires the construction of a catchment model."

Due to the above correction and Comment 5, the text after P9L19 was also modified:
"Taking into account the computational algorithm described in Chapter 4 (Methodology), on the basis of determined distributions of theoretical rainfall characteristics describing the operation of a storm overflow, a model for simulation of synthetic rainfall series was adopted for further analysis. The simulations carried out for this purpose included the modified Monte Carlo - Iman-Conover method (1982). This model gives the possibility to simulate independent variables on the basis of determined theoretical distributions.
In this method the variability of the considered variables is described by boundary (theoretical) distributions, and the basis for evaluation of their correlation is the Spearman correlation coefficient. The conditions, which must be met in order for the results obtained to be considered correct, are as follows:
– in the data obtained from simulation and measurements, the mean values ($\mu_1(x_1)$, $\mu_2(x_2)$,..., $\mu_i(x_i)$)$_s$ and the standard deviations ($\sigma_1(x_1)$, $\sigma_2(x_2)$,...,$\sigma_i(x_i)$)$_s$ of the variables ($x_i$) considered in j samples do not differ by more than 5%,
– theoretical distributions of $x_i$ variables obtained from simulation are consistent with those obtained from measurements; in order to meet this condition it is recommended to use the Kolmogorov-Smirnov test,
– the value of the correlation coefficient (R) between individual dependent variables ($x_i$) obtained for data from MC simulation does not differ by more than 5% from the value of R obtained for empirical data."

**Comment 21**
**Lines 12-8 to 12-11: This sentences are nearly a repeat of pages 7/8, but the steps are not denominated identical. Here 4 steps are listed, but the chapter consists of the two parts 5.1 and 5.2 only.**

Indeed, the text on page P12L8 to P12L11 is similar to this one on pages 7/8. The text (P12L8 to P12L11) lists in detail the stages that are discussed in the manuscript below.
To eliminate the similarity, the Section 5.2 has been divided into three following chapters:

5.2. Identification of empirical distributions and theoretical rainfall characteristics
Subchapter 5.2 includes analyses related to the determination of statistical distributions of the following variables: rainfall depth ($P_{tot}$), rainfall duration ($t_r$), average rainfall intensity ($i$), and number of rain events in a year of varied genesis.
Chapter 5.2 in the manuscript contains the text from P13L24 to P17L5.

5.3. Impact of rainfall genesis on the probability of overflow occurring
Subchapter 5.3 presents the determined relationship between the genesis of rainfall and the probability of overflow event, as well as the ranges of variation of average rainfall, taking into account the rainfall genesis, which determines the occurrence of storm overflow event.
Subchapter 5.3 in the manuscript contains the text from P17L6 to P17L27.

5.4. Impact of precipitation genesis on the annual number of overflow events
Subchapter 5.4 presents the annual number of overflow events caused by rainfall (convective, frontal, in convergence zones). At the same time, a comparative analysis of the annual number of overflow events obtained with a simplified logit model (based on the average rainfall intensity) and the number of overflow events obtained with an accurate model (based on the total amount of rainfall and its duration) is carried out in this chapter.
Subchapter 5.4 in the manuscript contains the text from P18L28 to P20L16 in the manuscript.

**Comment 22**

Line 13-19: The reviewer don't know what "values of free words" are? Possibly other readers will have the same problem.

The following sentence (P13L19 – P13L21): ,,On the basis of the relationships (eq. 5 and eq. 6) it can be concluded that the values of free words obtained in them are similar to the weighted average value of the catchment retention ($d_{av}$). The relative difference between the values of free words and retention of the catchment area does not exceed 5 % "
has been modified as follows:
,,On the basis of this relationships it can be concluded that the values of intercept obtained in them are similar to the weighted average value of the catchment retention ($d_{av}$). The relative difference between the values of intercept and retention of the catchment area does not exceed 5 %.

**Comment 23**

Line 1-16: The text within the brackets should be formulated as sub-clause or as an additional sentence.

The manuscript text has been corrected (see Comment and Answer 9).

**Comment 24**

Line 1-21: write "great great" instead of "big"

The amended sentence is set out in answer to Comment 10.

**Comment 25**

Line 1-29: two times determine

The following sentence (P1L29): ,,On the basis of the obtained results, the range of variability of average rainfall intensity was determined, which determines the discharge by storm overflow, as well as the annual number of discharges resulting from the occurrence of rain of different genesis."
has been modified as follows:
,,On the basis of the obtained results, the range of variability of average rainfall intensity was defined, which determines the storm overflow, as well as the annual number of overflows resulting from the occurrence of rain of different genesis."

**Comment 26**

Line 1-31: the results are suited for implementation

The following sentence (P1L31): ,,The obtained results enable their practical implementation in the assessment of storm overflows only on the basis of knowledge concerning the genetic type of rainfall."
has been modified as follows:
,,The results are suited for implementation in the assessment of storm overflows only on the basis of the genetic type of rainfall."

**Comment 27**

Line 2-2: three times the word "of" in series.

The following sentence (P2L2): ,,They can be used to develop warning systems, in which information on the predicted rainfall genesis is a component of the assessment of the operation of the stormwater system and the facilities located on it."
has been modified as follows:
„They may be used to develop warning systems in which information on the predicted rainfall genesis is an element of assessment of the rainwater system and its facilities. "

**Comment 28**
**Line 2-29: what was not taken into account when rainfall generators were used to simulate**

The following sentence (P2L29): ,,It seems puzzling why the fact that the time course and dynamics of rainfall are the result of complex movements of air masses (Serrano et al., 2009; Alhammoud et al., 2014; Dayan et al., 2015) was not taken into account when modelling rainfall generators to simulate storm overflows."
has been modified as follows:
,,It seems puzzling why the fact that the time course and dynamics of rainfall are the result of complex movements of air masses (Serrano et al., 2009; Alhammoud et al., 2014; Dayan et al., 2015) what was not taken into account when rainfall generators used to simulate storm overflows."

**Comment 29**
**Line 2-32: "concern simulations" sounds strange, write perhaps better "consider"**

The following sentence (P2L32): ,,The models created concern simulations of meteorological conditions changing in time and determining the distribution of temperature…."
has been modified as follows:
,,The models created consider simulations of meteorological conditions changing in time and determining the distribution of temperature…."

**Comment 30**
**Line 3-1: "course of precipitation phenomena" sounds strange.**

The following sentence (P3L1): ,,The models created concern simulations of meteorological conditions changing in time and determining the distribution of temperature, pressure and humidity, which affects the dynamics of air movement and, consequently, the course of precipitation phenomena."
has been modified as follows:
,,The models created concern simulations of meteorological conditions changing in time and determining the distribution of temperature, pressure and humidity, which affects the dynamics of air movement and, consequently, the patterns of precipitation phenomena."

**Comment 31**
**Line 3-4: "forecasting the operation" sounds strange, write "basis for the control of systems"**

The following sentence (P3L4): ,,This information may be the basis for forecasting the operation of the stormwater system and developing an early warning system against the risks of flash flood."
has been modified as follows:
,,This information may be the basis for control of the systems and developing an early warning system against the risks of flash flood."

**Comment 32**
**Line 3-11: Sometime it is written "model of the rainfall generator". The generator is a model, the word "model" seems to be unnessecary.**

The following sentence (P3L11): „In the model of the rainfall generator the genesis of rainfall was taken into account, which allowed to determine the curves showing the influence of rainfall genesis on the occurrence of overflow discharge in a single rainfall episode."
has been modified as follows:
„In the rainfall generator the genesis of rainfall was taken into account, which allowed to determine the curves showing the influence of rainfall genesis on the occurrence of storm overflow in a single rainfall episode."

**Comment 33**

**Line 3-22: a space is missed**

A space is inserted between the sentences (P3L22): ........ (Szeląg et al., 2016). The length .......

**Comment 34**
**Line 3-23: better "height difference", cancel "of ordinates"**

The following sentence (P3L23): ,,The maximum difference of ordinates in the catchment is 12.0 m and the average slope in the catchment is 7.1 %."
has been modified as follows:
,,The height difference in the catchment is 12.0 m and the average slope – 7.1 %."

**Comment 35**
**Line 4-10: "generated" better as "shaped"**

The following sentence (P4L10): ,,Precipitation is shaped by two different precipitation mechanisms: convective and stratiform (Houze, 2014)".
has been modified as follows:
,,Precipitation is generated by two different precipitation mechanisms: convective and stratiform (Houze, 2014)".

**Comment 36**
**Line 4-11: write "A third", since before only two mechanism are announced**

The following sentence (P4L11): ,,The third rainfall mechanism, which may have the above mentioned components, is related to the orographic lifting of air masses over mountains or hills (Smith, 1993)."
has been modified as follows:
,,A third rainfall mechanism, which may have the above mentioned components, is related to the orographic lifting of air masses over mountains or hills (Smith, 1993)."

**Comment 37**
**Line 4-11: write "which include both above mentioned components"**

The following sentence (P4L11): ,,The third rainfall mechanism, which may have the above mentioned components, is related to the orographic lifting of air masses over mountains or hills (Smith, 1993)."
has been modified as follows:
,,A third rainfall mechanism, which inlude both above mentioned components, is related to the orographic lifting of air masses over mountains or hills (Smith, 1993)."

**Comment 38**
**Line 4-16: write at the end "are" instead of "is"**

The following sentence (P4L16): ,,Convective precipitation induced by single thunderstorm cells, their complexes or squall lines is short-lived, but is characterized by high average intensity (Kane et al., 1987) and causes flash floods in many areas (Gaume et al., 2009; Marchi et al., 2010; Bryndal, 2015)."
has been modified as follows:
,,Convective precipitation induced by single thunderstorm cells, their complexes or squall lines is short-lived, but are characterized by high average intensity (Kane et al., 1987) and causes flash floods (Gaume et al., 2009; Marchi et al., 2010; Bryndal, 2015)."

**Comment 39**
**Line 5-2 and 5-13: line break (new paragraph)**

We agree with the Reviewer's #1 comment that the considerations on this subject presented in Chapter 3 may be shortened and partly moved to the introductory chapter of the article.

Therefore, the following changes have been made to the manuscript:

– the title of Chapter 3 (P4L7) has been changed as follows: "Rainfall data and analysis"

– the previous content presented in the text from P4L8 to P5L17 was modified in the following way and moved to Chapter 1 (after P3L7):

"Rainfall is universally classified into three types (Sumner, 1988): convective, cyclonic and orographic. The main distinguishing feature between convective precipitation in air mass and frontal precipitation in mid-latitudes is its spatial extent and duration. The range of convective precipitation associated with local air circulation is much smaller than in the case of travelling extratropical cyclones with weather fronts. Convective precipitation induced by single thunderstorm cells, their complexes or squall lines is short-lived, but are characterized by high average intensity (Kane et al., 1987) and causes flash floods (Gaume et al., 2009; Marchi et al., 2010; Bryndal, 2015). On the other hand, the lifespan of the mechanisms of creating cyclonic precipitation is much longer than that of convective precipitation – in the order of days rather than hours. Hence, the effect of this is long-term rainfall with a high depth (Frame et al., 2017), often causing regional floods (Barredo, 2007). The presented classification of precipitation types distinguished by Sumner (1988) due to the origin, developed for the British Isles and Western Europe, cannot be directly applied in practical hydrology in other regions of the continent, especially in its eastern and central parts. This is the result of exceptional variability of meteorological conditions occurring in the temperate zone of warm transition climate – on the borderline of air masses coming from the Atlantic and continental masses from the east (Twardosz and Niedźwiedź, 2001; Niedźwiedź et al., 2009; Twardosz et al., 2011; Łupikasza, 2016). Analysis of maximum rainfall of different duration in Poland carried out at the end of the 1990s (Kupczyk and Suligowski, 1997, 2011), supplemented by the analysis of synoptic situation (on the base of surface synoptic charts of Europe, published in Daily Meteorological Bulletin of the Institute of Meteorology and Water Management – IMGW in Warsaw) and a calendar describing the types of atmospheric circulation together with air masses and air fronts (Niedźwiedź, 2019), led to the separation of three types of genetic precipitation: convective in air mass, frontal and generated in convergence zones."

**Comment 40**

Line 5-17: "convergence zone" is not a type of precipitation, write better "generated in convergence zones"

The amended text is presented at the end of the answer to Comment 39.

**Comment 41**

Line 5-19: write "only these data were"

Fragment of the sentence (P5L19): ,,These data were taken into account in the conducted analyses,…."
has been modified as follows:
,,Only these data were taken into account in the conducted analyses, …."

**Comment 42**

Line 5-33: "variable" instead of "varied"

The following sentence (P5L33): ,,The second type (frontal rainfall) forms a group of precipitation in Kielce, in which the duration is very varied and ranges from 2.5 h to 10.5 h."
has been modified as follows:
,,The second type (frontal rainfall) forms a group of precipitation in Kielce, in which the duration is very variable and ranges from 2.5 h to 10.5 h."

**Comment 43**

Line 6-12: "precipitation emitted" sounds strange.

The following sentence (P6L12): ,,Transformation of air masses over the western part of the continent, lower speeds of movement of frontal zones, as well as weakening of the dynamics of processes in the front zone cause that precipitation in Kielce differ in intensity and duration in relation to precipitation emitted by Sumner (1988) as cyclonic."

has been modified as follows:

,,Transformation of air masses over the western part of the continent, lower speeds of movement of frontal zones, as well as weakening of the dynamics of processes in the front zone cause that precipitation in Kielce differ in intensity and duration in relation to precipitation defined by Sumner (1988) as cyclonic."

**Comment 44**
**Line 7-25: overflow discharges per year**

The following sentence (P7L25): ,,On this basis, distribution functions (CDF) are determined that describe the probability (Z) of exceeding the number of storm overflow discharges."

has been modified as follows:

,,On this basis, distribution functions (CDF) are determined that describe the probability of non exceeding the annual number of storm overflows."

**Comment 45**
**Line 10-10: Write ". is a simplification. It considers only a single"**

The following sentence (P10L10): ,,The second variant assumes simplification and considers a single independent variable, i.e. average rainfall intensity."

has been modified as follows:

,,The second variant is a simplification. It considers only a single."

**Comment 46**
**Line 10-18: set methods before the brackets.**

The above remark has already been taken up in answer to Comment 20 and the quotes have been moved to the end of the sentence.

**Comment 47**
**Line 10-30: write better "should be consistent"**

The following sentence (P10L30): ,,…theoretical distributions of $x_i$ variables obtained from simulation are consistent with those obtained from measurements; in order to meet this condition it is recommended to use the Kolmogorov-Smirnov test…"

has been modified as follows:

,,…theoretical distributions of $x_i$ variables obtained from simulation should be consistent with those obtained from measurements; in order to meet this condition it is recommended to use the Kolmogorov-Smirnov test…"

**Comment 48**
**Line 13-16: write "are valid" instead of "take place"**

The following sentence (P13L16): ,,Based on theoretical considerations conducted by Thorndahl and Willems (2008), who provided a generalised model for forecasting the volume of wastewater discharge via a storm overflow, it can be concluded that in this case the following relations take place:"

has been modified as follows:

,,Based on theoretical considerations conducted by Thorndahl and Willems (2008), who provided a generalised model for modeling the overflow volume, it can be concluded that in this case the following relations are valid:"

**Comment 49**

**Line 13-19: write "of this relationship" and cancel the "(eq. 5 and eq. 6)"**

The corrected sentence is in answer to Comment 22.

**Comment 50**

**Line 15-8: expressed better by**

The following sentence (P15L8): ,,Also, the variation in rainfall duration in episodes resulting from rainfall of different genesis in most cases is described by the Weibull distribution and only in the case of data measured over an annual cycle is it expressed by the GEV distribution (eq. 9)."
has been modified as follows:
,,Also, the variation in rainfall duration in episodes resulting from rainfall of different genesis in most cases is described by the Weibull distribution and only in the case of data measured over an annual cycle is it expressed better by the GEV distribution (eq. 9)."

**Comment 51**

**Line 18-3: Write "distinguished" instead of "made".**

The following sentence (P18L3): „Within the framework of the conducted analyses, the division of frontal rainfall events of the duration not longer than 4.5 h (related to the cold front – type I) and exceeding the given value (due to the displacement of the warm front – type II) was additionally made."
has been modified as follows:
„Within the framework of the conducted analyses, the division of frontal rainfall events of the duration not longer than 4.5 h (related to the cold front – frontal type I) and exceeding the given value (due to the displacement of the warm front – frontal type II) was additionally distinguished.

**Comment 52**

**Line 18-6: line break (new paragraph).**

Chapter 5.2 and its division is discussed in answer to Comment 21.

**Comment 53**

**Following words seems to be unnessecary: Line 1-20: was demonstrated; Line 1-31/32: knowledge concerning; Line 2-10: collecting; Line 2-19: in the work; Line 2-22: of simulation; Line 2-27: in its modeling; Line 3-26: the work; Line 4-6: article ;Line 4-17: in many areas; Line 5-3: of the phenomenon; Line 9-6: in the paper ;Line 10-10: in the analysis performer ;Line 11-13: in the rainfall episode ; Line 13-1: using the model.**

All words indicated by the reviewer have been deleted.

---

## Author Comment (AC3) · 4 Oct 2019

**Reviewer #1**

Thank you for carefully reading the manuscript and providing constructive suggestions and comments to improve the article. Our answers to question/comment are provided below.

**Comment 1**

**Wording: a lot of terms not common in the literature are used throughout the manuscript, e.g., receiver should be replaced by receiving waters. What do you mean by episode? Is it an event or a period or the event duration? Please be more specific! Movement of air could be replaced by advection. The term forecast is not appropriate in this context. I would replace this term by prediction. There is a lot of confusion regarding the usage of overflow and discharge. You do not predict discharge, only the number of overflow events.**

The authors fully agree with the comments made. The text has been amended and supplemented as suggested. Precipitation episodes in the manuscript mean independent precipitation events, which have been separated from long-term rainfall series. As a result, the first sentence in Chapter 4.1 (P9L4) was modified as follows:
One of the basic conditions allowing for the completion of a synthetic precipitation generator is the separation of single independent rain events (episodes) in the ranks of rainfalls.

**Comment 2**

**Independent validation: Even though the logit model seems to perform very well, it does not become clear how well the model works in my opinion. Is there any chance to add some comparisons with observations in the Figure 9 (at least averages as vertical lines)? Is it possible to perform a split sample test in order to analyze if the model chain provides reliable results for an independent period of time not involved into parameter estimation?**
**Transferability of the stochastic approach: as already mentioned, for me it remains unclear how the method could be transferred to other urban catchments. You argue that the catchment detention is physically meaningful. I agree in principle but I could imagine that a lot of other catchment characteristics might be relevant too. For instance, what is the impact of the network structure on the results? Even though the detention is tangible, its empirical deviation (Eq. 5 and 6) is rather empirical. Is there any chance to quantify this value with simplified hydrological calculations? This would support your argumentation regarding the transferability and the practical relevance. It might be worth to discuss the added value when compared to long-term simulation using hydrological (and hydrodynamic) models.**

Thank you very much for your comments. Chapter 4 has been modified. The order of its subchapters has been changed and in the new layout it is as follows:

4. Methodology
4.1. Simulation of the annual number of storm overflow events using a hydrodynamic model
4.2. Logistic regression
4.3. Separation of rain events and synthetic rainfall simulator
4.4. Annual rainfall series simulator

A new Subchapter 4.1 has been introduced in Chapter 4 entitled "Simulation of the annual number of storm overflow events using a hydrodynamic model" (P9L3). The content of it is as follows:
"One of the possible solutions allowing for the verification of empirical dependencies describing the operation of stormwater systems is the simulation of this operation with the use of a calibrated hydrodynamic model. It is an approach applied in engineering practice, which is confirmed by a number of works in this field (Bacchi et al. 2008; Doménech et al. 2010). Simulations performed with a

hydrodynamic model on the basis of rainfall data allow to verify the prediction capabilities of probabilistic models designed to simulate the quantity and quality of stormwater and the operation of separate objects located on the stormwater system (tanks, overflows).

Within the framework of the conducted analyses, a calibrated model of the catchment basin made in the SWMM (Storm Water Management Model) program was used to verify the number of storm overflow events. The total area of the examined catchment is 62 ha, while the area of partial catchments is from 0.12 ha to 2.10 ha. The number of stormwater junction in the catchment area is 200 and the number of stormwater pipes located in this area is 72. The retention depth of the imperviousness surfaces of the catchment area is 2.5 mm and the retention depth of the pervious surfaces is 6.0 mm. The roughness coefficient of imperviousness areas is equal to 0.025 [$m^{-1/3}$ s] and of pervious areas 0.250 [$m^{-1/3}$ s]. The roughness coefficient of the stormwater channels is equal to 0.018 [$m^{-1/3}$ s].

The results of simulation with hydrodynamic model were compared with the results of calculations made with the use of logit models for the data from the examined catchment from the period 2008-2016, which allowed to verify the determined relation $p = f(x_1, x_2, x_3, ..., x_i)$. Simultaneously, using the measurement data from the period 1961-2000, a simulation of the annual number of storm overflow events was performed, taking into account the precipitation genesis, which allowed for verification of the developed probabilistic model."

Subchapter 5.1 has also been modified. Its title is now "Logit model and its verification". The following text is included in this subchapter below the logit models description therein (P12L13-P13L11):
"The values of the *SENS* (sensitivity) and *SPEC* (specificity) coefficients are usually calculated to assess the predictive capability of the models. However, *SENS = f(SPEC)* may also be used for this purpose. In this case, the higher the area value (the maximum value is *AUC = 1*) between the *SENS = SPEC* and *SENS = f(SPEC)* curves, the more accurate the model will be (Fig. A1).

[Figure]

Fig. A1. Relation *SENS = f(SPEC)* for the logit curves $p = f(i)$ and $p = f(P_{tot}, t_r)$.

To verify the predictive capabilities of the logit models and their dependencies, a calibrated hydrodynamic model was used. The results of the simulation obtained with the hydrodynamic model for data from the period 2008-2016 were compared with the results of calculations made with the use of logit models $p = f(i)$ and $p = f((P_{tot}, t_r)$ (see Table A1).

Table A1: Comparison of measurements of the annual number of overflow events with the calculations results obtained with hydrodynamic and logit models

| Year | $Z_a$ | $Z_{SWMM}$ | $Z_{logit(P,t)}$ | $Z_{logit(i)}$ |
|------|-------|-----------|------------------|----------------|
| 2008 | 13 | 15 | 14 | 12 |
| 2009 | 15 | 16 | 16 | 14 |
| 2010 | 17 | 18 | 19 | 16 |
| 2011 | 19 | 20 | 19 | 17 |
| 2012 | 13* | 21 | 20/14* | 19/11* |
| 2013 | 13* | 22 | 20/13* | 20/11* |
| 2014 | 16* | 29 | 28/16* | 27/14* |
| 2015 | 25 | 26 | 26 | 23 |
| 2016 | 26 | 22 | 21 | 24 |
| 2017 | 16 | 17 | 17 | 14 |

The markings in Tab. 1 mean: $Z_a$ – the measured annual number of storm overflow events in the analyzed urban catchment area, SWMM – the modeled number of storm overflow events by means of a calibrated hydrodynamic model of the catchment area, $Z_{logit(P,t)}$ – the modeled number of storm overflow events by means of the relation $p = f(P_{tot}, t_r)$, $Z_{logit(i)}$ – modeled number of storm overflow events using the relation $p = f(i)$, * – storm overflow events in the period (2012-2014) determined on the basis of the maximum filling of the diversion chamber (DC)."

Referring to comment No. 2, the text in Subchapter 5.1 (from P13L12 to P13L23) has also been modified:
"The above relations (3), (4) are of a local character and can be applied in principle only in relation to the analysed catchment area. With this in mind, an attempt was made to build a universal model by modifying these relations. Based on the theoretical considerations of Thorndahl and Willems (2008), who investigated the relationships between the characteristics of catchment areas, including the limit retention that determines the volume of overflow, the following equations were determined:

$$0.566\, P_{tot} - 0.004\, t_r - 2.152 = 0 \,/\cdot \frac{1}{0.566} = P_{tot} - 0.007\, t_r - 3.802 = 0 \qquad (a1)$$

$$P_{tot} - 0.007 t_r - 3.802 = P_{tot} - 0.007 t_r - d_{av} = 0 \qquad (a2)$$

$$0.312\, i - 1.257 = 0 \,/\cdot \frac{1}{0.312} = i - 4.03 = 0 \qquad (a3)$$

$$i - 4.03 = i - d_{av} = 0 \qquad (a4)$$

Analysing the values of coefficients in equations (a2) and (a4), one can see that the values of the free word in these equations are close to the value of the weighted mean of the retention depth of the catchment area ($d_{av}$). The difference between the weighted mean of the catchment retention and the values of the intercept in the equations does not exceed 5 %. Simulation calculations of the annual number of overflow events in the tested catchment area confirm the above statement (Table A1). At the same time, this fact is supported by the results obtained in Fig. A2, which indicate that the results of the simulation of the annual number of overflow events using the hydrodynamic model are within the scope of a probabilistic solution. However, due to the fact that the presented analyses were performed only for a single catchment, it is necessary to verify the obtained results using examples of other catchments with different physical and geographical characteristics."

**Comment 3**
        **I would suggest revising the manuscript. Some things are too detailed (Section 3) or too general (e.g., the discussion on rainfall models that are not used here in Sect. 4.3; I would expect some methodology rather than an introduction to this topic). Sect. 5.2 should be sub-divided in order to increase readability.**

Thank you very much for your comments. The contents of Chapters 3, 4 and 5 have been modified. We consider Chapter 3 to be an important part of our work in terms of the goals we have set. The article discusses the problem of storm overflows in the urban catchment area, caused by rainfall of different origins. One of the elements of the proposed probabilistic model for modeling the number of storm overflow events is a generator of synthetic rainfall. In order to simulate this model, rainfall events with different genesis had to be taken into account. The conducted analyses allowed to determine the influence of the distribution of the average annual number of precipitation episodes with different genesis of precipitation on the variability of the number of storm overflow events. Therefore, in our opinion, it is important to familiarize the reader with the main processes and mechanisms occurring in the troposphere over central Europe, which are reflected in high precipitation. The presentation of this problem is presented in the introduction to Chapter 3 of the manuscript. The main characteristics of convective precipitation, frontal precipitation and convergence zones presented in this chapter are the result of the exceptional variability of meteorological conditions in this part of the European continent and are therefore presented in a broader context. However, we agree with the Reviewer's comment that the considerations on this subject presented in Chapter 3 may be shortened and partly moved to the introductory chapter of the article.

Therefore, the following changes have been made to the manuscript:

– the title of Chapter 3 (P4L7) has been changed as follows: "Rainfall data and analysis"
– the previous content presented in the text from P4L8 to P5L17 was modified in the following way and moved to Chapter 1 (after P3L7):
"Rainfall is universally classified into three types (Sumner, 1988): convective, cyclonic and orographic. The main distinguishing feature between convective precipitation in air mass and frontal precipitation in mid-latitudes is its spatial extent and duration. The range of convective precipitation associated with local air circulation is much smaller than in the case of travelling extratropical cyclones with weather fronts. Convective precipitation induced by single thunderstorm cells, their complexes or squall lines is short-lived, but is characterized by high average intensity (Kane et al., 1987) and causes flash floods in many areas (Gaume et al., 2009; Marchi et al., 2010; Bryndal, 2015). On the other hand, the lifespan of the mechanisms of creating cyclonic precipitation is much longer than that of convective precipitation – in the order of days rather than hours. Hence, the effect of this is long-term rainfall with a high depth (Frame et al., 2017), often causing regional floods (Barredo, 2007). The presented classification of precipitation types distinguished by Sumner (1988) due to the origin of the phenomenon, developed for the British Isles and Western Europe, cannot be directly applied in practical hydrology in other regions of the continent, especially in its eastern and central parts. This is the result of exceptional variability of meteorological conditions occurring in the temperate zone of warm transition climate – on the borderline of air masses coming from the Atlantic and continental masses from the east (Twardosz and Niedźwiedź, 2001; Niedźwiedź et al., 2009; Twardosz et al., 2011; Łupikasza, 2016). Analysis of maximum rainfall of different duration in Poland carried out at the end of the 1990s (Kupczyk and Suligowski, 1997, 2011), supplemented by the analysis of synoptic situation (on the base of surface synoptic charts of Europe, published in Daily Meteorological Bulletin of the Institute of Meteorology and Water Management – IMGW in Warsaw) and a calendar describing the types of atmospheric circulation together with air masses and air fronts (Niedźwiedź, 2019), led to the separation of three types of genetic precipitation: convective in air mass, frontal and convergence zones."

– also Figure 2 (P6L1) has been removed from Chapter 3.

The main changes in Chapter 4 have been discussed earlier. In addition, the following introduction text from the existing Chapter 4.3 ('Synthetic precipitation generator'), i.e. contained from P10L17 to P10L23, has been moved to the introductory chapter (P2L22):
"Multidimensional scaling and fractal geometry methods are used to simulate rainfall series (Rupp et al., 2009; Licznar et al., 2015; Müller-Thomy and Haberlandt 2015). Alternative approaches are

solutions based on multidimensional distributions created on the basis of theoretical distributions and copula functions (Vandenberghe et al., 2010; Vernieuwe et al., 2015). However, these solutions are relatively complex and require expert knowledge. To predict the annual number of storm overflow events, hydrodynamic models are usually used, whereas less frequent are empirical models (Szeląg et al. 2018). However, this approach is very local and in many cases requires the construction of a catchment model."

In view of the above changes, the text after P9L19 has also been modified as follows:
"Taking into account the calculation algorithm described in Chapter 4 (Methodology) and developed on the basis of theoretical distributions of rainfall characteristics describing the operation of a storm overflow, a model for simulating synthetic rainfall series was used for further analyses. The model was determined using the Monte Carlo method modified by Iman-Conover (1982). This model gives the possibility of simulation of independent variables taken into account in the research on the basis of theoretical distributions determined for them. In the Monte Carlo method used the variability of the considered variables is described by boundary (theoretical) distributions, and the basis for evaluation of their correlation is the Spearman correlation coefficient. The conditions, which must be met in order for the results obtained to be considered correct, are as follows:

– in the data obtained from simulation and measurements, the mean values $(\mu_1(x_1), \mu_2(x_2),...,\mu_i(x_i))_s$ and the standard deviations $(\sigma_1(x_1),\sigma_2(x_2),...,\sigma_i(x_i))_s$ of the variables $(x_i)$ considered in j samples do not differ by more than 5%,

– theoretical distributions of $x_i$ variables obtained from simulation are consistent with those obtained from measurements; in order to meet this condition it is recommended to use the Kolmogorov-Smirnov test,

– the value of the correlation coefficient $(R)$ between individual dependent variables $(x_i)$ obtained for data from MC simulation does not differ by more than 5 % from the value of $R$ obtained for empirical data.

If the above mentioned conditions are met, the results of the simulation performed with the IC method can be considered correct. If this is not the case, the sample size of the MC needs to be increased (Wu and Tsang, 2004). In order to limit the sample size and improve the efficiency of the Iman-Conover algorithm, a modification has been developed by using the Latin-Hybercube algorithm, which is part of the layered sampling methods aimed at improving the "uniformity" of the numbers generated from the boundary distributions.

On the basis of the determined boundary distributions of rainfall characteristics, simulations of synthetic rainfall series were performed with the use of the Monte Carlo method with Iman-Conover modifications and taking into account the Latin-Hypercube algorithm."

With reference to Subchapter 5.2, the text of it was divided into the following three Subchapters:

5.2. Identification of empirical distributions and theoretical rainfall characteristics
Subchapter 5.2 includes analyses related to the determination of statistical distributions of the following variables: rainfall depth ($P_{tot}$), rainfall duration ($t_r$), average rainfall intensity ($i$), and number of rain events in a year of varied genesis.
Chapter 5.2 in the manuscript contains the text from P13L24 to P17L5.

5.3. Impact of rainfall genesis on the probability of storm overflow occurring
Subchapter 5.3 presents the determined relationship between the genesis of rainfall and the probability of overflow event, as well as the ranges of variation of average rainfall intensity, taking into account the rainfall genesis, which determines the occurrence of storm overflow event.
Subchapter 5.3 in the manuscript contains the text from P17L6 to P17L27.

5.4. Impact of precipitation genesis on the annual number of overflow events

Subchapter 5.4 presents the annual number of overflow events caused by rainfall (convective, frontal, in convergence zones). At the same time, a comparative analysis of the annual number of overflow events obtained with a simplified logit model (based on the average rainfall intensity) and the number of overflow events obtained with an accurate model (based on the total rainfall and its duration) is carried out in this chapter.
Subchapter 5.4 in the manuscript contains the text from P18L28 to P20L16 in the manuscript.

The text in P18L30 has been modified as follows:
"Simulation calculations performed with the calibrated hydrodynamic model of the catchment area show that the results of the simulation of the annual number of storm overflow events (Fig. A2) caused by precipitation of different genesis are within the scope of the solution obtained with the probabilistic model. This confirms that the probabilistic model developed in the paper is an alternative solution to the hydrodynamic model. The above fact is also confirmed by the annual number of overflow events caused by rainfall of different genesis obtained on the basis of measurements. (Fig. A2). The conformity of simulation results obtained with the probabilistic and hydrodynamic model may indicate that the values of free words determined in the relevant equations may correspond to the weighted average retention of the catchment area.

[Figure]

**Fig. A2.** (a) Distribution function *(CDF)* showing the annual number of storm overflow events caused by convective rainfall *($Z_c$)*; (b) Distribution function showing the annual number of overflow events caused by frontal rainfall *($Z_f$)*; (c) Distribution function showing the annual number of overflow events caused by rainfall in convergence zones *($Z_{cz}$)*; (d) Curve showing the probability of non exceeding the annual number of overflow events *(Z)*; *M* – number of rain events in a year*: M = const($x_i$)* – in the probabilistic model the average number of rain events from the period 1961-2000 was adopted in the calculations; *M = var($x_i$)* – in the probabilistic model theoretical distributions describing the number of rain events in a year were adopted (Table A1)."

**Specific comments:**
**P2L5; National guidelines?**

Indeed, the quoted guidelines are foreign. The paper also takes into account national Polish guidelines in the form of the Regulation of the Minister of the Environment of 18 November 2014 on the conditions to be met when discharging sewage into water or soil and on the substances particularly harmful to the aquatic environment.

**P3L18; I would suggest adding some more details on the catchment area (e.g. dryweather flow).**

The characteristics of the catchment area have been extended. In item P3L24 of the manuscript the following additional information has been added:
"Analysis of the measurement data (2008-2017) concerning the analysed catchment area showed that the antecedent period there lasted 0.16–60 days and storms occurred 27–47 times a year. The average annual air temperature during the period under consideration varied from 8.1 to 9.6°C. At the same time, the analysis of measurement data of flows recorded with the MES1 flow meter showed that in the antecedent periods the temporary stream of stormwater was from 0.001 to 0.009 $m^3s^{-1}$, which indicates the occurrence of infiltration in the stormwater system under study."

**Figure 4: When I read the paper for the first time, my first idea was that you compare genesis of rainfall vs. not distinguishing the genesis as shown in Figure 4. In Figure 9, however, you mainly compare the average number of events vs. a modelled number of events (including the comparison genesis vs. generalization of events?)**

Taking as a starting point the diagram of the algorithm in Fig. 4 (P8L8), two approaches to the simulation of the annual number of overflow events were analysed. In the first one the genesis of rainfall $M = const(P_{tot}, t_r)$ *or $M = const(i)$* (Fig. A2d) was omitted. In the second approach the developed rainfall generator (Fig. 5), described in Subchapter 4.4 ("Simulator of synthetic annual rainfall series"), was used and the results obtained are shown in Fig. A2d in the form of $M = var(P_{tot}, t_r)$ curve. The additional dependencies showed in Fig. A2a, A2b, A2c are a detailed description of the results of the simulation analyses and illustrate the annual number of overflow events caused by rainfall of various origins (convective, frontal, convergence zones). Indeed, in the original version of the manuscript, this may lead to confusion. The corrected manuscript therefore contains the following table of symbols, parameters and abbreviations at the end of the article (Table A2):

| N.m. | Symbol | Meaning | |
|------|--------|---------|---|
| 1 | *AUC* | area under curve | |
| 2 | *CDF* | cumulative distribution function of probability density | |
| 3 | $d_{av}$ | weighted mean of the retention depth of the catchment area | |
| 4 | *f(x)* | probability density function | |
| 5 | $F(x)_c$ | theoretical distribution to simulate rainfall characteristics due to rainfall: | convective |
| 6 | $F(x)_f$ | | frontal |
| 7 | $F(x)_{cz}$ | | in convergence zones |
| 8 | $F(\zeta)_c$ | theoretical distribution to simulate the annual number of rainfall events: | convective |
| 9 | $F(\zeta)_f$ | | frontal |
| 10 | $F(\zeta)_{cz}$ | | in convergence zones |
| 11 | *i* | average rainfall intensity [$L·s^{-1} ha^{-1}$] | |
| 12 | *IC* | Iman-Conover method | |
| 13 | *M* | annual number of rainfall events | |

| 14 | $M = const(i)$ | calculation variant (annual number of overflow events), in which for simulation constant average annual number of rainfall events is used and to identify the overflow in a rainfall episode the following logit models were applied: | logit model $p = f(i)$ |
|---|---|---|---|
| 15 | $M = const(P_{tot}, t_r)$ | | logit model $p = f(P_{tot}, t_r)$ |
| 16 | $M = var(i)$ | calculation variant (annual number of overflow events), in which the annual number of rainfall events caused by precipitation (convective, frontal, in convergence zones) is modelled, and for identification of overflow in a rainfall episode the following logit models were applied: | logit model $p = f(i)$ |
| 17 | $M = var(P_{tot}, t_r)$ | | logit model $p = f(P_{tot}, t_r)$ |
| 18 | $M_c$ | annual number of rainfall events caused by rainfall: | convective |
| 19 | $M_f$ | | frontal |
| 20 | $M_{cz}$ | | in convergence zones |
| 21 | $MC$ | Monte Carlo method | |
| 22 | $N$ | number of samples in the Monte Carlo simulation | |
| 23 | $p$ | probability of a storm overflow event | |
| 24 | $P_{tot}$ | total rainfall [mm] | |
| 25 | $t_r$ | rainfall duration [min] | |
| 26 | $R$ | Spearman's correlation coefficient | |
| 27 | $R_z^2$ | counting error | |
| 28 | $SENS$ | sensitivity | |
| 29 | $SPEC$ | specificity | |
| 30 | $SWMM$ | Storm Water Management Model | |
| 31 | $Z$ | annual number of storm overflow events | |
| 32 | $Z_c$ | annual number of storm overflow events due to rainfall: | convective |
| 33 | $Z_f$ | | frontal |
| 34 | $Z_{cz}$ | | in convergence zones |
| 35 | $x_i$ | independent variables included in the logit model | |
| 36 | $\alpha_i$ | values of estimated coefficients in the logit model | |
| 37 | $\alpha, \beta, \sigma, \lambda, \mu, \gamma, \zeta$ | empirical coefficients estimated in statistical distributions | |
| 38 | $(\mu_1(x_1), \mu_2(x_2),...,\mu_i(x_i))_s$ | mean value of variable $x_i$ in the data set obtained from simulation using the Iman-Conover method | |
| 39 | $(\sigma_1(x_1), \sigma_2(x_2),...,\sigma_i(x_i))_s$ | value of standard deviation of variable $x_i$ in the data set obtained from simulation using the Iman-Conover method | |

**P5L9; The discussion on an "increase in the roughness of the substrate" is awkward in my opinion and not correct. Is this discussion really needed here? I would suggest rephrasing this section in a way that makes the explanations more concise, given the topic of the paper.**

Thank you for the valuable comment and signalling that one of the reasons given for the variability in the frequency, intensity and duration of rainfall associated with atmospheric fronts in Central Europe was unclear and awkward. The discussion on this issue was omitted, as mentioned above (in response

to Comment No. 3). The existing content presented from P4L8 to P5L17 was modified and moved to Chapter 1 (after P3L7).

**P17L4; This statement remains unclear to me. Please be more precise! Why do you abandon this approach? I found this explanation confusing. I thought that M was modelled or even assumed to be constant? How is this related to your argumentation?**

The text of the manuscript in P17L3 has been modified as follows:
"The aim of the simulation calculations was to compare the results of calculations of the annual number of overflow events by means of a model taking into account the genesis of rainfall and a simplified model, in which only a differentiated number of rainfall events was included with the omission of the rainfall genesis. Reference data constituting the basis for the comparison was the number of storm overflow events obtained taking into account the rainfall genesis – $M = var(P_{tot}, t_r)$ (Fig. A2d). Due to the fact that the determined statistical distribution describing the annual number of rainfall events (Fig. 6a) indicated overestimation of the minimum values and underestimation of the maximum values, the obtained results of the simulation of the annual number of overflow events on the basis of such distribution would differ from the values obtained on the basis of the empirical distribution. Thus, the comparison of the simulation results between the simplified model and the model taking into account the rainfall genesis would be burdened with an error, which would have a negative impact on the evaluation and interpretation of the calculation results. Obtained results of the simulation with a simplified model and one taking into account rainfall genesis can of course be compared, but it would be difficult to draw conclusions from this comparison to determine how the simplification introduced in the simulation of the annual number of overflow events affects the results of the simulation."

In the corrected manuscript, items P17L4 and P17L5 were removed.

**P18L31; : I don't think that there is any exact model. Please rephrase**

Thank you for your comments. The term 'exact model' has been replaced by the term 'model that takes into account the genesis of rainfall'.

**P2L19; Do you mean Thornsal and Willams (2008)?**

The order of names has been changed: 'Willems and Thorndahl (2008) is now replaced by ''Thorndahl and Willems (2008)'.

**P11L3; What is IC? Iman-Conover?**

Indeed, IC is the abbreviation for the Iman-Conover method.

**P3L2; Consider dropping knowledge.**

The word 'knowledge' has been removed from the text.

---

## Author Response (AR1)

**Dear Editor and Referees,**

Thank you for the quality of your proofreading and comments; they have greatly improved the manuscript. The methodology and results were adjusted accordingly. Below, we address the points risen by the two anonymous reviewers and state how we would like to address them in a revised version of the manuscript. Our replies to the reviewers' comments are written in black and green and  normal font to distinct them from the reviewers' comments.

We believe we have substantially addressed all of the outstanding comments and issues, and we look forward to your second review of the work.

On the behalf of all co-authors,

Yours sincerely,

Bartosz Szeląg

**Reviewer #1**

Thank you for carefully reading the manuscript and providing constructive suggestions and comments to improve the article. Our answers to question/comment are provided below.

**Comment #1**
      **Wording: a lot of terms not common in the literature are used throughout the manuscript, e.g., receiver should be replaced by receiving waters. What do you mean by episode? Is it an event or a period or the event duration? Please be more specific! Movement of air could be replaced by advection. The term forecast is not appropriate in this context. I would replace this term by prediction. There is a lot of confusion regarding the usage of overflow and discharge. You do not predict discharge, only the number of overflow events.**

The authors fully agree with the comments made. The text has been amended and supplemented as suggested. Precipitation episodes in the manuscript mean independent precipitation events, which have been separated from long-term rainfall series. As a result, the first sentence in Section 4.1 was modified as follows:
„One of the basic conditions allowing for the completion of a synthetic precipitation generator is the separation of single independent rain events (episodes) in rainfall time series."

**Comment #2**
      **Independent validation: Even though the logit model seems to perform very well, it does not become clear how well the model works in my opinion. Is there any chance to add some comparisons with observations in the Figure 9 (at least averages as vertical lines)? Is it possible to perform a split sample test in order to analyze if the model chain provides reliable results for an independent period of time not involved into parameter estimation?**
**Transferability of the stochastic approach: as already mentioned, for me it remains unclear how the method could be transferred to other urban catchments. You argue that the catchment detention is physically meaningful. I agree in principle but I could imagine that a lot of other catchment characteristics might be relevant too. For instance, what is the impact of the network structure on the results? Even though the detention is tangible, its empirical deviation (Eq.**

**5 and 6) is rather empirical. Is there any chance to quantify this value with simplified hydrological calculations? This would support your argumentation regarding the transferability and the practical relevance. It might be worth to discuss the added value when compared to long-term simulation using hydrological (and hydrodynamic) models.**

Thank you very much for your comments. The change in the order of the contents discussed in the manuscript has a positive influence on the clarity of the considerations. Moreover, a very helpful comment was made concerning the supplementation of the calculation results with simulations with the calibrated hydrodynamic model of the catchment. These calculations were performed and the results were found to be consistent with the results obtained by means of the probabilistic model, which confirms the universal character of the relation given in the paper. The comparison of curves describing the average rainfall intensity limiting the occurrence of storm overflow (Fig. 6) was also valuable from the point of view of model verification; the curves were obtained based on simulation results and measurement data.

Section 4 has been modified. The order of its sections has been changed and in the new layout it is as follows:
4. Methodology
4.1. Simulation of the annual number of storm overflow events using a hydrodynamic model
4.2. Logistic regression
4.3. Separation of the rain event and synthetic rainfall simulator
4.4. Simulator of annual rainfall series
A new Section 4.1 has been introduced in Section 4 entitled "Simulation of the annual number of storm overflow events using a hydrodynamic model". The content of it is as follows:
"One of the possible solutions allowing for the verification of empirical dependencies describing the operation of stormwater systems is the simulation of this operation with the use of a calibrated hydrodynamic model. It is an approach applied in engineering practice, which is confirmed by a number of works in this field (Bacchi et al., 2008; Andrés-Doménech et al., 2010). The simulations performed with a hydrodynamic model based on rainfall data allow the verification of the prediction capabilities of the probabilistic models designed to simulate the quantity and quality of stormwater, and the operation of separate objects located in the stormwater system (tanks, overflows).
Within the framework of the conducted analyses, a calibrated model of the catchment basin made in the SWMM (storm water management model) program was used to verify the annual number of storm overflows (Figure 1). The total area of the analysed catchment is 62 ha, while the area of partial catchments ranges from 0.12 ha to 2.10 ha. The number of stormwater junctions in the catchment area is 200, and the number of stormwater pipes is 72. The retention depth of the imperviousness surfaces of the catchment area is 2.5 mm and the retention depth of the pervious surfaces is 6.0 mm. The roughness coefficient of impervious areas is equal to 0.025 $m^{-1/3} \cdot s$, and that of pervious areas is 0.250 $m^{-1/3} \cdot s$. The roughness coefficient of the stormwater channels is equal to 0.018 $m^{-1/3} \cdot s$.
The results of the simulation with the hydrodynamic model were compared with the results of calculations made with the use of logit models for the data from the examined catchment from the period 2008-2017, which allowed the verification of the determined relation $p = f(x_1, x_2, x_3, ..., x_i)$. Simultaneously, using the measurement data from the period of 1961-2000, a simulation of the annual number of storm overflow events was performed, taking into account the precipitation genesis, which enabled the verification of the developed probabilistic model."

Section 5.1 has also been modified. Indeed, at present, the SENS and SPEC values form the basis for the assessment of the predictive capabilities of the logit model in the majority of studies. Graphical interpretation of the relationship between them is expressed by the AUC value. Therefore, we would like to thank you for your remark, because the curves in Figure 5 are a valuable addition to the results of the calculations and are easy to read when interpreting the results.
Section 5.1 title is now "Logit model and its verification". The following text is included in this section below the logit models description therein:
"The values of the SENS (sensitivity) and SPEC (specificity) coefficients are usually calculated to assess the predictive capability of the models. However, SENS = $f$(SPEC) may also be used for this purpose. In this case, the greater the area value (the maximum value is AUC = 1) between the SENS = SPEC and SENS = $f$(SPEC) curves is, the more accurate the model will be (Figure 5).

[Figure]

Figure 5: Relation SENS = $f$(SPEC) for the logit curves $p = f(i)$ and $p = f(P_{tot}, t_r)$

To verify the predictive capabilities of the logit models and their dependencies, a calibrated hydrodynamic model was used. The results of the simulation obtained with the hydrodynamic model for data from the period 2008-2017 were compared with the results of the calculations made with the use of the logit models $p = f(i)$ and $p = f(P_{tot}, t_r)$ (see Table 1).

Table 1: Comparison of the measurements of the annual number of overflow events with the calculations results obtained with hydrodynamic and logit models

| Year | $Z_a$ | $Z_{SWMM}$ | $Z_{logit(P,t)}$ | $Z_{logit(i)}$ |
|------|-------|------------|------------------|----------------|
| 2008 | 13 | 15 | 14 | 12 |
| 2009 | 15 | 16 | 16 | 14 |
| 2010 | 17 | 18 | 19 | 16 |
| 2011 | 19 | 20 | 19 | 17 |
| 2012 | 13* | 21 | 20/14* | 19/11* |
| 2013 | 13* | 22 | 20/13* | 20/11* |
| 2014 | 16* | 29 | 28/16* | 27/14* |
| 2015 | 25 | 26 | 26 | 23 |
| 2016 | 26 | 22 | 21 | 24 |
| 2017 | 16 | 17 | 17 | 14 |

\* – storm overflow events in the period 2012-2014 determined on the basis of the maximum filling of the diversion chamber (DC). Explanations of symbols are in the Appendix."

In order to assess the convergence of the results of calculations obtained with the use of the probabilistic model with the measurements, the ranges of variability of rain intensity were calculated, which determines the occurrence of storm overflow, taking into account the rainfall genesis. For comparison, Fig. 8b shows the results of calculations (made on the basis of measurement data) of the value of the $i$ parameter that determines a storm overflow. Based on Fig. 8b it can be stated that the results of the calculations obtained with the use of the probabilistic model are satisfactory.

[Figure]

Figure 8: (a) Impact of rainfall genesis on: a) the probability of storm overflow, (b) rainfall intensity distribution determining storm overflow in a single episode.

Referring to Comment #2, the text in Section 5.1 (from P13L12 to P13L23) has also been modified:
"The above relations (3) and (4) are of local character and can be applied in principle only in relation to the analysed catchment area. With this in mind, an attempt was made to build a universal model by modifying these relations. Based on theoretical considerations of Thorndahl and Willems (2008), who investigated the relationships between the characteristics of catchment areas, including the limit retention that determines the volume of overflow, the following equations were determined:

$$0.566\,P_{tot} - 0.004\,t_r - 2.152 = 0\ /\cdot\frac{1}{0.566} = P_{tot} - 0.007\,t_r - 3.802 = 0 \tag{5}$$

$$P_{tot} - 0.007t_r - 3.802 = P_{tot} - 0.007t_r - d_{av} = 0 \tag{6}$$

$$0.312\,i - 1.257 = 0\ /\cdot\frac{1}{0.312} = i - 4.03 = 0 \tag{7}$$

$$i - 4.03 = i - d_{av} = 0 \tag{8}$$

On the basis of these relationships, it can be concluded that the values of the intercept obtained in them are similar to the weighted average value of the catchment retention ($d_{av}$). The relative difference between the values of intercept and retention of the catchment area does not exceed 5 %. The simulation calculations of the annual number of overflow events in the tested catchment area confirm the above statement (Table 1). In addition, this fact is supported by the results obtained in Figure 5, which indicate that the results of the simulation of the annual number of overflow events using the hydrodynamic model are

within the scope of a probabilistic solution. However, because the presented analyses were performed only for a single catchment, it is necessary to verify the obtained results using examples of other catchments with different physical and geographical characteristics."

Indeed, we agree with the Reviewer that the average retention of a catchment is a not simple parameter to determine. In view of the above, the authors envisage a continuation of the studies which will aim at linking the value of the intercept determined in equations (5) to (8) to the CN curves. However, this requires further analysis, which will continue at a later stage.

**Comment #3**
    **I would suggest revising the manuscript. Some things are too detailed (Section 3) or too general (e.g., the discussion on rainfall models that are not used here in Sect. 4.3; I would expect some methodology rather than an introduction to this topic). Sect. 5.2 should be sub-divided in order to increase readability.**

Thank you very much for your comments. Of course, the introduction of subsequent sections will significantly improve the readability of the contents discussed in the manuscript.
The contents of Section 3, 4 and 5 have been modified. We consider Section 3 to be an important part of our work in terms of the goals we have set. The article discusses the problem of storm overflows in the urban catchment area, caused by rainfall of different origins. One of the elements of the proposed probabilistic model for modelling the number of storm overflow events is a generator of synthetic rainfall. In order to simulate this model, rainfall events with different genesis had to be taken into account. The conducted analyses allowed to determine the influence of the distribution of the average annual number of precipitation episodes with different genesis of precipitation on the variability of the number of storm overflow events. Therefore, in our opinion, it is important to familiarize the reader with the main processes and mechanisms occurring in the troposphere over central Europe, which are reflected in high precipitation. The presentation of this problem is presented in the introduction to Section 3 of the manuscript. The main characteristics of convective precipitation, frontal precipitation, and rainfall in convergence zone presented in this Section are the result of the exceptional variability of meteorological conditions in this part of the European continent and are therefore presented in a broader context. However, we agree with the Reviewer's comment that the considerations on this subject presented in Section 3 may be shortened and partly moved to the introductory Section of the article.
Therefore, the following changes have been made to the manuscript:
– the title of Section 3 has been changed as follows: "Rainfall data and analysis"
– the previous content presented in the text from P4L8 to P5L17 was modified in the following way and moved to Section 1 (after P3L7):
"Rainfall is universally classified into three types (Sumner, 1988; Smith, 1993): convective, cyclonic and orographic. The main distinguishing feature between convective precipitation in an air mass and frontal precipitation in mid-latitudes is its spatial extent and duration. The range of convective precipitation associated with local air circulation is much smaller than in the case of travelling extratropical cyclones with weather fronts. Convective precipitation induced by single thunderstorm cells, their complexes or squall lines is short-lived but is characterized by high average intensity (Kane et al., 1987) and causes flash floods (Gaume et al., 2009; Marchi et al., 2010; Bryndal, 2015). On the other hand, the lifespan of the mechanisms of creating cyclonic precipitation is much longer than that of convective precipitation – on the order of days rather than hours. Hence, the effect of this is long-term rainfall with a high depth (Frame et al., 2017) often cause regional floods (Barredo, 2007). The presented classification of precipitation types distinguished by Sumner (1988) due to the origin, developed for the British Isles and Western Europe, cannot be directly applied in practical hydrology in other regions of the continent, especially in its eastern and central parts due to exceptional variability of meteorological conditions occurring in the temperate zone of the warm transition climate – on the border of air masses coming from the Atlantic and continental masses from the east (Twardosz and Niedźwiedź, 2001; Niedźwiedź et al., 2009; Twardosz et al., 2011; Łupikasza, 2016). The analysis of maximum rainfall of different duration in Poland carried out at the end of the 1990s (Kupczyk and Suligowski, 1997, 2011), supplemented by an analysis of the synoptic situation (based on surface synoptic charts of Europe, published in the Daily Meteorological Bulletin of the Institute of Meteorology and Water Management – IMGW in Warsaw) and a calendar describing the types of atmospheric circulation together with air masses and air fronts (Niedźwiedź, 2019), led to the separation of three types of genetic precipitation: convective in air mass, frontal and generated in convergence zone."

– also Figure 2 (P6L1) has been removed from Section 3.

The main changes in Section 4 have been discussed earlier. In addition, the following introduction text from the existing Section 4.3 ("Synthetic precipitation generator"), i.e. contained from P10L17 to P10L23, has been moved to the introductory Section (P2L22):

"Multidimensional scaling methods and fractal geometry (Rupp et al., 2009; Licznar et al., 2015; Müller-Thomy and Haberlandt, 2015) are used to simulate rainfall series. An alternative solution is an approach based on the multidimensional distributions created based on theoretical distributions and copula functions (Vandenberghe et al., 2010; Vernieuwe et al., 2015). Despite numerous applications, these solutions are relatively complex and require expert knowledge. For the storm overflow simulation, hydrodynamic models are usually used, and less frequently, empirical models are used (Szeląg et al., 2018). Nevertheless, this approach to the simulation of the annual number of overflows is very local and, in many cases, requires the construction of a catchment model."

In view of the above changes, the text after P9L19 has also been modified as follows:
"Taking into account the computational algorithm described at the beginning of Section 4 (Methodology), based on the determined distributions of the theoretical rainfall characteristics describing the operation of a storm overflow, a model for the simulation of a synthetic rainfall series was adopted for further analysis. The simulations carried out for this purpose included the Monte Carlo method with modification of Iman-Conover (1982). This model provides the possibility simulating the independent variables based on the determined theoretical distributions. In this method, the variability of the considered variables is described by boundary (theoretical) distributions, and the basis for the evaluation of their correlation is the Spearman correlation coefficient. The conditions that must be met in order for the results obtained to be considered correct, are as follows.
– In the data obtained from simulation and measurements, the mean values $(\mu_1(x_1), \mu_2(x_2),...,\mu_i(x_i))_s$ and the standard deviations $(\sigma_1(x_1), \sigma_2(x_2),...,\sigma_i(x_i))_s$ of the variables $(x_i)$ considered in $j$ samples do not differ by more than 5 %.
– The theoretical distributions of $x_i$ variables obtained from simulation should be consistent with those obtained from measurements; in order to meet this condition it is recommended to use the Kolmogorov-Smirnov (KS) test.
– The value of the correlation coefficient (R) between the individual dependent variables $(x_i)$ obtained for the data from Monte Carlo (MC) simulation does not differ by more than 5 % from the value of R obtained for empirical data.
If the above mentioned conditions are met, the results of the simulation performed with the Iman-Conover (IC) method can be considered correct. If these conditions are not met, the sample size of the MC needs to be increased (Wu and Tsang, 2004). To limit the sample size and improve the efficiency of the Iman-Conover (IC) algorithm, a modification has been developed by using the Latin hybercube (LH) algorithm, which is part of the layered sampling methods aimed at improving the "uniformity" of the numbers generated from the boundary distributions.
Based on the determined boundary distributions of rainfall characteristics, the simulations of the synthetic rainfall series were performed with the use of the Monte Carlo (MC) method with Iman-Conover (IC) modifications and taking into account the Latin hybercube (LH) algorithm."

With reference to Section 5.2, the text of it was divided into the following three 4 Sections:
5.2. Identification of the empirical and theoretical distributions of the selected rainfall characteristics
Section 5.2 includes analyses related to the determination of statistical distributions of the following variables: rainfall depth ($P_{tot}$), rainfall duration ($t_r$), average rainfall intensity ($i$), and number of rain events in a year of varied genesis.
Section 5.2 in the manuscript contains the text from P13L24 to P17L5 (in original manuscript).

5.3. Impact of rainfall genesis on the probability of storm overflow occurrence
Section 5.3 presents the determined relationship between the genesis of rainfall and the probability of overflow event, as well as the ranges of variation of average rainfall intensity, taking into account the rainfall genesis, which determines the occurrence of storm overflow event.
Section 5.3 in the manuscript contains the text from P17L6 to P17L27 (in original manuscript).

**5.4. Impact of rainfall genesis on the average rainfall intensity occurrence of a storm overflow**

Section 5.4 presents the results of calculations of the impact of rainfall genesis on the rainfall intensity limit values determining the occurrence of storm overflow. Section 5.4 covers the text from P18L3 to P18L27.

**5.5. Impact of rainfall genesis on the annual number of overflow events**

Section 5.5 presents the annual number of overflow events caused by rainfall (convective, frontal, and in convergence zone). At the same time, a comparative analysis of the annual number of overflow events obtained with a simplified logit model (based on the average rainfall intensity) and the number of overflow events obtained with an accurate model (based on the total rainfall and its duration) is carried out in this Section.
Section 5.5 in the manuscript contains the text from P18L28 to P20L16 (in original manuscript).

The text in P18L30 has been modified as follows:
"The simulation calculations performed with the calibrated hydrodynamic model of the catchment area show that the results of the simulation of the annual number of storm overflow events (Figure 9d) caused by precipitation of different genesis are within the scope of the solution obtained with the probabilistic model. This finding confirms that the probabilistic model developed in the paper is an alternative solution to the hydrodynamic model. This fact is also confirmed by the annual number of overflow events caused by rainfall of different genesis obtained on the basis of measurements (Figure 9d). The conformity of the simulation results obtained with the probabilistic and hydrodynamic model may indicate that the values of the intercept determined in the relevant equations may correspond to the depth of the weighted average retention of the catchment area."

[Figure]

Figure 9: Distribution function (CDF) showing the annual number of storm overflow events ($Z$) caused by: (a) convective rainfall, (b) frontal rainfall, (c) rainfall in convergence zone; (d) Curve showing the probability of non-exceeding the annual number of overflows ($Z$). Explanations of the other symbols are in the Appendix.

**Specific comments:**
**P2L5; National guidelines?**

Indeed, the quoted guidelines are foreign. The paper also takes into account national Polish guidelines in the form of the Regulation of the Minister of the Environment of 18 November 2014 on the conditions to be met when discharging sewage into water or soil and on the substances particularly harmful to the aquatic environment.

**P3L18; I would suggest adding some more details on the catchment area (e.g. dryweather flow).**

The characteristics of the catchment area have been extended. In item P3L24 of the manuscript the following additional information has been added:
"The analysis of the measurement data (2008-2017) concerning the analysed catchment area showed that the antecedent period lasted 0.16–60 days, and storms occurred 27–47 times a year. The average annual air temperature during the period under consideration varied from 8.1 to 9.6 °C. In addition, the analysis of the measurement data of flows recorded with the MES1 flow meter showed that in the antecedent periods, the temporary stream of the stormwater was from 0.001 to 0.009 $m^3 \cdot s^{-1}$, which indicates the occurrence of infiltration in the stormwater system under study."

**Figure 4: When I read the paper for the first time, my first idea was that you compare genesis of rainfall vs. not distinguishing the genesis as shown in Figure 4. In Figure 9, however, you mainly compare the average number of events vs. a modelled number of events (including the comparison genesis vs. generalization of events?)**

Thank you kindly for your valuable remark.
Taking as a starting point the diagram of the algorithm in Fig. 4 (P8L8), two approaches to the simulation of the annual number of overflow events were analysed. In the first one the genesis of rainfall $M = const(P_{tot}, t_r)$ or $M = const(i)$ (Fig. 9) was omitted. In the second approach the developed rainfall generator (Fig. 5), described in Section 4.4 ("Simulator of synthetic annual rainfall series"), was used and the results obtained are shown in Fig. 9d in the form of $M = var(P_{tot}, t_r)$ curve. The additional dependencies showed in Fig. 9a, 9b, 9c are a detailed description of the results of the simulation analyses and illustrate the annual number of overflow events caused by rainfall of various origins (convective, frontal, in convergence zone).
Taking into account the valuable suggestion of the Reviewer, the appendix contains a list of symbols used in the manuscript. Indeed, in the original version of the manuscript, this may lead to confusion. The corrected manuscript therefore contains the following table of symbols and abbreviations (Appendix):

Appendix 1. List of symbols and abbreviations

[revised manuscript text omitted]

**P5L9; The discussion on an "increase in the roughness of the substrate" is awkward in my opinion and not correct. Is this discussion really needed here? I would suggest rephrasing this section in a way that makes the explanations more concise, given the topic of the paper.**

Thank you for the valuable comment and signalling that one of the reasons given for the variability in the frequency, intensity and duration of rainfall associated with weather fronts in Central Europe was unclear and awkward. The discussion on this issue was omitted, as mentioned above (in response to Comment #3). The existing content presented from P4L8 to P5L17 was modified and moved to Section 1.

**P17L4; This statement remains unclear to me. Please be more precise! Why do you abandon this approach? I found this explanation confusing. I thought that M was modelled or even assumed to be constant? How is this related to your argumentation?**

Thank you very much for your remark. It turned out to be very useful in the improvement of substantive assessments resulting from simulations and thus in the improvement of the conclusions formulated. Indeed, the aim of the study was to compare the results of calculations of the annual number of storm overflows, taking into account the simplifications associated with omitting or including rainfall genesis in the analyses. Currently, in section 4.4 ("Simulator of annual synthetic rainfall series") the calculation variants considered in the paper are discussed. In the paper finally calculations for all considered simulation variants were made. The variant in which the changing number of rain events in the year $M = f(P_{tot}, t_r, M = var)$ was taken into account was also analysed. The results obtained for this case may have a great practical significance in the methodology of designing storm overflows. The results of the simulation showed that omitting the precipitation genesis and taking into account only the variability of the annual number of precipitation events, i.e. variant $M = f(P_{tot}, t_r, M = var)$, results in overestimation of the annual number of storm overflows. Thus, in engineering practice, it means that the height of the overflow edge (level) is inflated, which results in hydraulic overload (e.g. of a rainwater treatment plant).
In the corrected manuscript, items P17L4 and P17L5 were removed.

**P18L31; I don't think that there is any exact model. Please rephrase**

Thank you for your comments. The term "exact model" has been replaced by the term "model that takes into account the genesis of rainfall".

**P2L19; Do you mean Thornsal and Willams (2008)?**

The order of names has been changed: 'Willems and Thorndahl (2008) is now replaced by "Thorndahl and Willems (2008)'.

**P11L3; What is IC? Iman-Conover?**

Indeed, IC is the abbreviation for the Iman-Conover method.

**P3L2; Consider dropping knowledge.**

The word "knowledge" has been removed from the text.

**Reviewer #2**

We thank the reviewer for the effort and the time spend on this manuscript.
Thank you very much for putting forward very concrete proposals for improving it. The article after the correction was checked by a specialist in hydrology and mathematical modelling in urban drainage basins.

Below we present detailed responses to the following comments.

**Comment #1**

**The authors have used some variables. It would be useful to add a table with the most important variables, their meaning and unit.**

At the end of the paper (in Appendix) there is a table with a list of all used symbols and abbreviations.

Appendix 1. List of symbols and abbreviations

[revised manuscript text omitted]

The following sentence (P10L28):
,,in the data obtained from simulation and measurements, the mean values $(\mu_1(x_1), \mu_2(x_2),...,\mu_i(x_i))_s$ and the standard deviations $(\sigma_1(x_1), \sigma_2(x_2),...,\sigma_i(x_i))_s$ of the variables $(x_i)$ considered in j samples do not differ by more than 5 %''
has been modified as follows:
,, in the data obtained from simulation and measurements, the mean values $(\mu_1(x_1), \mu_2(x_2),...,\mu_i(x_i))_s$ and the standard deviations $(\sigma_1(x_1), \sigma_2(x_2),...,\sigma_i(x_i))_s$ of the variables $(x_i)$ considered in $j$ samples do not differ by more than 5 % .''

**Comment #2**

      In line 9-11 Ptot and tr are mentioned, but defined later in line 9-29. There it would be better to write q = Ptot / tr = 166.7 . . .

      The reviewer knows q as specific discharge ore runoff rate, but not as rain intensity. In English parers for rain intensity stands often I or i (sometimes PI for precipitation intensity).

The following text (P9L11):

„In order to obtain the best possible matching of theoretical data (precipitation characteristics including $P_{tot}$ and $t_r$ values for precipitation of appropriate genesis) with empirical data, the following statistical distributions were considered …"
has been modified as follows:

„To obtain the best possible fit of the theoretical data to the empirical precipitation data (including: rainfall depth – $P_{tot}$, rainfall duration – $t_r$ and average rainfall intensity – $i$, for precipitation of appropriate genesis), the following statistical distributions were considered …"

The designations in Figures 8 and 9 below have thus been changed:

[Figure]

Figure 8: (a) Impact of rainfall genesis on: a) the probability of storm overflow, (b) rainfall intensity distribution determining storm overflow in a single episode.

[Figure]

Figure 9: Distribution function (CDF) showing the annual number of storm overflow events (Z) caused by: (a) convective rainfall, (b) frontal rainfall, (c) rainfall in convergence zone; (d) Curve showing the probability of non-exceeding the annual number of overflows (Z). Explanations of the other symbols are in the Appendix.

In view of the modification of the designations in Figures 8 and 9, further following corrections have been made:
- P13L27; text „($P_{tot}$, $t_r$, q, M)" has been modified as follows: „($P_{tot}$, $t_r$, $i$, $M$)"
- P18L18; text „q = 4.49–35.50 L s$^{-1}$ ha$^{-1}$" has been modified as follows: „$i$ = 4.49–35.50 L·s$^{-1}$·ha$^{-1}$"
- P18L22; text „it corresponds to q = 4.49–35.50 L·s$^{-1}$·ha$^{-1}$" has been modified as follows: „it corresponds to $i$ = 4.49–35.50 L·s$^{-1}$·ha$^{-1}$"
- P18L27; text „q > 4.86 L s$^{-1}$ ha$^{-1}$" has been modified as follows: „$i$ > 4.86 L·s$^{-1}$·ha$^{-1}$"

**Comment #3**
     **In table 1 and figure 8 the terms "frontal, type I" and frontal, type II" are used, but they are defined first in line 18-2. Why do the authors not use the terms "cold front" and "warm front"? In table 1 the symbol M is used, but nowhere declared.**

Table 1 (P14L3) is supplemented by additional explanatory notes: frontal type I – cold front, frontal type II – warm front (see answer to Comment #5).
In response #1 there are descriptions of symbols included in the work.

**Comment #4**

   In line 7-25 CDF's "describe the probability of exceeding the flow number of storm overflow discharges." But in the paper CDF is used generally, as customary. On the contrary in picture 8b) CDF is the distribution of the rain intensity. Beside "exceeding" seems to be not correct, since CDF's represent the probability of undershooting!

Of course, we agree with the reviewer. We have corrected the use of the CDF term in the text.

**Comment #5**

   Table 1: The sequential arrangement of the variables is not perfect. The order could be all $P_{tot}$, $t_r$ , q, and M ore all annual values, convective frontal . ..

Table 1 has been modified (in accordance with the remark above) to the following form:

| Variable | Distribution | Model parameters | $p$ (KS) | $p$ (Chi) |
|---|---|---|---|---|
| $P_{tot}$ (all events) | Weibull | $\beta = 0.772$; $\gamma = 5.158$; $\mu = 3.00$ | 0.121 | 0.096 |
| $t_r$ (all events) | GEV | $\zeta = 0.466$; $\sigma = 129.355$; $\mu = 108$ | 0.096 | 0.071 |
| $i$ (all events) | log-normal | $\sigma = 1.932$; $\mu = 0.855$ | 0.112 | 0.096 |
| $M$ (all events) | Poisson | $\lambda = 32.80$ | 0.624 | 0.053 |
| $P_{tot}$ (convective) | Weibull | $\beta = 0.821$; $\gamma = 3.102$; $\mu = 3.00$ | 0.477 | 0.412 |
| $t_r$ (convective) | beta | $\alpha = 1.391$; $\beta = 1.173$; $c = 5.5$; $d = 150$ | 0.268 | 0.173 |
| $i$ (convective) | log-normal | $\sigma = 2.557$; $\mu = 0.694$ | 0.238 | 0.211 |
| $M$ (convective) | Poisson | $\lambda = 14.33$ | 0.871 | 0.756 |
| $P_{tot}$ (frontal) | Weibull | $\beta = 0.968$; $\gamma = 6.054$; $\mu = 3.00$ | 0.353 | 0.314 |
| $t_r$ (frontal) | Weibull | $\beta = 1.201$; $\gamma = 164.99$; $\mu = 150$ | 0.639 | 0.589 |
| $i$ (frontal) | log-normal | $\sigma = 1.485$; $\mu = 0.644$ | 0.906 | 0.878 |
| $M$ (frontal) | Poisson | $\lambda = 15.95$ | 0.372 | 0.831 |
| $P_{tot}$ (frontal, type I) | Weibull | $\beta = 0.862$; $\gamma = 4.535$; $\mu = 3.00$ | 0.631 | 0.425 |
| $t_r$ (frontal, type I) | beta | $\alpha = 1.221$; $\beta = 1.372$; $c = 150$; $d = 270$ | 0.200 | 0.145 |
| $i$ (frontal, type I) | log-normal | $\sigma = 1.701$; $\mu = 0.612$ | 0.104 | 0.085 |
| $P_{tot}$ (frontal, type II) | Weibull | $\beta = 1.065$; $\gamma = 7.222$; $\mu = 3.00$ | 0.397 | 0.342 |
| $t_r$ (frontal, type II) | beta | $\alpha = 0.829$; $\beta = 1.562$; $c = 266$; $d = 650$ | 0.270 | 0.226 |
| $i$ (frontal, type II) | log-normal | $\sigma = 1.289$; $\mu = 0.611$ | 0.059 | 0.056 |
| $P_{tot}$ (in convergence zone) | log-normal | $\sigma = 0.603$; $\mu = 3.00$ | 0.969 | 0.856 |
| $t_r$ (in convergence zone) | Weibull | $\beta = 0.802$; $\gamma = 276.138$; $\mu = 650$ | 0.947 | 0.879 |
| $i$ (in convergence zone) | log-normal | $\sigma = 1.296$; $\mu = 0.497$ | 0.942 | 0.923 |
| $M$ (in convergence zone) | Poisson | $\lambda = 2.55$ | 0.067 | 0.652 |

where: type I – cold front, type II – warm front.

**Comment #6**

   Figures: Partly the units and symbols are missed. The caption of figures should be understandable and clear enough without reading the text. Figure 6 and 7: It would be favourable, if the both axes would have the same range. Not every reader is experienced in such analysis. One sentence or two sentences would be useful to explain, what the background of such pictures is. Instead of "Observed Value" (y-axis) it is recommended to write "Empirical Quantile".

The caption could be: "Comparison of empirical and theoretical quantiles concerning the number of rainfall episodes and distinguishing rainfall types"

In Figures 6 and 7 (P15L14-P16) the descriptions of X and Y axes have been modified in accordance with the drawings below.
The X axis is described as:
- theoretical quantile ($P_{tot}$),
- theoretical quantile ($t_r$),
- theoretical quantile ($i$ ),
- theoretical quantile ($M$),
The Y-axis is described as:
- empirical quantile ($P_{tot}$),
- empirical quantile ($t_r$),
- empirical quantile ($i$ ),
- empirical quantile ($M$)

[Figure]

Figure 6: Comparison of empirical and theoretical quantiles concerning the number of rainfall episodes and distinguishing rainfall types: (a) all events (b) convective, (c) frontal, (d) in convergence zone.

The title of Figure 6 has been corrected according to the reviewer's remark.

[Figure]

Figure 7: Comparison of empirical and theoretical quantiles concerning $P_{tot}$ and $t_r$ values for: (a) all events, (b) convective, (c) frontal, and (d) rainfall in convergence zone.

It is possible, but unreasonable, to perform Figure 6 in such a way as to maintain an identical range of variation in the annual number of rainfall events of different genesis. This is due to the large variation in the independent variable. Identically, this issue relates to the rainfall depth and its duration.

**Comment #7**
**Figure 6: Is it right, that the sum of the highest values of b), c) and d) = 55 should be equal to the highest value of a)?**

Thank you for your valuable insight. The maximum values for the number of rainfall events caused by convective, frontal, and rainfall in convergence zone were recorded for the same year. This means that the sum does not have to be 55, but it may be less, as was obtained in the case under consideration.

**Comment #8**
**Figure 8 (a): p is the probability of overflow discharge. But for wich variable stands CDF here? The reviewer has not found any remarks. Therefore he don't understand in line 18-5 why "the percentiel value p = 0,50 is as high as 0,90"? It looks like that the CDF-value 0,50 is as high as 0,90? But what means CDF here?**

Indeed, the markings introduced are misleading. The CDF value means the probability of non-exceedance the probability of stormwater overflow by a storm overflow in a rainfall episode. Thus, further modifications have been made to the manuscript:
- P18L5 now: „percentile value p= 0.50" after modification: „percentile value 0.50"
- P19L9 now: „for example, for p= 0.05" after modification: „for example, for the percentile value 0.05"
- P19L10 now: „for the percentile value p= 0.95" after modification: „for the percentile value 0.95"
- P20L4 now: „The influence of the theoretical distribution of the number of rainfall events per year on the values of $0.99 > p > 0.50$ is confirmed by Szeląg et al. (2018)" after modification: „The influence of the theoretical distribution of the number of rainfall events per year on the values of percentiles 0.50–0.99 is confirmed by Szeląg et al. (2018)."

**Comment #9**
**Line 1-15: The Model is innovative, formulate more clear, what the reasons are**

[revised manuscript text omitted]

**Comment #11**
> Line 2-14: write "such discharges". Mostly the words "overflow discharge" are used, but in some further cases only the word "discharge" (for example 8-4 and 18-29), while "overflow discharge" is meant. Please check such cases.

The vocabulary was modified in the paper. Expression „overflow discharge" has been replaced by „storm overflow", and the expression „annual number of overflow discharges" has been replaced by „annual number of overflows".

**Comment #12**
> Line 5-22: write better "It concerns events with high intensity and short duration.

The following manuscript text (P5L22): „These data were taken into account in the conducted analyses, as the launch of a new device, the SEBA electronic rain gauge (tipping-bucket SEBA rain gauge), a few years later in the state measuring network, resulted in the recording of significantly lower precipitation levels (by several percent) high intensity and short-lived, compared to measurements recorded by a traditional pluviograph (Kotowski et al., 2011)"

has been modified as follows:

„Only these data were taken into account in the conducted analyses, as the launch of a new device, the SEBA electronic rain gauge (tipping-bucket SEBA rain gauge), a few years later in the state measuring network resulted in the recording of significantly lower precipitation levels (by several percent). The data concerns events with high intensity and short duration (Kotowski et al., 2011)."

**Comment #13**

     **Lines 7-7 to 7-11: this paragraph seems to be a repeat.**

Based on this Comment, the following manuscript text (P7L7 to P7L11): „Within the conducted analyses, an innovative probabilistic model was proposed for forecasting the number of storm overflow discharges (Figure 4). This model allows for the forecast of the annual number of discharges and the simulation of the number of events per year, taking into account the genesis of rainfall (convective in air mass, frontal, convergence zones precipitation), which is typical for countries located in central Europe and other regions of world. Although the paper focuses on the genesis of rainfall developed by Kupczyk and Suligowski (1997, 2011), the proposed approach is universal. The distribution of rainfall data may be based on local conditions determining the movement of air masses, which has a key impact on the dynamics of rainfall events. The time range of particular rainfall groups can then be determined on the basis of meteorological, synoptic and statistical analysis in the periods of high precipitation sums or precipitation intensity in a given area (Llasat, 2001; Rigo and Llasat, 2004; Millán et al., 2005; Langer and Reimer, 2007; Federico et al., 2008; Lazri et al., 2012; Berg and Haerter, 2013)."
has been modified as follows (P7L11):

„An innovative probabilistic model is proposed for modelling the annual number of storm overflows (Figure 3). This model allows for the prediction of the annual number of overflows and the simulation of the number of events per year, taking into account the genesis of rainfall, which is typical for countries located in central Europe and other regions of the world. Although the paper focuses on the genesis of rainfall developed by Kupczyk and Suligowski (1997, 2011), the proposed approach is universal. The distribution of rainfall data may be based on local conditions determining the advection of air masses, which has a key impact on the dynamics of rainfall events. The time range of particular rainfall groups can then be determined based on meteorological, synoptic and statistical analysis in the periods of high precipitation totals or precipitation intensity in a given area (Llasat, 2001; Rigo and Llasat, 2004; Millán et al., 2005; Langer and Reimer, 2007; Federico et al., 2008; Lazri et al., 2012; Berg and Haerter, 2013)."

**Comment #14**

     **Lines 8-9 to 9-2: Both sentences sound similar.**

The following sentences are indeed similar: „The paper presents the following stages of construction of a probabilistic model on the example of an urban catchment located in the area of Kielce city. In the following sections the individual steps of the above mentioned calculation algorithm of the probabilistic model (separation of rainfall events, creation of a logistic regression model, development of a rainfall generator) are discussed in detail (Figure 4).

Therefore, the text has been modified as follows (P8L9):
„The paper presents a probabilistic model for simulating the annual number of storm overflows. In this model, an innovative solution is to use the logistic regression method to analyse the impact of rainfall genesis on the functioning of a storm overflow in the example of a catchment located in the city of Kielce (central Poland)."

**Comment #15**

     **Line 9-5: What is meant by "in the ranks of"**

The following sentence (P9L5) has been modified: „One of the basic conditions allowing for the completion of a synthetic precipitation generator is the separation of single independent rain events in the ranks of rainfall"
as follows:
„One of the basic conditions allowing for the completion of a synthetic precipitation generator is the separation of single independent rain events (episodes) in rainfall time series."

**Comment #16**

> Lines 9-9 to 9-18: this paragraph seems to be a repeat.

In the first quoted sentence (P9L9) the reader was informed only about empirical distributions, determined on the basis of separated rain episodes. In the next paragraph of the text this information is detailed and the methodology of research is presented. Thus, although the sentences P9L9 and P9L18 are similar, both are of significant importance in the applied research methodology.

**Comment #17**

> Line 9-24: "simulate objects" sounds strange, write better "simulate the influence of constructions on flow procsses"

The following sentence (P9L24): „It is also used to simulate objects located in rainwater drainage networks (storm overflows) (Szeląg et al., 2018)"
has been modified as follows:
„This model is also used to simulate the influence of constructions on flow processes (Szeląg et al., 2018)."

**Comment #18**

> Line 10-14: The investigation period is 1961 to 2000 (page 5). Here the years 2012-2014 are discussed?

Description of the data used to construct the model (P10L11-15): „The second variant assumes simplification and considers a single independent variable, i.e. average rainfall intensity. To determine the logit model, the results of measurements of the operation of the investigated storm overflow have been used from the years 2009-2011, when 69 overflow of 188 precipitation events occurred, and from the years 2012-2014, when 42 overflow of 93 precipitation events occurred."
has been made more specific:
„In the urban catchment area, continuous flow measurements were carried out in the period 2008-2011 (69 storm overflows during 188 rainfall events were separated at that time), whereas in the years 2012-2014, only fillings in the diversion chamber were measured (42 overflows during 93 rainfall episodes were separated). The reason for this was the construction works carried out in the analysed catchment and a large amount of suspended solids limiting the operation of the measuring devices. Since 2015, MES1 and MES2 flow meters have been installed, which also allow the measurement of the volume of stormwater discharged by overflow. Thus, based on data from the period 2008-2014, a logit model was developed, while data from the period 2015-2017 were used to verify it."

**Comment #19**

> Line 11-14: What is meant by "period separating subsequent rainfall events"?

The following sentence (P11L14): „Currently conducted research in the field of rainfall simulators based on multidimensional boundary distributions combined with the so-called dome functions take into account the distribution of rainfall in the rainfall episode (Vernieuwe et al., 2015), spatial diversity of rainfall (Dai et al., 2014), seasons (Khedun et. al., 2014) and the period separating subsequent rainfall events (Balistrocchi and Bacchi, 2011)."
has been modified as follows:
„A literature review (Vendenberge et al., 2008) shows that the seasons of the year were taken into account in models for predicting rainfall distribution based on copula functions."

**Comment #20**

> Line 11-15 to 11-24: This part concerns not methods. Similar discussions are in the first parts of the paper.

The text from P11L15 to P11L24 and P10L17 to P10L24 is shortened and included in the introduction (P2L22) in the following form:

„Multidimensional scaling methods and fractal geometry (Rupp et al., 2009; Licznar et al., 2015; Müller-Thomy and Haberlandt, 2015) are used to simulate rainfall series. An alternative solution is an approach based on the multidimensional distributions created based on theoretical distributions and copula functions (Vandenberghe et al., 2010; Vernieuwe et al., 2015). Despite numerous applications, these solutions are relatively complex and require expert knowledge. For the storm overflow simulation, hydrodynamic models are usually used, and less frequently, empirical models are used (Szeląg et al., 2018). Nevertheless, this approach to the simulation of the annual number of overflows is very local and, in many cases, requires the construction of a catchment model."

Due to the above correction and Comment #5, the text after P9L19 was also modified:

"Taking into account the computational algorithm described at the beginning of Section 4 (Methodology), based on the determined distributions of the theoretical rainfall characteristics describing the operation of a storm overflow, a model for the simulation of a synthetic rainfall series was adopted for further analysis. The simulations carried out for this purpose included the Monte Carlo method with modification of Iman-Conover (1982). This model provides the possibility simulating the independent variables based on the determined theoretical distributions. In this method, the variability of the considered variables is described by boundary (theoretical) distributions, and the basis for the evaluation of their correlation is the Spearman correlation coefficient. The conditions that must be met in order for the results obtained to be considered correct, are as follows.

– In the data obtained from simulation and measurements, the mean values $(\mu_1(x_1), \mu_2(x_2),...,\mu_i(x_i))_s$ and the standard deviations $(\sigma_1(x_1), \sigma_2(x_2),...,\sigma_i(x_i))_s$ of the variables $(x_i)$ considered in $j$ samples do not differ by more than 5 %.
– The theoretical distributions of $x_i$ variables obtained from simulation should be consistent with those obtained from measurements; in order to meet this condition it is recommended to use the Kolmogorov-Smirnov (KS) test.
– The value of the correlation coefficient (R) between the individual dependent variables $(x_i)$ obtained for the data from Monte Carlo (MC) simulation does not differ by more than 5 % from the value of R obtained for empirical data."

**Comment #21**

**Lines 12-8 to 12-11: This sentences are nearly a repeat of pages 7/8, but the steps are not denominated identical. Here 4 steps are listed, but the Section consists of the two parts 5.1 and 5.2 only.**

Indeed, the text on page P12L8 to P12L11 is similar to this one on pages 7/8. The text (P12L8 to P12L11) lists in detail the stages that are discussed in the manuscript below.
To eliminate the similarity, the Section 5.2 has been divided into four following Sections:

5.2. Identification of the empirical and theoretical distributions of the selected rainfall characteristics
Section 5.2 includes analyses related to the determination of statistical distributions of the following variables: rainfall depth $(P_{tot})$, rainfall duration $(t_r)$, average rainfall intensity $(i)$, and number of rain events in a year of varied genesis.
Section 5.2 in the manuscript contains the text from P13L24 to P17L5.

5.3. Impact of rainfall genesis on the probability of storm overflow occurrence
Section 5.3 presents the determined relationship between the genesis of rainfall and the probability of overflow event, as well as the ranges of variation of average rainfall, taking into account the rainfall genesis, which determines the occurrence of storm overflow event.
Section 5.3 in the manuscript contains the text from P17L6 to P17L27.

5.4. Impact of rainfall genesis on the average rainfall intensity occurrence of a storm overflow
Section 5.4 presents the results of calculations of the impact of rainfall genesis on the rain intensity limit values that determine the occurrence of storm overflow. Section 5.4 includes the text from P18L3 to P18L27.

5.5. Impact of rainfall genesis on the annual number of overflow events
Section 5.5 presents the annual number of overflow events caused by rainfall (convective, frontal, and in convergence zone). In addition, a comparative analysis of the annual number of overflow events obtained with a simplified logit model (based on

the average rainfall intensity) and the number of overflow events obtained with an accurate model (based on the total amount of rainfall and its duration) is carried out in this Section.

Section 5.5 in the manuscript contains the text from P18L28 to P20L16 in the manuscript.

**Comment #22**
> **Line 13-19: The reviewer don't know what "values of free words" are? Possibly other readers will have the same problem.**

The following sentence (P13L19 – P13L21): „On the basis of the relationships (eq. 5 and eq. 6) it can be concluded that the values of free words obtained in them are similar to the weighted average value of the catchment retention ($d_{av}$). The relative difference between the values of free words and retention of the catchment area does not exceed 5 % "
has been modified as follows:
„On the basis of these relationships, it can be concluded that the values of the intercept obtained in them are similar to the weighted average value of the catchment retention ($d_{av}$). The relative difference between the values of intercept and retention of the catchment area does not exceed 5 %."

**Comment #23**
> **Line 1-16: The text within the brackets should be formulated as sub-clause or as an additional sentence.**

The manuscript text has been corrected (see Comment and Response #9).

**Comment #24**
> **Line 1-21: write "great" instead of "big"**

The amended sentence is set out in answer to Comment #10.

**Comment #25**
> **Line 1-29: two times determine**

The following sentence (P1L29): „On the basis of the obtained results, the range of variability of average rainfall intensity was determined, which determines the discharge by storm overflow, as well as the annual number of discharges resulting from the occurrence of rain of different genesis."
has been modified as follows:
„Based on the obtained results, the range of the variability of the average rainfall intensity, which determines the storm overflow, and the annual number of overflows resulting from the occurrence of rain of different genesis were defined."

**Comment #26**
> **Line 1-31: the results are suited for implementation**

The following sentence (P1L31): „The obtained results enable their practical implementation in the assessment of storm overflows only on the basis of knowledge concerning the genetic type of rainfall."
has been modified as follows:
„The results are suited for the implementation in the assessment of storm overflows only based on the genetic type of rainfall."

**Comment #27**
> **Line 2-2: three times the word "of" in series.**

The following sentence (P2L2): „They can be used to develop warning systems, in which information on the predicted rainfall genesis is a component of the assessment of the operation of the stormwater system and the facilities located on it."
has been modified as follows:

„The results may be used to develop warning systems in which information about the predicted rainfall genesis is an element of the assessment of the rainwater system and its facilities."

**Comment #28**
   **Line 2-29: what was not taken into account when rainfall generators were used to simulate**

The following sentence (P2L29): „It seems puzzling why the fact that the time course and dynamics of rainfall are the result of complex movements of air masses (Serrano et al., 2009; Alhammoud et al., 2014; Dayan et al., 2015) was not taken into account when modelling rainfall generators to simulate storm overflows."
has been modified as follows:
„It seems puzzling that the time course and the dynamics of the rainfall as the result of air masses advection (Serrano et al., 2009; Alhammoud et al., 2014; Dayan et al., 2015) were not taken into account when rainfall generators were used to simulate storm overflows."

**Comment #29**
   **Line 2-32: "concern simulations" sounds strange, write perhaps better "consider"**
   **Line 3-1: "course of precipitation phenomena" sounds strange.**

The following sentence (P2L32): „ The models created concern simulations of meteorological conditions changing in time and determining the distribution of temperature, pressure and humidity, which affects the dynamics of air movement and, consequently, the course of precipitation phenomena."
has been modified as follows:
„The models created consider simulations of meteorological conditions changing in time and determining the distribution of temperature, pressure and humidity, which affects the dynamics of air advection and, consequently, the patterns of precipitation phenomena."

**Comment #30**
   **Line 3-4: "forecasting the operation" sounds strange, write "basis for the control of systems"**

The following sentence (P3L4): „This information may be the basis for forecasting the operation of the stormwater system and developing an early warning system against the risks of flash flood."
has been modified as follows:
„This information may be the basis for control of the systems and the development of an early warning system against the risks of flash floods."

**Comment #31**
   **Line 3-11: Sometime it is written "model of the rainfall generator". The generator is a model, the word "model" seems to be unnessecary.**

The following sentence (P3L11): „In the model of the rainfall generator the genesis of rainfall was taken into account, which allowed to determine the curves showing the influence of rainfall genesis on the occurrence of overflow discharge in a single rainfall episode."
has been modified as follows:
„In the rainfall generator, the genesis of rainfall is taken into account, which allows determining the curves that show the influence of rainfall genesis on the occurrence of storm overflow in a single rainfall episode."

**Comment #32**
   **Line 3-22: a space is missed**

A space is inserted between the sentences (P3L22): "... (Szeląg et al., 2016). The length …."

**Comment #33**

      **Line 3-23: better "height difference", cancel "of ordinates"**

The following sentence (P3L23): „The maximum difference of ordinates in the catchment is 12.0 m and the average slope in the catchment is 7.1 %."

has been modified as follows:

„The height difference in the catchment is 12.0 m and the average slope is 7.1 %."

**Comment #34**

      **Line 4-10: "generated" better as "shaped"**

The sentence (P4L10) has been removed.

**Comment #35**

      **Line 4-11: write "A third", since before only two mechanism are announced**
      **Line 4-11: write "which include both above mentioned components"**

The sentence (P4L11) has been removed.

**Comment #36**

      **Line 4-16: write at the end "are" instead of "is"**

The following sentence (P4L16): „Convective precipitation induced by single thunderstorm cells, their complexes or squall lines is short-lived but is characterized by high average intensity (Kane et al., 1987) and causes flash floods (Gaume et al., 2009; Marchi et al., 2010; Bryndal, 2015)" has not been modified.

**Comment #37**

      **Line 5-2 and 5-13: line break (new paragraph)**

We agree with the Reviewer's #1 comment that the considerations on this subject presented in Section 3 may be shortened and partly moved to the introductory Section of the article.

Therefore, the following changes have been made to the manuscript:

– the title of Section 3 (P4L7) has been changed as follows: "Rainfall data and analysis*"*

– the previous content presented in the text from P4L8 to P5L17 was modified in the following way and moved to Section 1 (after P3L7):

"Rainfall is universally classified into three types (Sumner, 1988; Smith, 1993): convective, cyclonic and orographic. The main distinguishing feature between convective precipitation in an air mass and frontal precipitation in mid-latitudes is its spatial extent and duration. The range of convective precipitation associated with local air circulation is much smaller than in the case of travelling extratropical cyclones with weather fronts. Convective precipitation induced by single thunderstorm cells, their complexes or squall lines is short-lived but is characterized by high average intensity (Kane et al., 1987) and causes flash floods (Gaume et al., 2009; Marchi et al., 2010; Bryndal, 2015). On the other hand, the lifespan of the mechanisms of creating cyclonic precipitation is much longer than that of convective precipitation – on the order of days rather than hours. Hence, the effect of this is long-term rainfall with a high depth (Frame et al., 2017) often cause regional floods (Barredo, 2007). The presented classification of precipitation types distinguished by Sumner (1988) due to the origin, developed for the British Isles and Western Europe, cannot be directly applied in practical hydrology in other regions of the continent, especially in its eastern and central parts due to exceptional variability of meteorological conditions occurring in the temperate zone of the warm transition climate – on the border of air masses coming from the Atlantic and continental masses from the east (Twardosz and Niedźwiedź, 2001; Niedźwiedź et al., 2009; Twardosz et al., 2011; Łupikasza, 2016). The analysis of maximum rainfall of different duration in Poland carried out at the end of the 1990s (Kupczyk and Suligowski, 1997, 2011), supplemented by an analysis of the synoptic situation (based on surface synoptic charts of Europe, published in the Daily Meteorological Bulletin

of the Institute of Meteorology and Water Management – IMGW in Warsaw) and a calendar describing the types of atmospheric circulation together with air masses and air fronts (Niedźwiedź, 2019), led to the separation of three types of genetic precipitation: convective in air mass, frontal and generated in convergence zone."

**Comment #38**
   **Line 5-17: "convergence zone" is not a type of precipitation, write better "generated in convergence zones"**

The amended text is presented at the end of the answer to Comment #37.

**Comment #39**
   **Line 5-19: write "only these data were"**

Fragment of the sentence (P5L19): „These data were taken into account in the conducted analyses,…."
has been modified as follows:
„Only these data were taken into account in the conducted analyses,…."

**Comment #40**
   **Line 5-33: "variable" instead of "varied"**

The following sentence (P5L33): „The second type (frontal rainfall) forms a group of precipitation in Kielce, in which the duration is very varied and ranges from 2.5 h to 10.5 h."
has been modified as follows:
„The second type (frontal rainfall) forms a group of precipitation in Kielce, in which the duration is very variable and ranges from 2.5 h to 10.5 h."

**Comment #41**
   **Line 6-12: "precipitation emitted" sounds strange.**

The following sentence (P6L12): „Transformation of air masses over the western part of the continent, lower speeds of movement of frontal zones, as well as weakening of the dynamics of processes in the front zone cause that precipitation in Kielce differ in intensity and duration in relation to precipitation emitted by Sumner (1988) as cyclonic."
has been modified as follows:
„The transformation of the air masses over the western part of the continent, the lower speeds of the air advection and the weakening of the dynamics of processes in the front zone cause precipitation in Kielce to differ in intensity and duration in relation to precipitation defined by Sumner (1988) as cyclonic."

**Comment #42**
   **Line 7-25: overflow discharges per year**

The following sentence (P7L25): „On this basis, distribution functions (CDF) are determined that describe the probability (Z) of exceeding the number of storm overflow discharges."
has been modified as follows:
„On this basis, distribution functions (CDF) are determined that describe the probability of non-exceeding the number of storm overflows."

**Comment #43**
  **Line 10-10: Write ". is a simplification. It considers only a single"**

The following sentence (P10L10): „The second variant assumes simplification and considers a single independent variable, i.e. average rainfall intensity."
has been modified as follows:
„The second variant is a simplification. This variant considers only a single independent variable, i.e., average rainfall intensity."

**Comment #44**
  **Line 10-18: set methods before the brackets.**

The above remark has already been taken up in answer to Comment 20 and the quotes have been moved to the end of the sentence.

**Comment #45**
  **Line 10-30: write better "should be consistent"**

The following sentence (P10L30): „…theoretical distributions of $x_i$ variables obtained from simulation are consistent with those obtained from measurements; in order to meet this condition it is recommended to use the Kolmogorov-Smirnov test…"
has been modified as follows:
„… theoretical distributions of $x_i$ variables obtained from simulation should be consistent with those obtained from measurements; in order to meet this condition it is recommended to use the Kolmogorov-Smirnov (KS) test.…"

**Comment #46**
  **Line 13-16: write "are valid" instead of "take place"**

The following sentence (P13L16): „Based on theoretical considerations conducted by Thorndahl and Willems (2008), who provided a generalised model for forecasting the volume of wastewater discharge via a storm overflow, it can be concluded that in this case the following relations take place:"
has been modified as follows:
„Based on theoretical considerations of Thorndahl and Willems (2008), who investigated the relationships between the characteristics of catchment areas, including the limit retention that determines the volume of overflow, the following equations were determined:"

**Comment #47**
  **Line 13-19: write "of this relationship" and cancel the "(eq. 5 and eq. 6)"**

The corrected sentence is in answer to Comment 22.

**Comment #48**
  **Line 15-8: expressed better by**

The following sentence (P15L8): „Also, the variation in rainfall duration in episodes resulting from rainfall of different genesis in most cases is described by the Weibull distribution and only in the case of data measured over an annual cycle is it expressed by the GEV distribution (eq. 9)."
has been modified as follows:
„In addition, the variation in rainfall duration of the episodes resulting from rainfall of different genesis is described, in most cases, by the Weibull distribution and only in the case of the data measured over an annual cycle is it expressed better by the GEV distribution (eq. 11)."

**Comment #49**

      **Line 18-3: Write "distinguished" instead of "made".**

The following sentence (P18L3): „Within the framework of the conducted analyses, the division of frontal rainfall events of the duration not longer than 4.5 h (related to the cold front – type I) and exceeding the given value (due to the displacement of the warm front – type II) was additionally made."
has been modified as follows:
„Within the framework of the conducted analyses, the division of frontal rainfall events of the duration not longer than 4.5 h (related to the cold front – frontal type I) and exceeding the given value (due to the displacement of the warm front – frontal type II) was additionally distinguished."

**Comment #50**

      **Line 18-6: line break (new paragraph).**

Section 5.2 and its division is discussed in answer to Comment #21.

**Comment #51**

      **Following words seems to be unnessecary: Line 1-20: was demonstrated; Line 1-31/32: knowledge concerning; Line 2-10: collecting; Line 2-19: in the work; Line 2-22: of simulation; Line 2-27: in its modeling; Line 3-26: the work; Line 4-6: article; Line 4-17: in many areas; Line 5-3: of the phenomenon; Line 9-6: in the paper; Line 10-10: in the analysis performer; Line 11-13: in the rainfall episode; Line 13-1: using the model.**

All words indicated by the reviewer have been deleted.

[revised manuscript text omitted]

---

## Author Response (AR2)

**Dear Professor Dr. Markus Weiler**

Thank you very much for handling our paper. Following the review process, our paper has improved greatly and we are appreciative.

We have improved the terminology (rainfall episode was replaced by rainfall event) throughout the manuscript. Also we have made the suggested correction to the fig. 6 (the regression line was replaced by the 1:1 line).

We look forward to hearing about the final status of the paper.

Sincerely,
Bartosz Szeląg

[revised manuscript text omitted]